# Sequence anticipation and spike-timing-dependent plasticity emerge from a predictive learning rule

Matteo Saponati [1,2,3] ✉ & Martin Vinck [1,3] ✉

Intelligent behavior depends on the brain's ability to anticipate future events. However, the learning rules that enable neurons to predict and fire ahead of sensory inputs remain largely unknown. We propose a plasticity rule based on predictive processing, where the neuron learns a low-rank model of the synaptic input dynamics in its membrane potential. Neurons thereby amplify those synapses that maximally predict other synaptic inputs based on their temporal relations, which provide a solution to an optimization problem that can be implemented at the single-neuron level using only local information. Consequently, neurons learn sequences over long timescales and shift their spikes towards the first inputs in a sequence. We show that this mechanism can explain the development of anticipatory signalling and recall in a recurrent network. Furthermore, we demonstrate that the learning rule gives rise to several experimentally observed STDP (spike-timing-dependent plasticity) mechanisms. These findings suggest prediction as a guiding principle to orchestrate learning and synaptic plasticity in single neurons.

Predicting the future is pivotal in guiding interactions with the world, for example in reward learning[1,2] and in action planning[3]. Predicting future states entails that a system can anticipate and signal events ahead of time. Indeed, there is evidence for anticipatory neural activity in various brain systems[4–10]. Furthermore, the predictability of sensory events can evoke different neuronal signals, in particular enhanced firing rates for surprising inputs, which may guide the update of model predictions in other brain areas[10–13]. Yet, the associations among sensory events and their predictability should not only result in specific patterns of neural activity, but should also have specific consequences for synaptic plasticity and neuronal outputs[14].

In particular, one would expect that synaptic inputs that carry much information about the future receive high credit, whereas those synaptic inputs that are redundant and predicted by other inputs are downregulated. We conceptualize this credit assignment as a form of predictive plasticity. As a consequence of credit assignment to predictive synaptic inputs, neurons might learn to anticipate and signal future events that are predicted, which can

then lead to the adaptive behavior of the organism. Importantly, predictive relationships between events must eventually lead to plasticity formation at the level of a single neuron, which receives a limited set of inputs. However, it remains unclear how synaptic plasticity formation in individual neurons relates to predictive processing. Experimental evidence suggests numerous and complex synaptic plasticity mechanisms for single neurons, e.g. heterosynaptic plasticity[15–17], spike-timing-dependent plasticity (STDP)[18–20] and homeostatic plasticity[21,22]. These experimental studies have shown that synaptic adjustment is sensitive to the relative firing times of pre-synaptic inputs, the temporal relation between pre- and post-synaptic firing, and that neurons can simultaneously orchestrate plasticity at multiple synapses. These plasticity mechanisms greatly enrich the computational capabilities of neurons[23] and may underlie the biological substrate for the association between events across long temporal sequences. They may further account for the observation that repeated sequential activity is associated with subsequent recall or replay of sequences

[1]Ernst Strüngmann Institute (ESI) for Neuroscience in Cooperation with Max Planck Society, 60528 Frankfurt Am Main, Germany. [2]IMPRS for Neural Circuits, Max-Planck Institute for Brain Research, 60438 Frankfurt Am Main, Germany. [3]Donders Centre for Neuroscience, Department of Neuroinformatics, Radboud University, 6525 Nijmegen, The Netherlands. ✉e-mail: matteo.saponati@esi-frankfurt.de; martin.vinck@esi-frankfurt.de

at compressed time scales. Yet, a computational understanding of how these plasticity processes may contribute to the prediction of the future has to be reached.

We hypothesized that predictive plasticity may account for the existence of learning processes inside individual neurons, allowing neurons to learn temporal sequences and anticipate future events. We recapitulate this predictive mechanism as a spiking neuron model, where the cell anticipates future inputs by learning a low-dimensional model of its high-dimensional synaptic inputs. Based on this principle, we derive a predictive learning rule. We show how single neurons can learn to anticipate and recall sequences over long timescales, and that the described learning rule gives rise to several experimentally observed STDP mechanisms.

## Results

### Model of prediction at the single neuron level

We formalized the proposed predictive process in the following single-neuron model: In this model, at each moment in time $t$, the neuron integrates the present pre-synaptic inputs in the current state of the membrane potential and extracts from its dynamics a prediction of the future input states (see the Methods section for a detailed account of the model and analytical derivations). We first defined the membrane potential $v_t$ as a linear filter, such that the neuron updates its membrane potential recursively by encoding the actual input at time $t$ and the previous value of the membrane potential at time $t-1$ (Equation (1)). The membrane potential at a given time is the result of the temporal summation of previous synaptic inputs, and the membrane potential thereby encodes a compression of the high-dimensional input dynamics in time. This is described by the system of equations

$$\begin{cases} v_t = \alpha v_{t-1} + \mathbf{w}_t^\top \mathbf{x}_t - v_{\text{th}} s_{t-1} \\ s_t = H(v_t - v_{\text{th}}). \end{cases} \tag{1}$$

Here, the temporal integration of the inputs $\mathbf{x}_t$ is weighted by a synaptic weight vector $\mathbf{w}_t$, which gives different credit to different synapses. Together with the recurrent dynamics of the membrane voltage, we set a spiking threshold in Equation (1). Accordingly, if the membrane potential reaches the threshold $v_{\text{th}}$ at time step $t-1$, the cell fires a post-synaptic spike $s_t$ and the voltage is decreased by $v_{th}$ at the next timestep.

The objective of the neuron is to recursively compute a local prediction of its own inputs by using the temporal relations in the input spike trains. The prediction of the incoming pre-synaptic input at time step $t$ is given by the weight of the associated synapse and the previous state of the membrane potential, i.e.

$$\mathcal{L} \equiv \sum_{t=0}^{T} \mathcal{L}_t = \sum_{t=0}^{T} \frac{1}{2} ||\mathbf{x}_t - v_{t-1} \mathbf{w}_{t-1}||_2^2. \tag{2}$$

We then derived a predictive learning rule analytically by minimizing the mismatch between the actual input and the prediction. This mismatch can be interpreted as a prediction error, which can be computed with information available within the neuron and in real-time based on the dynamics of the inputs (see Methods). By letting the synaptic weights $\mathbf{w}_t$ evolve in real-time with the dynamics of the input, we obtained our predictive plasticity rule

$$\mathbf{w}_t = \mathbf{w}_{t-1} + \eta(\boldsymbol{\epsilon}_t v_{t-1} + \mathcal{E}_t \mathbf{p}_{t-1}). \tag{3}$$

Here, $\eta$ defines the timescale of plasticity, $\mathbf{p}_{t-1}$ is an input-specific eligibility trace (see Methods), $\boldsymbol{\epsilon}_t$ is the prediction error

$$\boldsymbol{\epsilon}_t \equiv \mathbf{x}_t - v_{t-1} \mathbf{w}_{t-1}, \tag{4}$$

that defines the sign and amplitude of plasticity, and $\mathcal{E}_t$ is a global signal

$$\mathcal{E}_t = (\boldsymbol{\epsilon}_t^\top \mathbf{w}_{t-1}), \tag{5}$$

given by the weighted sum of the prediction errors at each synapse. Consequently, synaptic weights undergo potentiation or depotentiation depending on the predictability of the inputs. Thus, a synapse gets respectively potentiated or suppressed if the associated input anticipates or is anticipated by other pre-synaptic inputs.

The computational steps of the predictive neuron model are as follows: (1) At each time step $t$, the objective function $\mathcal{L}_t$ is evaluated as the neuron learns to predict the current input (Fig. 1a, top); (2) the prediction error $\epsilon_t$ is used to drive plasticity and update the synaptic weights via Equation (3), and the current input $\mathbf{x}_t$ is encoded by updating the state variables of the neuron (Fig. 1a, bottom). The rule is composed of three terms: (1) A first-order correlation term $\mathbf{x}_t v_{t-1}$; (2) a heterosynaptic term $-v_{t-1}^2 \mathbf{w}_{t-1}$[16,17], which stabilizes learning[24–26] as has been observed experimentally[27–30]; (3) a global signal $\mathcal{E}_t$ that depends on synaptic variables and on the post-synaptic voltage. Accordingly, the prediction of future inputs can be computed at the synaptic level in the point-neuron approximation based only on information available within the single neuron. On a long timescale, the neuron learns a specific set of synaptic strengths by adjusting the synaptic weight continuously as it collects evidence in its membrane potential.

To illustrate the development of anticipatory firing for a simple example, we exposed the neuron to a sequence of two input spikes coming from two different pre-synaptic neurons that fire with a relative delay of 4 ms (Fig. 1b). In this simple scenario, the first pre-synaptic input is predictive of the following pre-synaptic input and should thus be potentiated, driving the neuron to fire ahead of the EPSP (excitatory post-synaptic potential) caused by the second input spike. We trained the model by repeating the input pattern for 300 epochs of duration $T = 500$ ms. During the training period, the neuron learns to adjust its output spike time and to eventually fire ahead of the pre-synaptic input 2, which arrives at 6 ms (Fig. 1b). The neuron converges onto an anticipatory "solution" by a selective adjustment of the synaptic weights (Fig. 1c, top). In particular, the neuron assigns credit to the pre-synaptic input 1, which arrives at 2 ms, and de-potentiates the strength of the input arriving at 6 ms. Accordingly, this leads to the anticipation of the predictable input (Fig. 1c, bottom). We further observed that the parameter space given by $(w_1, w_2)$ is partitioned in different regions depending on the number of spikes fired by the post-synaptic neuron (Fig. 1d). The symmetry of the weight dynamics is broken when the membrane potential reaches the threshold and an output spike is fired (Fig. 1d). The learning dynamics are qualitatively the same when the initial conditions lie in regions of multiple output spikes (Fig. S1).

### Prediction of temporal structures in the input spike trains

The simple case of two pre-synaptic inputs in a fixed sequence shown in Fig. 1b–d suggests that in the predictive plasticity model, the neuron can predict future inputs and generate anticipatory signals. However, in the brain, neurons receive many hundreds of synaptic inputs, yielding high-dimensional sequences that may be embedded in background, stochastic firing[31]. To investigate a relatively complex scenario, we considered a temporal sequence determined by $N$ presynaptic neurons that fire sequentially with fixed delays (Fig. 2a). To produce stochastic firing patterns, we used two types of noise sources, namely jitter of spike times in the sequence and random firing of the pre-synaptic neurons. The sequences were also embedded into a higher-dimensional input pattern, where $N$ additional pre-synaptic neurons fired randomly according to a homogeneous Poisson process. For each epoch, the population firing rate of the pre-synaptic inputs was constant across time, indicating that the sequence could not be detected based on the population firing rate alone (Fig. 2a). In addition,

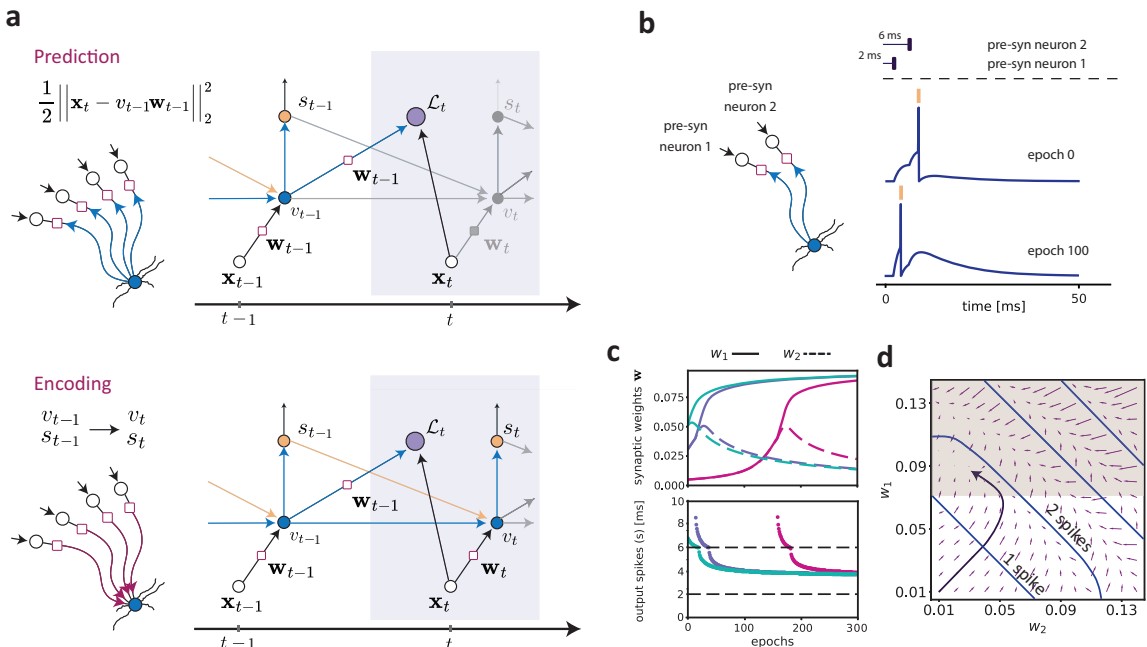

**Fig. 1 | Description of the predictive plasticity rule. a** Illustration of the model and the computational graph corresponding to the learning algorithm. Top: at time step $t$, the neuron computes a prediction of the new input $\mathbf{x}_t$ from the previous membrane potential $v_{t-1}$ and synaptic weight vector $\mathbf{w}_{t-1}$ (see Equation (2)). The prediction error is used to drive synaptic plasticity and update the synaptic weight vector $\mathbf{w}_{t-1}$ (see Equation (3)). Bottom: the neuron updates its membrane potential by encoding the actual input $\mathbf{x}_t$ via the learned weight vector $\mathbf{w}_t$ and its previous internal state $v_{t-1}$ (see Equation (1)). If the voltage exceeds the threshold, an output spike is emitted (shown in yellow) and this spiking event reduces the membrane potential by a constant value at the next time step. Otherwise, the value of the membrane potential $v_t$ is kept and passed to the next time step. **b** In the simulation illustrated here, we considered a pattern of two pre-synaptic spikes from two different pre-synaptic neurons with a relative delay of 4 ms. Shown are the dynamics of the membrane potential at the first training epoch and after 100 iterations. The

neuron learns to fire ahead of the input that arrives at 6 ms (i.e. pre-syn neuron 2). **c** Top: Dynamics of the weights for different initial conditions (i.e. the weights at epoch 0). The unbroken and dashed lines correspond, respectively, to the pre-synaptic inputs arriving at 2 ms ($w_1$, pre-synaptic neuron 1) and 6 ms ($w_2$, pre-synaptic neuron 2). Bottom: evolution of the output spike times across epochs. The bottom and top plot have the same color code. **d** The flow field in the parameter space was obtained by computing the difference between the weight vector ($w_1, w_2$) in the first epoch and after 10 epochs. The blue lines represent the partition given by the number of spikes that are fired. Note that when the synaptic weights are larger, the neuron fires more spikes. The black arrow shows the trajectory of the weights obtained by training the model for 500 epochs with initial conditions $\mathbf{w}_0 = (0.005, 0.005)$. The shaded region shows the section of the parameter space where the neuron fires ahead of the input at 6 ms from neuron 2.

the onset of the input sequence was random during each training epoch (Fig. 2a). Because of these sources of noise and jitter, the post-synaptic neuron received different realizations of the input pattern for each training epoch. We numerically solved the learning dynamics and studied the output spike pattern during learning.

We observed that during the first presentation of the stimulus, the neuron fired randomly for the entire duration of the epoch. Subsequently, the predictive learning mechanism led to structured output spike trains, and the neuron started to group its activity earlier in time, such that it eventually learned to fire for the first inputs in the sequence (Fig. 2b). During learning, the neuron kept a low output firing rate that reflects its selectivity (Fig. 2b, bottom plot). The anticipation of the pre-synaptic pattern is driven by the update of the synaptic weights (Fig. 2c). Initially, the neuron assigns uniform credit to all the pre-synaptic inputs, while firing randomly across the entire sequence. Subsequently, the neuron potentiates the inputs that anticipate the ones that are driving post-synaptic spikes, eventually assigning the most credit to the first inputs in the sequence. Because the learning dynamics follow the direction of reducing the overall prediction error, the objective function $\mathcal{L}_{norm}$ decreased across epochs (Fig. 2d, left). Furthermore, during learning, the total amount of depolarization across one stimulus presentation is reduced (Fig. 2d, right).

Further analyses demonstrate that the neuron model was able to predict and anticipate input sequences for a substantial range of model parameters (Fig. S2a, b). First, we found that the main results do not depend on the initial weight vector (Fig. S3a). Second, we show

that anticipatory firing emerges even for longer sequences (Fig. S3b) and increased noise amplitude and number of distractors (Fig. S3c). Third, the noise source does not qualitatively affect the behavior of the model across training epochs (Fig. S4a–d). Finally, we considered the case where the input pattern is composed of different sub-sequences which were spaced in time and belonged to independent subsets of pre-synaptic neurons. We show that the neuron exhibits anticipatory firing also in case of multiple sub-sequences (Fig. S5).

Together, these results show that an Integrate-and-Fire-like neuron with a predictive learning rule can learn to anticipate high-dimensional input sequences over short and long timescales. The neuron effectively uses the timing of each input spike and its temporal context across the spike pattern in a self-supervised manner. A synapse gets potentiated if, on average, the corresponding pre-synaptic input anticipates successive inputs that initially trigger post-synaptic spikes. The predictive plasticity mechanism relies solely on the temporal relation between inputs, it does not depend on initial conditions, and it is robust to several pattern disruptions. The learned solution of anticipating the input sequence thus decreases the number of fired spikes and the energy consumed by the neuron, which can be understood as a form of efficient coding[32,33].

## Sequence anticipation and recall in a network with recurrent connectivity
In the previous section, we studied the emergence of anticipatory firing in a single neuron receiving many pre-synaptic inputs. However, in

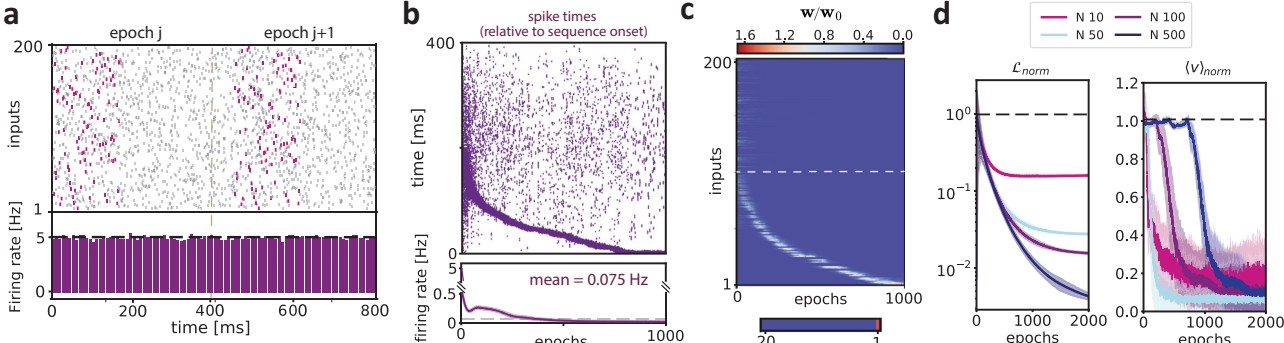

**Fig. 2 | Anticipation of spiking sequences. a** Top: Example spike sequence during different training epochs. A spiking sequence is defined by the correlated activity of a subset ($N = 100$) of pre-synaptic neurons. These $N$ pre-synaptic neurons fire sequentially with relative delays of 2 ms, resulting in a total sequence length of 200 ms (pink spike pattern). In each epoch, there are three different sources of noise: (1) jitter of the spike times (random jitter between -2 and 2 ms); (2) random background firing following a homogeneous Poisson process with rate $\lambda$ distributed between 0 and 10 Hz (see Methods); (3) another subset of 100 pre-synaptic neurons that fired randomly according to a homogeneous Poisson process with randomly distributed rates between 0 and 10 Hz. For each training epoch, the onset of the spike sequences is drawn from a uniform distribution with values between 0 and 200 ms. The bottom plot shows the population firing rate over 10 ms time bins (neuron membrane time constant). **b** Dynamics of the post-synaptic spiking activity during learning. The spike times are defined relative to the actual onset of the sequence in each respective epoch. The bottom plot shows the neuron's output

firing rate within each training epoch. This firing rate was computed across 100 independent simulations (shown are mean and standard deviation). **c** Top: Dynamics of the normalized synaptic weights $\mathbf{w}/\mathbf{w}_0$ as a function of the training epochs. Here $\mathbf{w}_0$ is the weight vector in epoch 0. Above the dashed white line are the 100 background pre-synaptic neurons that do not participate in the sequence. The synaptic weights are ordered along the y-axis from 1 to 100 following the temporal order of the sequence. Bottom: normalized weights of the first 20 inputs at epoch 1000, showing only the first input has been assigned credit. **d** Left: Normalized objective function $\mathcal{L}_{norm}$ (left plot) as a function of the training epochs. Different colors correspond to a different number of neurons participating in the sequence. Right: normalized cumulative membrane potential $\langle v \rangle$. The cumulative membrane potential was computed as the sum of the $v_t$ at each time step in the simulations. The panels show the mean and standard deviation computed over 100 different simulations.

cortical networks, each neuron may receive a large set of pre-synaptic inputs from other areas, as well as recurrent inputs from neurons in the same local network. We, therefore, investigated a more complex scenario of a network of recurrently coupled neurons that were endowed with a predictive learning rule. Our simulations were inspired by experimental observations of recall and spontaneous replay after learning. For example, a previous study in rat V1 has shown that the repeated presentation of a sequence of flashes (at different retinotopic locations) gradually leads to a reorganization of spiking activity in the order of the presented sequence[7]. The same study also showed that the presentation of only the first stimulus in the sequence leads to a compressed recall of the entire sequence[7]. Likewise, the sequential activation of neurons in the prefrontal cortex and hippocampus is known to lead to subsequent replay at a compressed timescale[34–38]. These findings have been interpreted in terms of a local reorganization of synaptic weight distributions as a result of repeated activation with an input sequence[7,34,35,37,38].

We wondered if a network of recurrently connected neurons with the predictive learning rule described above can develop sequence anticipation as well as (stimulus-evoked) sequence recall and spontaneous replay. We explored the dynamics of a network model where 10 neurons received an input sequence distributed across 80 external units. We defined the timing of external inputs to each neuron in the network, and the recurrent connections between neurons following a simplified retinotopic structure with recurrent excitation between nearby receptive fields. In particular, each neuron in the network received a unique set of inputs from 8 pre-synaptic neurons, which fired sequentially and exhibited stochastic background firing (purple and black in Fig. 3a, respectively). The neurons in the network were activated sequentially, such that the inputs into the first neuron arrived earliest, the inputs into the second neuron arrived slightly later, etc. (Fig. 3a). The network had a recurrent, nearest-neighbor connectivity scheme, such that each $n$-th neuron was connected to the neighboring $n-1$-th and $n+1$-th neuron (Fig. 3a). Thus, each neuron received a set of "afferent" pre-synaptic inputs together with the inputs from neighboring neurons (Fig. 3a). Both the synaptic connections from the

afferent inputs and the recurrent inputs were adjusted by plasticity according to the predictive learning rule described above. The recurrent connections between neurons in the network have a discontinuous effect in time - at the moment of the output spikes - and thus their contribution to the gradient can be neglected (see Methods). We show the activity of the network for three cases in Fig. 3b: (1) The "before" case, where only the pre-synaptic neurons corresponding to the first neuron in the network exhibited sequential firing. In this case, the background stochastic firing was still present in all 80 pre-synaptic neurons. (2) The "learning" or conditioning case, where we presented the entire sequence (which is repeated 2000 times). (3) The "after" or "recall" condition, which was the same as the before condition, but after learning. We observed that in the before condition, the network activity was relatively unstructured, with firing occurring in the period after the sequence due to the background stochastic firing. During learning, the neurons were active during a relatively long part of the sequence and showed a sequential activation pattern. After learning, the network showed sequential firing upon the presentation of the inputs to the first neuron in the network in the order of the sequence. This sequential firing took place at a compressed timescale. We found that the recall effect was due to the potentiation of the inputs from the $n-1$-th neuron to the $n$-th neuron, as well as potentiation of the first pre-synaptic inputs to the first neurons (Fig. 3c). Finally, we observed that sequential firing could also be triggered spontaneously due to the background stochastic activity of the pre-synaptic neurons (Fig. 3b; after-spontaneous). Thus, the network exhibited a form of activity that resembles the spontaneous replay of sequences.

We further characterized the evolution of the network's output during learning and the reconfiguration of synaptic weights. We found that after several hundreds of epochs, the network converged onto a stable, sequential output that was time-compressed (Fig. 3d). We furthermore quantified the number of neurons that needed to be activated in order for the network to recall the full sequence. We found that this required number of neurons decreased gradually across epochs, indicating a gradual reorganization of the synaptic weight distribution during learning (Fig. S6).

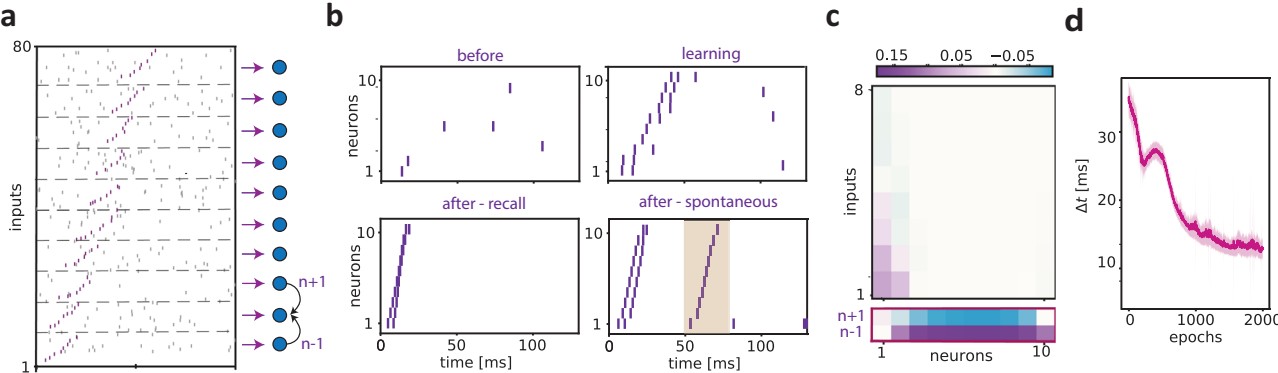

**Fig. 3 | Sequence anticipation and recall in a network with recurrent connectivity. a** In this example, we simulated a network of 10 neurons with nearest-neighbor recurrent connectivity, that is, each neuron $n$ in the network received inputs from the $n-1$-th and $n+1$-th adjacent neurons. The first and the last neuron only received inputs from the second and second last neurons in the network, respectively. Shown are the connections to the second neuron. Each neuron in the network received inputs from 8 pre-synaptic neurons that fire sequentially with relative delays of 2 ms, resulting in a total sequence length of 16 ms (pink spike pattern). The sequence onset of pre-synaptic inputs for the $n+1$-th neuron started 4 ms after the sequence onset for the $n$-th neuron in the network, etc. Each epoch contained two different sources of noise: (1) random jitter of the spikes in the sequence (between $-2$ and $2$ ms); random background firing of the pre-synaptic neurons according to a homogeneous Poisson process with rate $\lambda = 10$ Hz. Both the connections from the pre-synaptic neurons to the neurons in the network and the connections between the neurons in the network were plastic and modified according to the predictive learning rule described in the main text. **b** Raster plot of the network's activity during different epochs of training: (1) The "before" case, where only the pre-synaptic neurons corresponding to the first neuron in the network exhibited sequential firing. In this case, the background stochastic firing was still present in all the $8 \times 10 = 80$ pre-synaptic neurons. (2) The "learning" or conditioning case, where we presented the entire sequence (which was repeated 2000 times). (3) The "after" or "recall" condition, which was the same as the before condition (now after learning). (4) Same as (3), but an example where spontaneous recall occurs due to the background stochastic firing. The neurons are ordered as in panel **a**. **c** The synaptic weights matrix obtained at the end of training (epoch 1000). Top: The $i$-th column corresponds to the synaptic weights learned by the $i$-th neuron in the network, where the 8 entries correspond to the synaptic weights for the pre-synaptic inputs. Bottom: the nearest-neighbor connections in the network towards the $i$-th neurons. Note that the first and last neurons do not receive inputs from the $n-1$-th and $n+1$-th neurons, respectively. **d** Evolution of the duration of network activity across epochs. We computed the temporal difference between the last spike of the last neuron and the first spike of the first neuron to estimate the total duration of the network's activity. We computed the average duration and the standard deviation from 100 simulations with different stochastic background firing and random jitter of the spike times.

To generalize these findings, we studied a network with all-to-all connectivity, i.e. each neuron was recurrently connected to all the other neurons in the network. In this case, the network also learned to recall the full sequence on a relatively fast timescale (Fig. S7). The output of the network with all-to-all connectivity however differed from the example with recurrent connectivity between neighbors: After prolonged learning, the other neurons in the all-to-all network all fired shortly after the first neuron was activated (Fig. S7). We also studied a network scheme where each neuron received a random subset of the pre-synaptic inputs, that is, we did not enforce a sequential activation of the neurons consistent with the sequential order of the pre-synaptic firing. Furthermore, the network had a random, all-to-all connectivity scheme (Fig. S8). Similarly to the results of Fig. 3, the network exhibited a reorganization of the synaptic weight which led to the recall of the full sequence with a compressed timescale. These results were dependent on the total number of input subsets to each neuron in the network (Fig. S8e).

Together, these results show that a recurrently connected network of neurons each endowed with a predictive learning rule can spontaneously organize to fire preferentially at the beginning of a sequence, and recall (or replay) sequences at a compressed timescale.

**Emergence of spike-timing-dependent plasticity rules**
The results shown above clearly demonstrate that the potentiation of synaptic weights depends on the timing relationships between inputs. This suggests that there may be a connection between the predictive learning rule described here and the experimentally observed spike-time-dependent-plasticity (STDP) rules[19].

To systematically investigate the dependence of potentiation and depotentiation on the timing relationships between pre-synaptic inputs, we considered the simplified case of two inputs (as in Fig. 1). In Fig. 1, we had shown that the predictive learning scheme leads to asymmetric synaptic weights when two pre-synaptic inputs have different arrival times. To quantify how the asymmetry between the synaptic weights of the first and second input evolved with learning, we defined an asymmetry index (Fig. 4a). The asymmetry index was defined as $d_j - d_0$, where $d_j$ was the difference in weights at the $j$-th epoch $d_j = w_{1,j} - w_{2,j}$, and $d_0$ the initial difference in weights. Thus, positive values of the asymmetry index indicated that the synaptic weight for the first input became relatively large compared to the synaptic weight for the second input. To illustrate the behavior of the asymmetry index, we trained the model by repeating a sequence of two input spikes coming from two different pre-synaptic neurons having a relative delay of 4 ms for several epochs of duration $T = 500$ ms (as in Fig. 1). The parameter space in this case was defined by the initial weights ($w_{1,0}, w_{2,0}$). After 100 epochs, a small region of the parameter space still showed asymmetry indices around zero, while for most of the parameter space, there were positive asymmetry indices, indicating a convergence towards anticipatory activity. After 300 epochs, every initial condition led to the asymmetric solution and thus to a positive value of the asymmetry index. Thus, the observation that the first input was potentiated was generally observed for different initial states of the synaptic weights (Fig. 4a).

To directly investigate the relation between STDP and the predictive learning rule described here, we investigated the dependence of plasticity on the relative timing between inputs. To this end, we performed a simulation that resembled the standard STDP protocol (see Methods). We trained the predictive plasticity model with a sequence of two input spikes from two different pre-synaptic inputs $x_1$ and $x_2$ arriving at a relative delay $\Delta t$. To approximate the STDP protocol with a current injection that triggers a post-synaptic spike, the initial conditions were chosen such that $x_2$ triggered a post-synaptic spike, and $x_1$ was a sub-threshold input. A negative and positive delay $\Delta t$ indicated that $x_1$ arrived before or after $x_2$, respectively. We found

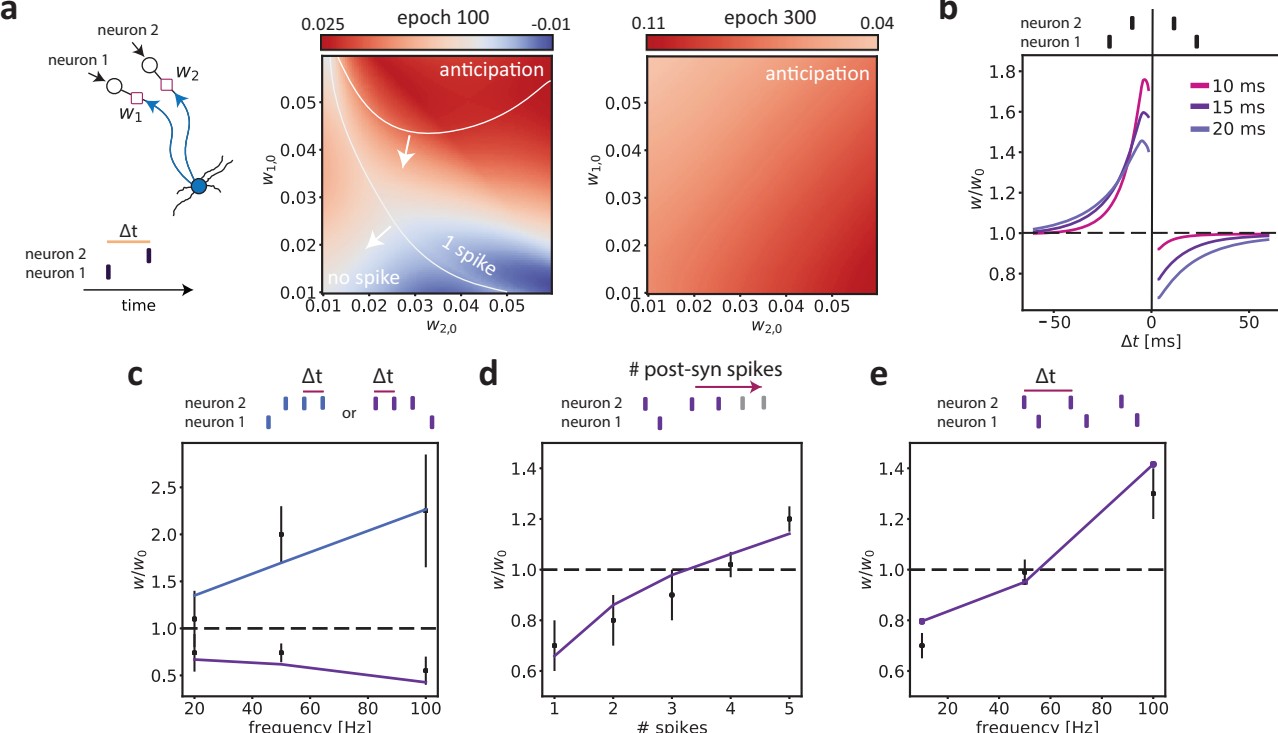

**Fig. 4 | Predictive learning rule gives rise to spike-timing-dependent plasticity mechanisms. a** Left: Illustration of the protocol, where a neuron receives inputs from two pre-synaptic neurons (associated with weights $w_1$ and $w_2$) with a delay of $\Delta t = 4$ ms. These inputs were repeated across epochs. Middle and right: Asymmetry index, computed as the difference between the initial weight vector ($w_{1,0}, w_{2,0}$) and the final vector after $j$ epochs: ($w_{1,j} - w_{1,0}$) − ($w_{2,j} - w_{2,0}$). Positive values of the asymmetry index thus indicate that $w_1$ increases relative to $w_2$. Shown are the asymmetry index after 100 and 300 epochs, as a function of the initial weights. The white lines divide three regions: (1) No spike; (2) A single spike; (3) A single spike before the second input (i.e. anticipation). The right panel shows, that for all initial weight conditions, the weight of the first input showed a relative increase as compared to the second input. **b** In order to model classic STDP protocols with current injection, one of the two inputs (pre-synaptic neuron 2) had a strong initial weight (i.e. did evoke a spike), and the other input (pre-synaptic neuron 1) was sub-threshold (i.e. did not evoke a spike). In this simulation, the weights for both inputs could be adjusted via the predictive learning rule (see Fig. S11 when the second input has a fixed weight). The y-axis shows the weight change (in percentage relative to the initial weight) of the sub-threshold input (i.e. pre-synaptic neuron 1) as a function of the delay $\Delta t$ between the two input spikes (see Methods). Negative and positive values of $\Delta t$ indicate that input 1 preceded or lagged input 2, respectively. Shown are the weight changes for different membrane time constants after 60 epochs. **c** In this simulation, the second input contained a burst of 3 spikes, which arrived after the first input, and each triggered a spike in the post-synaptic neuron. The input from pre-synaptic neuron 1 only had a sub-threshold effect. Shown is the weights change (as in **b**) versus the firing frequency, i.e. $1/\Delta t$, within the burst (total of 3 spikes per burst). The blue and purple lines refer to the case that input 1 preceded input 2 or lagged input 2, respectively. Data used with permission of Society for Neuroscience, from "Spine Ca2+ Signaling in Spike-Timing-Dependent Plasticity", Nevian and Sakmann, Journal of Neuroscience 26(43), 2006; permission conveyed through Copyright Clearance Center, Inc. (we refer to[42] for information about the sample size), RMS error: 0.868 for pre-post pairing and 0.206 for post-pre pairing. **d** Weights change (as in **b**) as a function of the number of spikes in the second input. The inputs from pre-synaptic neuron 2 each triggered a spike in the post-synaptic neuron. The input from pre-synaptic neuron 1 only had a sub-threshold effect. RMS error: 0.089. **e** Weights change (as in **b**) induced by increasing the frequency pairing. Here, the inputs from pre-synaptic neuron 2 always triggered a spike in the post-synaptic neuron, whereas the input from pre-synaptic neuron 1 only had a sub-threshold effect. The inputs from neuron 2 arrived 6 ms before the inputs from neuron 1. RMS error: 0.057. Data in panel **d** and **e** used with permission of The American Physiological Society, from `Contribution of Individual Spikes in Burst-Induced Long-Term Synaptic Modification", Froemke et al., Journal of neurophysiology 95(3), 2006; permission conveyed through Copyright Clearance Center, Inc. (we refer to[43] for information about the sample size).

that the potentiation of the first input was determined by the relative delay $\Delta t$ (Fig. 4b). Specifically, we observed an antisymmetric learning window with a similar time dependence as has been experimentally observed for STDP[18,19] (Fig. 4b). This window expanded as a function of the membrane time constant $\tau_m$ (Fig. 4b). The model exhibited such an antisymmetric learning window even though we did not explicitly implement any spike-timing-dependent LTP and LTD rule with a specific learning kernel.

In addition to the classical dependence on the relative delay between inputs, many other pre- and post-synaptic factors can influence the sign and amplitude of synaptic plasticity[39–41]. The nonlinear and history-dependent interactions associated with STDP are especially relevant when neurons receive complex input spiking patterns. We considered some of those complex STDP protocols to test if we can reproduce the nonlinear effects by means of the predictive learning rule:

(1) Experimental evidence indicates that the frequency of post-synaptic bursts after a pre-synaptic input can boost LTP while LTD remains unchanged[42]. To emulate this, we simulated a case where the inputs from pre-synaptic neuron 2 arrived in a burst, with each spike in the burst triggering a post-synaptic spike. We quantified the total weight change after training the model on this protocol. The effect of the intra-burst frequency on the synaptic weight change, as experimentally observed[42], was reproduced by our model (Fig. 4c). (2) Froemke et al. showed that adding more post-synaptic spikes after a post-pre pairing can convert LTD into LTP (see Fig. 6 in ref. 43). We tested the model on such multi-spike protocol and observed that our model can reproduce the transition measured in the experimental data (Fig. 4d). (3) The frequency of pre-and post-synaptic spike pairings can influence plasticity and convert LTD to LTP for high-frequency bursts[39,43,44]. The predictive plasticity model reproduced the dependence on the frequency pairing as observed in ref. 43 (Fig. 4e).

Interestingly, Fig. 4a shows that in early training (i.e. repetition) epochs, certain regions of the parameter space can lead to an asymmetric index close to zero. This suggests that the STDP window might have different forms depending on the parameter space and the training epoch, even though the neuron eventually converges onto anticipatory firing. Consistent with this observation, experimental studies have also observed symmetric STDP windows that are either LTP-dominated[45] or LTD-dominated[46]. In Fig. S9 we indeed show that the predictive learning rule can, for certain parameter settings, yield a symmetric STDP window that is either LTP- or LTD-dominated. Symmetric STDP windows can emerge even though the neuron eventually does converge onto an anticipatory solution. A key factor that determines the specific shape of the STDP window may be the initial strength of the synapse. In agreement, experimental work has shown that the amount of plasticity in a standard STDP protocol depends on the initial strength of the pre-synaptic weight[44]. To investigate this, we examined the potentiation of the first input depending on its initial synaptic weight. We found that there was a switch from potentiation to depotentiation as the initial synaptic strength increased (Fig. S10), consistent with the experimental observations[44].

To further relate our findings to experimental observations in which a current injection protocol was used, we performed simulations in which we fixed the synaptic weight of the supra-threshold input (i.e., in this case, it was not adjusted by plasticity). Also in this case, the model displayed an antisymmetric STDP kernel (Fig. S11).

Altogether, we showed that the predictive learning model can reproduce several linear and nonlinear STDP features.

## Discussion

The anticipation of future events is a core feature of intelligence and critical for the survival of the organism. Here, we studied how individual neurons can learn to predict and fire ahead of sensory inputs. We propose a plasticity rule based on predictive processing, where an individual neuron learns a low-rank model of the synaptic input dynamics in its membrane potential. Accordingly, the sign and magnitude of synaptic plasticity are determined by the timing of the presynaptic inputs. That is, synapses are potentiated to the degree that they are predictive of future input states, which provides a solution to an optimization problem that can be implemented using only information available at the single-neuron level. We show that neurons endowed with such plasticity rule can learn sequences over long timescales and shift their spikes towards the first inputs in a sequence (i.e. anticipation). Furthermore, neurons represent their inputs in a more efficient manner (i.e. with reduced overall membrane potential). This anticipatory mechanism was able to explain the development of anticipatory signaling and recall in response to sequence stimuli. Finally, we demonstrated that the learning rule described here gives rise to several experimentally observed STDP mechanisms, including: asymmetric STDP kernels[18,19], as well as symmetric ones[45,46] given the initial conditions; the frequency-dependence of STDP[43]; the number of post-synaptic spikes in a burst or post-pre pairing[39]; the dependence of de-potentiation on the initial synaptic strength[44]. Together, our results indicate that prediction may be a guiding principle that orchestrates learning and synaptic plasticity in single neurons, providing a different interpretation of STDP phenomena.

We first discuss how our results relate to previous theories of coding in cortical networks that emphasize the importance of predictions. An influential theory of cortical function is hierarchical predictive coding (HPC). The basic understanding of HPC is that the brain maintains a model or representation of current and future states in the outside world, and updates this model as new information comes in. HPC posits that the inference process is implemented by the feedforward routing of surprising or unpredicted signals (i.e. prediction errors), and the routing of sensory predictions down the hierarchy via feedback (FB) projections[10,11,47]. The

predictive plasticity mechanism that we described here differs from HPC models in several aspects, for example: (1) In HPC, prediction is the result of network interactions, in particular the cancellation of the feedforward drive by inhibitory feedback. In our model, prediction results from plasticity at a single neuron level. (2) Different from HPC, in our model the neuron does not explicitly transmit (encode) prediction and error signals. (3) Both in HPC and our model, neurons may exhibit reduced firing for predicted as compared to unpredicted sensory inputs. Yet, in our model, this is due to depotentiation of predictable inputs, whereas in HPC it is due to inhibitory feedback mediated by top-down projections. We note that our plasticity model is fully compatible with another flavor of predictive processing, namely "coding for prediction". According to this theory, neurons primarily transmit information about sensory inputs that carry predictive information about the future, as observed in the retinal neural circuits[48]. The findings here may also be relevant to understand the development of anticipatory firing in sensory systems[4,7,8], temporal difference learning[49,50], as well as the compression of sequences during resting state based on prior experience[34,51]. Finally, recent work showed that neural activity in the auditory cortex can be predicted roughly 10–20 ms in advance and that these predictions can be exploited at the single neuron level to achieve high performance in classification tasks[52]. However, prediction in the model of[52] does not happen in an unsupervised manner in time as their method relies on the combination of the single neuron prediction with a supervised teaching signal, a novel implementation of Contrastive Hebbian Learning[53].

Next, we discuss how our findings relate to STDP experiments and models. STDP is an established experimental phenomenon that has been widely observed in-vitro[18,19]. There is evidence for a variety of STDP kernels[40], which dependent on several post-synaptic variables like backpropagating action potentials (bAP)[54], post-synaptic bursts[42] and the dendritic location of inputs[30]. These experimental findings are all obtained in in-vitro preparations. Thus, it is unclear what the nature of STDP in-vivo is. The standard protocol for testing STDP has two major limitations that deviate from the normal physiological setting: (1) The protocol involves a current injection in the post-synaptic neuron. The current injection itself is not subject to (physiological) plasticity and might therefore not be a good "proxy" for post-synaptic depolarization induced by natural pre-synaptic inputs in-vivo. (2) Several studies have pointed out that different post-synaptic signals (e.g. spike times, depolarization level, dendritic spikes) are relevant for STDP[44,54,55]. It is still a manner of debate what is the crucial post-synaptic variable for plasticity[56]. In principle, STDP models might apply both to cases with artificial currents and physiological pre-synaptic inputs. An artificial depolarization caused by current injections can lead to plasticity in both our model and STDP models. Yet it is an open experimental question what is the nature of learning rules when it comes to physiological synaptic inputs and their timing relationships. For example, it is known that to induce LTP (Long Term Potentiation), it is not necessary to evoke a post-synaptic spike[55,57]. The learning rule proposed here predicts that, in-vivo, presynaptic inputs causing a post-synaptic spike will eventually become depotentiated if they are anticipated (i.e. predicted) by other presynaptic inputs in a sequence.

Comparing the present results to previous work, we emphasize that we did not construct a learning rule to reproduce experimentally observed STDP phenomena. Indeed, several phenomenological (i.e. descriptive) STDP models have previously been proposed to fit experimental data[58–61] providing mathematical tools to describe, model, and predict the behavior of neurons. However, these phenomenological models might not fully explain the computational significance of STDP mechanisms, nor the algorithms from which these biological implementations can emerge. Our approach differs from these models in that we took an optimization problem based on the

prediction of future inputs as a starting point. From this optimization problem, we derived a learning rule which gave rise to experimentally observed STDP mechanisms. Our results, together with previous studies[62–64], suggest that STDP is a consequence of a general learning rule given the particular state of the system, the stimulation protocol, and the specific properties of the input. As a consequence, several STDP learning windows which are described by other phenomenological rules are predicted by our model, as well as the dependence on synaptic strength and depolarization level.

An example of an established phenomenological model of STDP is the one developed by Clopath et al.[65]. The authors, guided by experimental evidence[30,44,66], modeled the role of the membrane voltage as the relevant post-synaptic variable for synaptic plasticity. The plasticity model described in[65] can accurately reproduce a wide range of experimental findings which, to our knowledge, is not possible with STDP learning rules that are only based on spike timing[61]. The Clopath rule is based on the two-threshold dynamics observed by Artola et al.[66] and the authors assume an Adaptive-Exponential I&F (AeI&F) model for the voltage dynamics[67], together with additional variables for the spike after-potential and an adaptive threshold. In agreement with the results of ref.[65], we were able to account for a wide range of phenomena with a simpler model of voltage dynamics, supporting the idea that the history-dependent effect of the membrane potential is pivotal to plasticity. Our model also predicts the experimental observation that the amount of LTP has an inverse dependence on the initial strength of the synaptic input[44]. To our knowledge, this finding is not described or predicted by the model of ref.[65], because it does not include a dependence on the initial strength of the synapse. Another unique feature of the learning rule described here is that it can produce different several STDP kernels (e.g. asymmetrical, symmetrical) depending on the initial conditions.

The learning rule in Equation (3) depends on synaptic mechanisms that are biologically plausible, as they rely on information that is locally available at the level of single neurons. In this model, synaptic plasticity depends on the interaction between synaptic variables and global signals, that in turn depend on the pre-and post-synaptic activity (the prediction errors $\epsilon_t$) and the strength of the synapses $\mathbf{w}_{t-1}$. These processes can be implemented with local mechanisms such as NMDA receptors, voltage-gated calcium channels (VGCCs)[68,69], and synaptic interactions via intracellular signals or membrane depolarization[16,28,55,56,70,71]. In particular, the second term in Equation (3) defines the interaction of a local trace of pre-synaptic inputs with a global, post-synaptic term $\mathcal{E}_t$. This global term entails that transient, unpredicted increases in the voltage contribute to LTP, whereas more sustained depolarization contributes to LTD. In fact, experimental evidence shows that different molecular pathways depending on global, post-synaptic variables - e.g. membrane depolarization[66,72], intracellular [Ca2+] transients[73] - are key in determining the sign and amplitude of synaptic plasticity.

The learning rule described here and phenomenological STDP models might also lead to similar behavior in terms of spiking output in response to sequences. In agreement with the present work, previous studies have shown that phase precession can lead to the learning of temporal sequences through an asymmetric learning window, as in spike-timing-dependent plasticity (STDP)[5,74,75]. Modeling studies have shown that a post-synaptic neuron endowed with an LTD-dominated STDP model can exhibit potentiation of the first synaptic inputs in a temporal sequence, leading to a decrease in the latency of the post-synaptic response[76,77]. A key difference with our work, however, is that the predictive plasticity rule described here does not produce asymmetric STDP under all conditions. In fact, the degree of potentiation and depotentiation in our model depends on the initial state. That is, there is no fixed STDP kernel in our model. Another difference is

that our model can anticipate sequences independent of the initial conditions of synaptic weights (in contrast to ref.[77]), for a wide range of sequence lengths, and pre-synaptic population size. In the model described here, we show that the anticipation of sequences is a convergence point during learning and thereby it is a general solution for a wide set of model parameters.

Thus, we propose that a single neuron perspective on prediction and anticipatory mechanisms is important as the implementation of any plasticity rule is ultimately achieved at the neuronal level, thereby guiding behavior at the system level. Yet, it is obvious that single neurons are embedded into networks and different means of communication can lead to more complex learning rules, in which the single-neuron learning rule described here might be one component. Indeed, it is possible that certain empirical phenomena like sequence recall additionally depend on network dynamics instead of single-neuron learning rules. For example, the faster recall of sequences in the visual system observed in[7] was reproduced in a recent work[78]. The authors developed a biologically realistic network model which differs from our implementation in several ways: (a) a network of both excitatory and inhibitory neurons, (b) a random Gaussian connection probability, (c) a leaky I&F model with conductance-based AMPA, GABA and NMDA synaptic currents, (d) several network hyperparameters such as synaptic delays, (e) a short-term depression model and a specific multiplicative, NN-STDP model. While the model in ref.[78] gives a biological explanation based on the conductive properties of the ion channels, the network implementation of our plasticity rule provides a principle approach to understanding fast sequence recall as it is observed also in other brain areas, e.g. different primary sensory areas[4,6] or the hippocampus[51]. We qualitatively reproduced the faster recall of sequences with a much simpler model, supporting the pivotal role of spike times and excitability for the phenomenon.

Our work is further related to recent approximation algorithms for learning in neural networks such as e-prop[79] or surrogate-gradients techniques[80] as our model provides an online approximation for training spiking neural networks (SNN). In ref.[79] the authors showed that learning in spiking recurrent neural networks can be decomposed into two terms, a global loss, and an eligibility trace which depends on the local state of neurons and results in synaptic weight changes according to local Hebbian plasticity rules. In our model, the optimization problem is defined directly at the level of single neurons. Thus, all learning is local in space, i.e. there is no global loss at the network level. An interesting question for future research is combining these two approaches by obtaining a completely local learning rule for optimization at the single and network level simultaneously. Indeed, experimental evidence[41,81] shows that local learning rules implemented by neurons provide a substrate on which global feedback signals act[82], which may provide a biological mechanism for error backpropagation[83]. As our loss function depends only on the membrane potential of the cell, our model avoids the problem of propagating the gradient through discrete spikes. It follows that we did not implement a surrogate-gradient approximation[80].

Looking forward, further investigation on single neurons anticipating local inputs and their interplay through network interactions is key to understanding how complex prediction strategies can emerge. Moreover, synapses can be located far from the soma along the dendritic arbor and might not be able to access the somatic membrane potential directly, with strong consequences on plasticity[30]. Other post-synaptic events such as NMDA spikes or plateau potentials can have an effect on plasticity rules based on membrane voltage, see e.g. in ref.[84]. Thereby, spatially segregated dendrites and spatiotemporal integration of events along the neuronal compartments could drastically increase the complexity of prediction obtainable at the single neuron level.

## Methods

### Neuron model

We used a leaky Integrate-and-Fire-like (LIF) model of the form

$$\tau_m \frac{dv(t)}{dt} = -v(t) + \mathbf{w}^\top \mathbf{x}(t) - v_{(\text{th})} \sum_j \delta(t - t_j).\qquad(6)$$

Here, $v \in \mathbb{R}$ is the membrane potential, $\tau_m$ is the membrane time constant, $\mathbf{x}(t) \in \mathbb{R}^N$ is the pre-synaptic input, $\mathbf{w} \in \mathbb{R}^N$ is the weight vector and $v_{(\text{th})}$ is the spiking threshold (the subscript $_{(\text{th})}$: "threshold"). The sum in the last term runs over all the post-synaptic spike times $t_j$ and $\delta(\cdot)$ is the Dirac delta function. Without loss of generality, we set the resting state of the membrane potential to zero. We used a discrete time-step $h$ to discretize equation (6), yielding a recurrence model of the form

$$\begin{cases} v_t = \alpha\, v_{t-1} - v_{(\text{th})} s_{t-1} + \mathbf{w}^\top \mathbf{x}_t \\ s_t = H(v_t - v_{(\text{th})}). \end{cases}\qquad(7)$$

Here, $\alpha \equiv 1 - h/\tau_m$ and $H(\cdot)$ is the Heaviside function. The variable $s_t \in \{0, 1\}$ takes binary values and indicates the presence or absence of an output spike at timestep $t$. If the voltage exceeds the threshold, an output spike is emitted, and this event reduces the membrane potential by a constant value $v_{(\text{th})}$ at the next time step. This implementation of the membrane potential reset relates our model to the spike response model[85]. We set $h = 0.05$ ms in all numerical simulations.

### Derivation of the learning rule

The predictive plasticity model entails that neurons predict future inputs by extracting information from the current state of the membrane potential. We formalized this as an optimization problem in time, and we defined the objective function $\mathcal{L}$ as the cumulative error in a given time window $T$

$$\mathcal{L} \equiv \sum_{t=0}^{T} \mathcal{L}_t \equiv \sum_{t=0}^{T} \frac{1}{2} \|\mathbf{x}_t - v_{t-1}\mathbf{w}\|_2^2,\qquad(8)$$

where $\|\cdot\|_2$ is the $l_2$-norm. The objective is to obtain the minimal difference between the input $\mathbf{x}_t$ and its prediction via $v_{t-1}$ and $\mathbf{w}$. We assume that the mismatch is evaluated at each timestep $t$. The gradient of $\mathcal{L}$ w.r.t. to $\mathbf{w}$ can be computed in a recursive manner by unrolling the computation via backpropagation-through-time (BPTT)[86]. At each timestep, $t$, the exact gradient of $\mathcal{L}$ can be written as the contribution of two terms,

$$\nabla_{\mathbf{w}} \mathcal{L} = \sum_{t=0}^{T} \frac{1}{2} \left( \nabla_{\mathbf{w}} \mathcal{L}_t + \frac{\partial \mathcal{L}_t}{\partial v_{t-1}} \nabla_{\mathbf{w}} v_{t-1} \right).\qquad(9)$$

The first term accounts for the direct effect of a weight change on $\mathcal{L}_t$, while the second accounts for its indirect effect via the membrane potential $v_{t-1}$.

The first term of the gradient is given by,

$$\nabla_{\mathbf{w}} \mathcal{L}_t = -2(\mathbf{x}_t - v_{t-1}\mathbf{w}_t) v_{t-1},\qquad(10)$$

which propagates the prediction error selectively to each input and scales it by the feedback signal determined by $v_{t-1}$. This term can be interpreted as a time-shifted version of Oja's rule[25], where $v_{t-1}$ plays the role of the linear output variable. In fact, Oja's rule can be derived as an approximated gradient descent method for dimensionality reduction problems[87]. The second term of equation (9) has a contribution given by the direct effect of $v_{t-1}$ on the prediction,

$$\frac{\partial \mathcal{L}_t}{\partial v_{t-1}} = -2(\mathbf{x}_t - v_{t-1}\mathbf{w})^\top \mathbf{w},\qquad(11)$$

i.e., the sum of the prediction errors weighted by their relative synaptic strengths. To obtain real-time learning, we want to comprise the remaining term of equation (9) via dynamical updates. We obtain this by differentiating the dynamical rule of the membrane potential in (1)

$$\nabla_{\mathbf{w}} v_t = \left( \alpha - v_{(\text{th})} \frac{\partial s_{t-1}}{\partial v_{t-1}} \right) \nabla_{\mathbf{w}} v_{t-1} + \mathbf{x}_t.\qquad(12)$$

The first term contains the Jacobian $J_t$ of equation (1)

$$J_t = \frac{\partial v_t}{\partial v_{t-1}} = \alpha - v_{(\text{th})} \frac{\partial s_{t-1}}{\partial v_{t-1}},\qquad(13)$$

which holds the contribution of the linear recurrent term and of the threshold nonlinearity for the spiking output $s_{t-1}$. Similarly to the adjoint method[88], we define an influence vector $\mathbf{p}_t \equiv \nabla_{\mathbf{w}} v_t$ such that it obeys the recursive equation

$$\mathbf{p}_t = J_t \mathbf{p}_{t-1} + \mathbf{x}_t,\qquad(14)$$

which follows straightforward from equation (12) and gives a forward-pass dynamical update of the gradient[89]. Finally, we define the prediction error $\boldsymbol{\epsilon}_t$ at timestep $t$ as

$$\boldsymbol{\epsilon}_t \equiv \mathbf{x}_t - v_{t-1}\mathbf{w},\qquad(15)$$

that defines the sign and amplitude of plasticity, and the global signal

$$\mathcal{E}_t \equiv \boldsymbol{\epsilon}_t^\top \mathbf{w}.\qquad(16)$$

All together, the exact gradient of Equation (9) can be written as

$$\nabla_{\mathbf{w}} \mathcal{L} = -\sum_{t=0}^{T} (\boldsymbol{\epsilon}_t v_{t-1} + \mathcal{E}_t \mathbf{p}_{t-1}).\qquad(17)$$

After the end of the period $[0, T]$ the exact gradient can be used to update the weight vector via gradient descent

$$\begin{aligned} \mathbf{w}_k &= \mathbf{w}_{k-1} - \eta \nabla_{\mathbf{w}} \mathcal{L} \\ &= \mathbf{w}_{k-1} + \eta \sum_{t=0}^{T} (\boldsymbol{\epsilon}_t v_{t-1} + \mathcal{E}_t \mathbf{p}_{t-1}). \end{aligned}\qquad(18)$$

Here, $\eta$ is the learning rate parameter and the index $k$ represents the $k$-th iteration during the training period. We are interested in an online learning rule where the weight update forms part of the dynamics of the model, and takes place in real-time with the prediction of the pre-synaptic inputs. This is a typical method in stochastic optimization theory for recursive objective functions[90] and online signal processing[87]. We approximate the learning equation (18) with the current estimate of the gradient,

$$\begin{aligned} \mathbf{w}_t &= \mathbf{w}_{t-1} - \eta \nabla_{\mathbf{w}} \mathcal{L} \simeq \mathbf{w}_{t-1} - \eta \nabla_{\mathbf{w}} \mathcal{L}_t |_{\mathbf{w} = \mathbf{w}_{t-1}} \\ &= \mathbf{w}_{t-1} + \eta (\boldsymbol{\epsilon}_t v_{t-1} + \mathcal{E}_t \mathbf{p}_{t-1}). \end{aligned}\qquad(19)$$

Our approximated learning rule is completely online as it only requires information available at time step $t$.

Theoretical studies suggest that such stochastic approximation works when $\eta$ is sufficiently small[87,89]. In our case, the passive memory capacity of the membrane potential is given by its time constant $\tau_m$. In the limit of $\eta\tau_m \ll 1$, the changes in the weights are slow compared to

voltage changes and the following relation holds:

$$\mathbf{w}_{t'} \simeq \mathbf{w}_t \quad \forall \, t' \, : \, |t - t'| < \tau_m . \tag{20}$$

Thus the weight change is negligible, and the learning rule is approximately exact in the time window defined by the membrane time constant $\tau_m$. The timescale separation discussed above should apply to biological neurons as the membrane time constant is in the order of $1 - 10$ ms while synaptic plasticity happens at a timescale in the order of $10^2 - 10^3$ ms, requiring several repetitions of the same stimulation protocol.

## Jacobian and surrogate-gradient method

The first term in equation (13) allows the gradient to flow at every time step $t$ via the dynamics of the membrane potential $v_t$. The second term has a discontinuous effect in time (at the moment of the output spikes) and depends on the specific nonlinear function. This latter term can be approximated following the surrogate-gradient method[80]

$$-v_{(th)} \frac{\partial s_{t-1}}{\partial v_{t-1}} \simeq \gamma f(v_{t-1}, v_{(th)}) , \tag{21}$$

where $f$ is a continuous function of $v_t$ and $\gamma$ is a scaling factor. In general, the backpropagation of the gradient through the reset mechanisms is neglected[79,80,91]. Here we defined the membrane potential $v_t$ as the output variable of the system and the loss function and $s_t$ as a hidden variable for the objective function. Therefore, our implementation directly avoids the problem of backpropagating through discrete output variables. Given these two arguments, we considered $\gamma = 0$ throughout the paper. In the network implementations of Fig. 3, Fig. S7 and Fig. S8, each neuron in the network receives inputs from other neurons, and the associated recurrent connections are defined by the specific connectivity scheme. The contributions to the gradient of the recurrent connections from a neuron $j$ to a neuron $i$ in the network are proportional to

$$\propto \frac{\partial s_{t,j}}{\partial s_{t,i}} \frac{\partial s_{t,i}}{\partial v_{t,i}} , \tag{22}$$

and have a discontinuous effect in time (at the moment of the output spikes). Thus, we neglected these contributions to the gradient.

## Optimization and initialization scheme

Equation (19) defines a completely online optimization scheme that can be implemented locally by single neurons. In Fig. 3, Fig. S5, Fig. S6, Fig. S7, Fig. S8, and Fig. S10 we updated the synaptic weights following the online approximation of the gradient in Equation (19). For the results of Fig. 1, Fig. S1, Fig. 2b, c, Fig. 4, Fig. S3, Fig. S4, Fig. S9 and Fig. S11 we added a scaling term to the predictive plasticity rule as,

$$\mathbf{w}_t = \mathbf{w}_{t-1} + \eta \, \mathbf{w}_{t-1} \odot \left( \boldsymbol{\epsilon}_t \, v_{t-1} + \mathcal{E}_t \, \mathbf{p}_{t-1} \right) . \tag{23}$$

That is, we multiplied the learning rate with $\mathbf{w}_{t-1}$ to ensure non-negative values of the weights. Consequently, the weight update at time step $t$ was proportional to the synaptic weight value at time step $t - 1$. In Fig. 2d and Fig. S2 we did not use the online approximation of the gradient, and we optimized the full model (Equation (18)) using the Adam optimizer[92].

For the example in Fig. 1, for Fig. 2b, c, Fig. 3 and for the spike-timing-dependent plasticity protocols in Fig. 4, the initial weights were assigned to fixed values for the different cases considered. In Fig. 2d, the initial weights were randomly drawn from a truncated normal distribution. We bonded the truncated normal distribution to obtain positive values of the initial weights. The variance of the normal distribution was scaled by the squared root of the total input size $N$.

## Simulations

The spiking sequences were defined by a set of ordered spike times at which the pre-synaptic neurons were active. For all simulations, the inputs were convolved with an exponential kernel with $\tau_x = 2$ ms to replicate the fast dynamics of post-synaptic currents. We added two sources of noise to the simulations in Fig. 2-3 and corresponding Supplementary Figs. 1) In each epoch, the spikes that were part of the sequence were randomly shifted by an amount $\Delta t$ uniformly distributed between [-2, 2] (in ms). 2) Each pre-synaptic also exhibited stochastic background firing following a homogeneous Poisson process with constant rate $\lambda$ uniformly distributed between 0 and 10 Hz. In addition, in Fig. 2, the onset of the sequence relative to the time window was randomly chosen between 0 and 200 ms, and there were an additional 100 neurons that fired randomly. We trained the model by numerically solving the dynamics during each epoch. The model was fully determined by 5 hyperparameters: the timestep $h$, the membrane time constant $\tau_m$, the spiking threshold $v_{th}$, the learning rate $\eta$ and the input time constant $\tau_x$. To quantify the performance (in terms of sequence anticipation) in Fig. S3, we fixed the number of training epochs, we performed 100 numerical simulations for each condition, and we labeled successful simulations based on two criteria: 1) The synaptic weight associated with the first spike in the sequence is bigger than all the other synaptic weights (input selectivity). 2) The output latency is smaller than 20 ms after the onset of the input sequence (fast anticipation). We computed the error as 1 minus the percentage of successful trials in the set of 100 simulations.

## Spike-timing-dependent plasticity protocols

For the STDP simulations, we used a 2-dimensional input $\mathbf{x} = (x_1, x_2)$ and we simulated classical pre-before-post and post-before-pre pairing, as typically performed in STDP experiments[18,19]. To approximate the STDP protocol with a current injection that triggers a post-synaptic spike, the initial conditions were chosen such that $x_2$ triggered a post-synaptic spike, and $x_1$ was a sub-threshold input. For the results in Fig. 4b–e, we changed the number and the timing of the pre-synaptic neuron spikes according to each experimental protocol. We numerically solved the dynamics of the model by repeating the input pattern a number of times as in the experimental protocols. We computed the weight change of the sub-threshold input as the ratio of the synaptic weight before and after each simulation protocol.

We reproduced the same burst-dependent pairing protocol as in ref. 42 by decreasing the delay $\Delta t$ between post-synaptic spikes (Fig. 4c). The bursts were triggered by three input spikes. We simulated the 1-$n$ protocol as in ref. 43 by pairing post-pre-post inputs with a 100 Hz burst frequency. The number of input spikes from the supra-threshold input was increased systematically as in the experimental protocol (Fig. 4d). We reproduced the frequency pairing protocol as in ref. 43 by pairing 5 post-pre inputs with different intra-pairing frequencies as in the experimental protocol (Fig. 4e).

## Reporting summary

Further information on research design is available in the Nature Portfolio Reporting Summary linked to this article.

## Data availability

The data that support the findings are generated by a custom code provided in the following code repository. The experimental data in Figure 4c-d-e were used with permission of Society for Neuroscience, from "Spine Ca2+ Signaling in Spike-Timing-Dependent Plasticity", Nevian and Sakmann, Journal of Neuroscience 26(43), 2006[42], and with permission of The American Physiological Society, from "Contribution of Individual Spikes in Burst-Induced Long-Term Synaptic Modification", Froemke et al., Journal of neurophysiology 95(3), 2006[43]; permission conveyed through Copyright Clearance Center, Inc.

## Code availability

The code to reproduce the figures in the main text and in the Supplementary Information is freely available at: https://github.com/matteosaponati/predictive_neuron. The scripts contained in the repository require Python 3.8, NumPy[93], SciPy[94], Matplotlib[95], PyTorch[96].

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

## Acknowledgements

This work was supported by an ERC Starting Grant (SPATEMP) and a BMF Grant (Bundesministerium fuer Bildung und Forschung, Computational Life Sciences, project BINDA, 031L0167). We thank Wolf Singer and Andreas Bahmer for helpful comments, edits, and insightful discussions.

## Author contributions

Conceptualization: M.S., M.V. Mathematical analysis: M.S., M.V. Simulations: M.S. Writing: M.S., M.V. Supervision: M.V.

## Funding

## Competing interests

The authors declare no competing interests.
