## [Peer Review File · Nature Communications]

REVIEWER COMMENTS

Reviewer #1 (Remarks to the Author):

The paper is focused on a specific “predictive” learning rule for spiking neural network (SNN) whose goal is to, as authors put it, “give a credit to those inputs that are informative about future events and discard those inputs that are expected and redundant”. Specifically, authors consider classical discrete-time SNNs with leaky-integrate-and-fire neurons. The predictive learning rule seems to be inspired by the well-known Oja’s rule. The main idea of the proposed rule is to adjust neuron’s weights w at timestep t in such a way as to minimize the distance (l_2 norm) between the previous timestep vector of weights $w(t-1)$ scaled by the previous timestep neuron’s membrane potentials $v(t-1)$ and the current timestep vector of the neuron’s pre-synaptic inputs $x(t)$, i.e. minimize the Euclidean distance between $x(t)$ and $w(t-1)*v(t-1)$. The proposed rule is actually a bit more involved as minimization at timestep t seems to be performed over a range of T prior timesteps, i.e. the goal is to minimize the sum of $||x(t') - w(t'-1)*v(t'-1)||$ over t' , where t' spans from $t-T$ to t .

Another major result is a derivation of a local (online) version of the proposed prediction rule, which depends only on local information available at timesteps t and $t-1$, i.e. $x(t)$, $v(t-1)$, and $s(t-1)$, where the latter is a neuron’s output spike, generated at timestep $t-1$ when membrane potential $v(t-1)$ exceeds a specific threshold.

The proposed rule is studied in several scenarios – by considering a neuron with two inputs and applying a repetitive pattern of two spikes (Fig. 1), more complex scenarios of learning with spatio-temporal patterns with background noise (Fig. 2) and for recurrently connected SNNs (Fig. 3). In the final study (Fig. 4), the authors compare the predictive rule with common STDP rules.

The proposed learning rule and its local version are intriguing. However, I have difficulty understanding some results and have related several concerns.

1) I assume that figure 2b shows the results of 100 simulations. Why are average output spike times evolving to zero upon training? Given that “the sequence onset times are homogeneously distributed between 0 and 200 ms”, as shown in Fig. 2a, this behavior is surprising. It seems to me that the desired outcome after learning with the proposed rule would be to spike whenever the first spike of the pattern is presented so that output spike times should be uniformly distributed between 0 and 200 ms? The results make sense if a neuron reacts to background noise spikes unrelated to a pattern. Perhaps, I am reading the results wrong or missing something here. In any case, it would be great to conduct more experiments, for example, presenting patterns in each epoch with the same onset times, and then showing results for a control experiment with different onset times.

2) Line 206: “Since the learning dynamics follows the direction of reducing prediction error, the objective function L decreases across epochs (Fig. 2d)”. Fig. 2d shows a sudden drop at the beginning and constant L for the rest of the training, which seems to contradict line 206 sentence.

3) The Figure 3 results of “recall of learned sequences” are not convincing. Figure 3b bottom seems to show that all neurons are learning to spike together, starting with the ones with larger indexes, and progressively more so with more training. With more training, it seems that the output spike of the first neuron would eventually be treated as the first spike of a “pattern” by all other neurons, and other neurons will generate their output spikes when they receive the output spike from the first neuron. So, I wonder if there are limitations on the kind of patterns that can be recalled? In any case, it would be great to show recall for other types of patterns as well.

4) The manuscript is very difficult to read. The papers’ contribution is also not very clear as the prior work is mentioned throughout the manuscript. For example, the local rule derivation seems to benefit from the work on eprop and surrogate gradient learning.

I urge authors first to give a precise and brief mathematical formulation of the proposed learning rule, i.e., moving equations 2, 3, and possibly 16 from the methods to the beginning of the main text, instead of having an abstract discussion that is more confusing than helping.

Supplementary figure S4 is better than Fig. 1a, which does not give the whole picture. I suggest merging the two or adding more details on Fig. 1a, e.g., showing how $v(t)$ is updated. A related comment is that the weight update is confusing in Fig. S4. I believe the indexes for the weights (red squares) shown at the top should be incremented by one.

There are also many typos and ambiguities.

Left and right qualifiers in the caption of Fig. 1c should be swapped

In the caption of Fig. 1c, “Straight (dashed) line correspond to the input at 2 ms (6 ms)” seems to refer to horizontal lines at 2 s and 6 s on panel c right, which are all straight and dashed. The same problem is in Figure S1.

Figure 1 caption, last sentence: shared  shaded

It is not obvious from Equation 2 why the output spike will reset $v(t)$ to zero, which is mentioned in the text.

Many equations do not specify timestep subscript for w . For example, the timestep subscripts seem to be missing in Eqs. 2 and 3

Line 16: I don't think that the use of "surprising information" is appropriate. The proposed learning rule relies on repeated application of patterns so that the first spike of a pattern can hardly be qualified as surprising.

Line 94: "Thus a synapse gets potentiated if associated input anticipates other presynaptic inputs and it gets suppressed if the related input can be predicted by the past". I have a hard time understanding this sentence.

Line 223: "The plasticity of a pre-synaptic input is defined by its relative timing and it gets potentiated if, on average, it anticipates successive inputs triggering post synaptic spikes". I have a hard time figuring out what "it" refers to.

Fig. 2 caption: homogeneously distributed  uncorrelated and uniformly distributed?

Reviewer #2 (Remarks to the Author):

In this paper the authors present a learning rule for detecting repetitive sequences of synaptic input based on the idea that the state of the membrane potential of a neuron can encode a prediction of future input. This is a novel idea and it is interesting to see their spike-timing-dependent rule derived in a principled way, rather than the more common approach of imposing a kernel to replicate experimental findings. The algorithmic perspective is a strong point of the manuscript. However, the results need further development and more effort needs to be made to place the work in context. This is a rather brief simulation study, and the current manuscript leaves open questions about the broader relevance of the learning rule, regimes in which it is effective, and whether it can be differentiated from other proposed rules in terms of what it can do or what it predicts experimentally. Overall these concerns dampen enthusiasm for the manuscript in its present form.

Major comments:

1. Two applications are presented -- sequence detection in Figure 2, and sequence replay in Figure 3. Based on the basic demonstrations presented, I do not have a clear understanding of the full potential and limitations of the proposed rule. I feel a deeper exploration is warranted for a manuscript at this level. For instance, some typical questions for sequence detection: What is the sequence specificity of the response once it is learned? Can multiple sequences be stored by the same neuron, and if so, what is the capacity, and how does it depend on overlap between participating synapses? How do the results depend on synaptic and membrane time constants? For replay, a single example is shown of sequence completion, but the properties of this phenomenon are not characterized in any generality. For instance, how many inputs are sufficient to trigger replay? Again I was wondering whether such a network has capacity for learning more than one sequence. Because the rule is temporally local, it is unclear what would happen if two sequences shared synapses.

2. As presented, the claims about emergent STDP are slightly misleading. The experimental plasticity protocols the authors claim to replicate involve current injection to elicit postsynaptic spikes. In their implementation in Figure 4, the postsynaptic spike is elicited by another synapse. I assume the triggering synapse is also subject to plasticity. If this is the case, do the results still hold if that synaptic weight is frozen to provide a more faithful approximation of the experimental protocols? Given the difference in implementation, it is also a hard to see how the proposed rule can provide a new interpretation of these classic experimental results. As it stands, wouldn't it require a neuron to have an internal model of current injection as well as the synaptic input? If I have misunderstood, then please clarify the text.

3. Many STDP-type rules have been proposed over the years, so it important to communicate in the manuscript how this one differs in terms of what it can achieve computationally or what it predicts experimentally. The authors have noted that the two main problems studied (Figures 2 and 3) can be solved with previous approaches, so they may wish to focus on making testable predictions that are unique to their model. While this work is certainly based on an interesting idea, more effort should be made to convince biologists in the audience that this could be a viable alternative to current models.

Minor comments:

1. In common with similar studies, it is inaccurate to imply (as the authors do in the abstract) that the rule is truly local. This rule requires that each synapse has access to the somatic membrane potential, but in real neurons, synapses can be located 100s of microns away on the dendrites. So it is only local in the context of the simplifying assumption of a point neuron. See e.g. Bono and Clopath, Nat. Comms. 2017, for an extension of ref. [42] to a more realistic scenario.

2. The claimed relationship to predictive coding theories in the introduction and discussion is hard to follow. The cited studies [4-7] are about using top-down generative models of sensory input for perception, whereas the main application of the learning rule is sequence detection and replay. The authors may wish to lead with the latter for the biological motivation, and save the more tangential links for the discussion.

3. Where multiple runs of simulations are performed, please provide error bars in the plots (e.g. Figure S2)

3. For a simulation study it is preferable to have access to the code during review. In my view this should be considered to be part of the manuscript.

REVIEWER 1

The paper is focused on a specific “predictive” learning rule for spiking neural network (SNN) whose goal is to, as authors put it, “give a credit to those inputs that are informative about future events and discard those inputs that are expected and redundant”. Specifically, authors consider classical discrete-time SNNs with leaky-integrate-and-fire neurons. The predictive learning rule seems to be inspired by the well-known Oja’s rule. The main idea of the proposed rule is to adjust neuron’s weights w at timestep t in such a way as to minimize the distance (l2 norm) between the previous timestep vector of weights $w(t-1)$ scaled by the previous timestep neuron’s membrane potentials $v(t-1)$ and the current timestep vector of the neuron’s pre-synaptic inputs $x(t)$, i.e. minimize the Euclidean distance between $x(t)$ and $w(t-1)*v(t-1)$. The proposed rule is actually a bit more involved as minimization at timestep t seems to be performed over a range of T prior timesteps, i.e. the goal is to minimize the sum of $\|x(t') - w(t'-1)*v(t'-1)\|^2$ over t' , where t' spans from $t-T$ to t .

Another major result is a derivation of a local (online) version of the proposed prediction rule, which depends only on local information available at timesteps t and $t-1$, i.e. $x(t)$, $v(t-1)$, and $s(t-1)$, where the latter is a neuron’s output spike, generated at timestep $t-1$ when membrane potential $v(t-1)$ exceeds a specific threshold.

The proposed rule is studied in several scenarios – by considering a neuron with two inputs and applying a repetitive pattern of two spikes (Fig. 1), more complex scenarios of learning with spatio-temporal patterns with background noise (Fig. 2) and for recurrently connected SNNs (Fig. 3). In the final study (Fig. 4), the authors compare the predictive rule with common STDP rules.

The proposed learning rule and its local version are intriguing. However, I have difficulty understanding some results and have related several concerns.

We thank the reviewer for their comments which helped to improve the manuscript. The main additions to the manuscript are colored in dark red.

Reviewer point 1) I assume that figure 2b shows the results of 100 simulations. Why are average output spike times evolving to zero upon training? Given that “the sequence onset times are homogeneously distributed between 0 and 200 ms”, as shown in Fig. 2a, this behavior is surprising. It seems to me that the desired outcome after learning with the proposed rule would be to spike whenever the first spike of the pattern is presented so that output spike times should be uniformly distributed between 0 and 200 ms? The results make sense if a neuron reacts to background noise spikes unrelated to a pattern. Perhaps, I am reading the results wrong or missing something here. In any case, it would be great to conduct more experiments, for example, presenting patterns in each epoch with the same onset times, and then showing results for a control experiment with different onset times.

In the original manuscript, we did not describe the simulation clearly enough, which made the definition of the time axis confusing. The explanation for the observation made by the reviewer is that, in the original manuscript, the output spike times were aligned to the onset of the sequence in Figure 2b. Thus, the results reported in the original manuscript were as one would expect, and one should not expect a homogeneous distribution of spike times. We have now improved our manuscript to clarify this issue. First, we have changed the title in Figure 2b as “spike times (relative to sequence onset)” and changed the y-label to “time [ms]”. Second, we have explicitly mentioned this information in the Figure caption:

‘Dynamics of the post-synaptic spiking activity during learning. The spike times are defined relative to the actual onset of the sequence in each respective epoch.’

Figure 2: Anticipation of spiking sequences. **a**) Top: Example spike sequence during different training epochs. A spiking sequence is defined by the correlated activity of a subset ($N = 100$) of pre-synaptic neurons. These N pre-synaptic neurons fire sequentially with relative delays of 2 ms, resulting in a total sequence length of 200 ms (pink spike pattern). In each epoch, there are three different sources of noise: (1) jitter of the spike times (random jitter between -2 and 2 ms); (2) random background firing following an homogeneous Poisson process with rate λ distributed between 0 and 10 Hz (see Methods); (3) another subset of 100 pre-synaptic neurons that fired randomly according to an homogeneous Poisson process with randomly distributed rates between 0 and 10 Hz. For each training epoch, the onset of the spike sequences is drawn from an uniform distribution with values between 0 and 200 ms. The bottom plots shows the population firing rate over 10 ms time bins (neuron membrane time constant). **b**) **Dynamics of the post-synaptic spiking activity during learning. The spike times are defined relative to the actual onset of the sequence in each respective epoch.** The bottom plot shows the neuron’s output firing rate within each training epoch. This firing rate was computed across 100 independent simulations (shown are mean and standard deviation). **c**) Top: Dynamics of the normalized synaptic weights \vec{w}/\vec{w}_0 as a function of the training epochs. Here \vec{w}_0 is the weight vector in epoch 0. Above the dashed white line are the 100 background pre-synaptic neurons that do not participate in the sequence. The synaptic weights are ordered along the y-axis from 1 to 100 following the temporal order of the sequence. Bottom: normalized weights of the first 20 inputs at epoch 1000, showing only the first input has been assigned credit. **d**) Left: Normalized objective function \mathcal{L}_{norm} (left plot) as a function of the training epochs. **Different colors correspond to a different number of neurons participating in the sequence.** Right: normalized cumulative membrane potential $\langle v \rangle$. The cumulative membrane potential was computed as the sum of the v_i at each time step in the simulations. The panels shows the mean and standard deviation computed over 100 different simulations.

Conform the request of the reviewer, we have performed additional experiments, by presenting patterns in each epoch with the same onset times, and then showing results for a control experiment with different onset times. The results of this experiment show that we obtain the same behavior, i.e. sequence anticipation, in both cases. These results are shown in Figure S4a-b (see next page). In the same Figure S4, we also show how sequence learning is affected by the presence of two noise sources, i.e. either background homogeneous Poisson process or jitter in the input spike times.

Reviewer point 2) Line 206: “Since the learning dynamics follows the direction of reducing prediction error, the objective function L decreases across epochs (Fig. 2d)”. Fig. 2d shows a sudden drop at the beginning and constant L for the rest of the training, which seems to contradict line 206 sentence.

We thank the reviewer for this comment. In the original version of Figure 2d we used an initialization where all the synaptic weights were fixed to the same value. The initial value of the synaptic weights was close to the minimum reached during training, leading to a fast convergence to the steady value of the loss. In the revised manuscript, we have changed the initialization method, which provides a better generalization and validity of the model. Specifically, the synaptic weights are now initialized randomly from a truncated Gaussian distribution. The mean value of the initial synaptic weights is now higher, which leads to consistent decrease of the loss to the steady value. We described the initialization method in a novel subsection of the Methods which we called *Optimization and initialization scheme*:

“In Figure 2d, the initial weights were randomly drawn from a truncated normal distribution. We bonded the truncated normal distribution to obtain positive values of the initial weights. The variance of the normal distribution was scaled by the squared root of the total input size N . ”

Panel d of Figure 2 now shows the dynamics of the normalized loss \mathcal{L}_{norm} and the normalized cumulative membrane potential $\langle v \rangle_{norm}$ across epochs for different sizes of the pre-synaptic sequence (see legend).

Figure S4: Effect of individual noise sources - Relates to Figure 2 of the main text. **a)** Simulations were performed as in Figure 2, however without any noise source (no spike time jitter, no distractor neurons, no background firing, no random onset of the sequence). Dynamics of the synaptic weights \bar{w} (left) and of the post-synaptic spiking activity (right) across training, without any source of noise. **b)** Same as in **a**, but now with one noise source, in this case the sequence onset drawn from a uniform distribution with values between 0 and 100 ms. **c)** Same as in **a**, with now one noise source, namely random background firing following an homogeneous Poisson process with rate distributed between 0 and 10 Hz. **d)** Same as in **a**, with again one noise source, namely jitter of the input spike times (random jitter between -2 and 2 ms).

Reviewer point 3) The Figure 3 results of “recall of learned sequences” are not convincing. Figure 3b bottom seems to show that all neurons are learning to spike together, starting with the ones with larger indexes, and progressively more so with more training. With more training, it seems that the output spike of the first neuron would eventually be treated as the first spike of a “pattern” by all other neurons, and other neurons will generate their output spikes when they receive the output spike from the first neuron. So, I wonder if there are limitations on the kind of patterns that can be recalled? In any case, it would be great to show recall for other types of patterns as well.

We agree with the the reviewer that longer training in the all-to-all connectivity case would lead to neurons in the network firing shortly after the first input, which exactly reflects the sequence. This is of course a consequence of the all-to-all connectivity, which is likely not a very realistic network architecture. In the revised version of the manuscript, we did novel simulation experiments where we considered a more realistic network architecture which has only local interaction between the units (i.e. no all to all connectivity). We now show this network example (Figure 3, see next page). In this case, the network converges to a compressed sequence with an asymptotic duration (Figure 3d). Note that we have also increased the number of inputs to each neuron in the network. In the revised version of the manuscript, we also did novel simulations for the all-to-all connectivity case. We kept the revised version of the original Figure 3 as a Supplementary Figure S7 (see next pages).

In the revised version of the manuscript, we updated the description of the results of Figure 3 as follows:

“In the previous section, we studied the emergence of anticipatory firing in a single neuron receiving many pre-synaptic inputs. However, in cortical networks, each neuron may receive a large set of pre-synaptic inputs from other areas, as well as recurrent inputs from neurons in the same local network. We therefore investigated a more complex scenario of a network of recurrently coupled neurons that were each endowed with a predictive learning rule. Our simulations were inspired by experimental observations of recall and spontaneous replay after learning. For example, a previous study in rat V1, has shown that the repeated presentation of a sequence of flashes (at different retinotopic locations) gradually leads to a reorganization of spiking activity in the order of the presented sequence [1]. The same study also showed that the presentation of only the first stimulus in the

Figure 3: Sequence anticipation and recall in a network with recurrent connectivity. **a)** In this example, we simulated a network of 10 neurons with nearest-neighbour recurrent connectivity, that is each neuron n in the network received inputs from the $n - 1$ -th and $n + 1$ -th adjacent neurons in the network, respectively. Shown are the connections to the second neuron. Each neurons in the network received inputs from 8 pre-synaptic neurons that fire sequentially with relative delays of 2 ms, resulting in a total sequence length of 16 ms (pink spike pattern). The sequence onset of pre-synaptic inputs for the $n + 1$ -th neuron started 4 ms after the sequence onset for the n -th neuron in the network, etc. Each epoch contained two different sources of noise: (1) random jitter of the spikes in the sequence (between -2 and 2 ms); random background firing of the pre-synaptic neurons according to an homogeneous Poisson process with rate $\lambda = 10$ Hz. Both the connections from the pre-synaptic neurons to the neurons in the network, as well as the connections between the neurons in the network were plastic and modified according to the predictive learning rule described in the main text. **b)** Raster plot of the network’s activity during different epochs of training: (1) The “before” case, where only the pre-synaptic neurons corresponding to the first neuron in the network exhibited sequential firing. In this case the background stochastic firing was still present in all of the $8 \times 10 = 80$ pre-synaptic neurons. (2) The “learning” or conditioning case, where we presented the entire sequence (which was repeated 2000 times). (3) The “after” or “recall” condition, which was the same as the before condition (now after learning). (4) Same as (3), but an example where spontaneous recall occurs due to the background stochastic firing. The neurons are ordered as in panel **a**. **c)** The synaptic weights matrix obtained at the end of training (epoch 1000). Top: The i -th column corresponds to the synaptic weights learned by the i -th neuron in the network, where the 8 entries correspond to the synaptic weights for the pre-synaptic inputs. Bottom: the nearest-neighbour connections in the network towards the i -th neurons. Note that the first and last neuron do not receive inputs from the $n - 1$ -th and $n + 1$ -th neuron, respectively. **e)** Evolution of the duration of network activity across epochs. We computed the temporal difference between the last spike of the last neuron and the first spike of the first neuron to estimate the total duration of the network’s activity. We computed the average duration and the standard deviation from 100 simulations with different stochastic background firing and random jitter of the spike times.

sequence leads to a compressed recall of the entire sequence [1]. Likewise, the sequential activation of neurons in prefrontal cortex and hippocampus is known to lead to subsequent replay at a compressed time-scale [2, 3, 4, 5, 6]. These findings have been interpreted in terms of a local reorganization of synaptic weight distributions as a result of repeated activation with an input sequence [1, 2, 3, 6, 5]. We wondered if a network of recurrently connected neurons with the predictive learning rule described above can develop sequence anticipation as well as (stimulus-evoked) sequence recall and spontaneous replay. We explored the dynamics of a network model where 10 neurons received an input sequence distributed across 80 external units. Each neuron in the network received a unique set inputs from 8 pre-synaptic neurons, which fired sequentially and exhibited stochastic background firing (purple and black in Figure 3a, respectively). The neurons in the network were activated sequentially, such that the inputs into the first neuron arrived earliest, the inputs into the second neuron arrived slightly later, etc. (Figure 3a). The network had a recurrent, nearest-neighbour connectivity scheme, such that each n -th neuron was connected to the neighbouring $n - 1$ -th and $n + 1$ -th neuron (Figure 3a). Thus, each neuron received a set of “afferent” pre-synaptic inputs together with the inputs from the neighbouring neurons (Figure 3a). Both the synaptic connections from the afferent inputs and the recurrent inputs were adjusted by plasticity according to the predictive learning rule described above.

We show the activity of the network for three cases in Figure 3b: (1) The “before” case, where only the pre-synaptic neurons corresponding to the first neuron in the network exhibited sequential firing. In this case the background stochastic firing was still present in all of the 80 pre-synaptic neurons. (2) The “learning” or conditioning case, where we presented the entire sequence (which is repeated 2000 times). (3) The “after” or “recall” condition, which was the same as the before condition, but after learning. We observed that in the before condition, the network activity was relatively unstructured, with firing occurring in the period after the sequence due to the background stochastic firing. During learning, the neurons were active during a relatively long part of the sequence and showed a sequential activation pattern. After learning, the network

Figure S7: Anticipation and recall of sequences with all-to-all connected network. Relates to Figure 3 of the main text. We explored the dynamics of a network model where $N = 8$ neurons received an input sequence distributed across a total of 16 pre-synaptic neurons. Each neuron in the network received 2 pre-synaptic inputs. The pre-synaptic neurons fired in a sequential manner with delays of 2 ms. Each epoch contains two different sources of noise: (1) jitter of the spike times (random jitter between -2 and 2 ms, which is applied randomly in each epoch); and (2) random firing following an homogeneous Poisson process with rate $\lambda = 10$ Hz. The network has a recurrent, all-to-all connectivity scheme and is arranged along the time dimension of the sequence, such that each neuron receives part of the input in a subsequent manner. The sequence onset of pre-synaptic inputs for the $n + 1$ -th neuron started 8 ms after the sequence onset for the n -th neuron in the network, etc. Accordingly, the pattern that each neuron tries to predict is composed by the external pre-synaptic input and by the internally generated activity of the network. **a)** Illustration of the network model, where only 5 neurons of the network are represented for simplicity. **b)** Top: Network spiking activity when presenting only part of the input sequence, namely the sequential input to the first 3 neurons in the network. Note that the other network neurons still received inputs from their corresponding pre-synaptic neurons due to stochastic background firing. Middle: Network spiking activity during the first training epoch, when the entire sequence was presented. Bottom: The first plot shows the network spiking activity at the end of training (epoch 2000) when we presented only part of the sequence (as in the top plot). This figure shows that after learning, the network learns to recall the entire sequence. The second plot shows the network spiking activity when we presented only part of the sequence, but also the end of the sequence (i.e. activating the last two network neurons). There is no effect of activating the neurons at the end of the sequence. Note that random background firing can also elicit spontaneous recall of the sequence, outside the stimulation period. **c)** The synaptic weight matrix obtained at the end of training (epoch 1000). The i -th column (and the top 10 rows) corresponds to the synaptic weights learned from the other neurons to the i -th neuron in the network. The bottom 2 entries correspond to the weights for the pre-synaptic inputs. **d)** Difference between the latency of the first spike of each single neuron in the first epoch of training and after training. Positive values indicate that an earlier latency due to training. The panel shows the mean and standard deviation computed over 100 different simulations.

showed sequential firing upon the presentation of the inputs to the first neuron in the network in the order of the sequence. This sequential firing took place at a compressed time scale. We found that the recall effect was due to the potentiation of the inputs from the $n - 1$ -th neuron to the n -th neuron, as well as potentiation of the first pre-synaptic inputs to the first neurons (Figure 3c). Finally, we observed that sequential firing could also be triggered spontaneously due to the background stochastic activity of the pre-synaptic neurons (Figure 3c; after-spontaneous). Thus, the network exhibited a form of activity that resembles spontaneous replay of sequences.

We further characterized the evolution of the network's output during learning, and the reconfiguration of synaptic weights. We found that after several hundreds of epochs, the network converged onto a stable, sequential output that was time-compressed (Figure 3d). We furthermore quantified the number of neurons that needed to be activated in order for the network to recall the full sequence. We found that this required number of neurons decreased gradually across epochs, indicating a gradual reorganization of the synaptic weight distribution during learning (Figure S6).

To generalize these findings, we also studied a network with all-to-all connectivity, i.e. each neuron was recurrently connected to all of the other neurons in the network. In this case, the network also learned to recall the full sequence on a relatively fast timescale (Figure S7). The output of the network with all-to-all connectivity however differed from the example with recurrent connectivity between neighbours: After prolonged learning,

the other neurons in the all-to-all network all fired shortly after the first neuron was activated (Figure S7).

Together, these results show that a recurrently connected network of neurons each endowed with a predictive learning rule can spontaneously organize to fire preferentially at the beginning of a sequence, and recall (or replay) sequences at a compressed time scale. ”

Reviewer point 4) The manuscript is very difficult to read. The papers’ contribution is also not very clear as the prior work is mentioned throughout the manuscript. For example, the local rule derivation seems to benefit from the work on eprop and surrogate gradient learning. I urge authors first to give a precise and brief mathematical formulation of the proposed learning rule, i.e., moving equations 2, 3, and possibly 16 from the methods to the beginning of the main text, instead of having an abstract discussion that is more confusing than helping. Supplementary figure S4 is better than Fig. 1a, which does not give the whole picture. I suggest merging the two or adding more details on Fig. 1a, e.g., showing how $v(t)$ is updated. A related comment is that the weight update is confusing in Fig. S4. I believe the indexes for the weights (red squares) shown at the top should be incremented by one.

We thank the reviewer for the important comments and suggestions, which helped improving the readability and accessibility of the manuscript. We have followed the suggestions of the reviewer and we believe the revised manuscript now has a clearer structure. We describe what we have done in the following:

1. We have now extended the Methods and Discussion sections of the revised manuscript to clarify the relationship with approximation algorithms for training spiking neural networks, namely the eprop algorithm [7] and the surrogate gradient descend approximation [8]). In the Methods and Discussion sections of the revised manuscript, we clarified our online approximation and we stressed why and how our model does not require approximations of the type of [7] and [8]. Respectively to these works, we have done the following:

- In [7], the authors discuss an approximation to the exact gradient when dealing with recurrent network of spiking neurons, where the basic approximation of the eprop algorithm is to neglect the recurrent interaction between neurons in the computation of the gradient. In the revised manuscript, we clarified the relations between the model proposed here and the eprop algorithms. In the Methods section:

“We are interested in an online learning rule where the weight update forms part of the dynamics of the model, and takes place in real-time with the prediction of the pre-synaptic inputs. This is a typical method in stochastic optimization theory for recursive objective functions [9] and online signal processing [10]. We approximate the learning equation (16) with the current estimate of the gradient

$$\vec{w}_t = \vec{w}_{t-1} - \eta \vec{\nabla}_{\vec{w}} \mathcal{L} \simeq \vec{w}_{t-1} - \eta \vec{\nabla}_{\vec{w}} \mathcal{L} \Big|_{\vec{w}=\vec{w}_{t-1}} = \vec{w}_{t-1} - \eta \left[\vec{\epsilon}_t v_{t-1} + (\vec{\epsilon}_t^\top \vec{w}_{t-1}) \vec{p}_{t-1} \right]. \quad (1)$$

Different from other works [7], our loss function (7) is defined locally at the single neuron level and we thus directly avoid the problem of backpropagation of the gradient through the recurrent interactions between neurons. Therefore, our approximated learning rule is completely online as it only requires information available at time step t . ”

and in the Discussion section:

“Our work is further related to novel approximation algorithms for learning in neural networks such as e-prop [7] or surrogate-gradients techniques [8] as our model provides an online approximation for training spiking neural networks (SNN). In [7] the authors showed that learning in spiking recurrent neural networks can be decomposed into two terms, a global loss, and an eligibility trace which depends on the local state of neurons and results in synaptic weight changes according to local Hebbian plasticity rules. In our model the optimization problem is defined directly at the level of single neurons. Thus, all learning is local in space, i.e. there is no global loss at the network level. By definition, our model hence avoids the problem of propagating the gradient

on recurrent connections [7]. An interesting question for future research is to combine these two approaches by obtaining a completely local learning rule for optimization at the single and network level simultaneously. Indeed, experimental evidence [11, 12] shows that local learning rules implemented by neurons provide a substrate on which global feedback signals act [13], which may provide a biological mechanism for error backpropagation [14]. ”

- The surrogate gradient approximation has been used to avoid the binary output problem when backpropagating the gradient in network of spiking neurons [15, 8, 16]. In the revised manuscript, we clarified the relations between the model proposed here and the surrogate gradient approximation. In the Methods section:

“The first term in equation (12) allows the gradient to flow at every time step t via the dynamics of the membrane potential v_t . The second term has a discontinuous effect in time (at the moment of the output spikes) and depends on the specific nonlinear function. This latter term can be approximated following the surrogate-gradient method [8]

$$-v_{(th)} \frac{\partial s_{t-1}}{\partial v_{t-1}} \simeq \gamma f(v_{t-1}, v_{(th)}), \quad (2)$$

where f is a continuous function of v_t and γ is a scaling factor. In general, the backpropagation of the gradient through the reset mechanisms is neglected [15, 8, 7]. Here we defined the membrane potential v_t as the output variable of the system and the loss function and s_t as an hidden variable for the objective function. Therefore, our implementation directly avoids the problem of backpropagating through discrete output variables. Given these two arguments, we considered $\gamma = 0$ throughout the paper. ”

and in the Discussion section:

“As our loss function depends only on the membrane potential of the cell, our model avoids the problem of propagating the gradient through discrete spikes. It follows that we did not implement a surrogate-gradient approximation [8]. ”

2. We have followed the suggestions of the reviewer to improve the readability of the first section of the Results. As suggested by the reviewer, we inserted the main equations of the model (now Equation 1,2 and 3, respectively) in the main text. In accordance with these changes, we updated Figure 1 (see next page) in light of the new structure of the Results section. Figure 1 now provides both a graphical description of the model and the algorithmic representation via the computational graph, as it was in Figure S4. We therefore removed Figure S4 from the Supplementary Information. Finally, we checked and corrected all the subscripts for the weights in the computational graph.

The first paragraph of the revised Results section is as follows:

We formalized the proposed predictive process in the following single-neuron model: In this model, at each moment in time t , the neuron integrates the present pre-synaptic inputs in the current state of the membrane potential and extracts from its dynamics a prediction of the future input states (see the Methods section for a detailed account of the model and analytical derivations). We first defined the membrane potential v_t as a linear filter, such that the neuron updates its membrane potential recursively by encoding the actual input at time t and the previous value of the membrane potential at time $t-1$ (Equation (3)). The membrane potential at a given time is the result of the temporal summation of previous synaptic inputs and the membrane potential thereby encodes a compression of the high-dimensional input dynamics in time. This is described by the system of equations

$$\begin{cases} v_t = \alpha v_{t-1} + \vec{w}_t^\top \vec{x}_t - v_{th} s_{t-1} \\ s_t = H(v_t - v_{th}). \end{cases} \quad (3)$$

Figure 1: Description of the predictive plasticity rule. **a**) Illustration of the model and the computational graph corresponding to the learning algorithm. Top: at time step t , the neuron computes a prediction of the new input \vec{x}_t from the previous membrane potential v_{t-1} and synaptic weight vector \vec{w}_{t-1} (see Equation (4)). The mismatch between the present input and its prediction is computed locally at every synapse. The prediction error is used to drive synaptic plasticity and update the synaptic weight vector \vec{w}_{t-1} (see Equation (5)). Bottom: the neuron updates its membrane potential by encoding the actual input \vec{x}_t via the learned weight vector \vec{w}_t and its previous internal state v_{t-1} (see Equation (3)). If the voltage exceeds the threshold, an output spike is emitted (shown in yellow) and this spiking event reduces the membrane potential by a constant value at the next time step. Otherwise, the value of the membrane potential v_t is kept and passed to the next time step. **b**) In the simulation illustrated here, we considered a pattern of two pre-synaptic spikes from two different pre-synaptic neurons with a relative delay of 4 ms. Shown are the dynamics of the membrane potential at the first training epoch and after 100 iterations. The neuron learns to fire ahead of the input that arrives at 6 ms (i.e. pre-syn neuron 2). **c**) Top: Dynamics of the weights for different initial conditions (i.e. the weights at epoch 0). The unbroken and dashed lines correspond, respectively, to the pre-synaptic inputs arriving at 2 ms (w_1 , pre-synaptic neuron 1) and 6 ms (w_2 , pre-synaptic neuron 2). Bottom: evolution of the output spike times across epochs. The bottom and top plot have the same color code. **d**) The flow field in the parameter space was obtained by computing the difference between the weight vector (w_1, w_2) in the first epoch and after 10 epochs. The blue lines represent the partition given by the number of spikes that are fired. Note that when the synaptic weights are larger, the neuron fires more spikes. The black arrow shows the trajectory of the weights obtained by training the model for 500 epochs with initial conditions $\vec{w}_0 = (0.005, 0.005)$. The shaded region shows the section of the parameter space where the neuron fires ahead of the input at 6 ms from neuron 2.

10 Here, the temporal integration of the inputs \vec{x}_t is weighted by a synaptic weight vector \vec{w}_t , which gives
 11 different credit to different synapses. Together with the recurrent dynamics of the membrane voltage, we
 12 set a spiking threshold in Equation (3). Accordingly, if the membrane potential reaches the threshold
 13 at time step $t - 1$, the cell fires a post-synaptic spike s_t and the voltage is decreased by v_{th} at the next
 14 timestep.

15 The objective of the neuron is to recursively compute a local prediction of its own inputs by using the
 16 temporal relations in the input spike trains. The prediction of the incoming pre-synaptic input at time
 17 step t is given by the weight of the associated synapse and the previous state of the membrane potential,
 18 i.e.

$$\mathcal{L}_t \equiv \sum_{t=0}^T \frac{1}{2} \|\vec{x}_t - v_{t-1} \vec{w}_{t-1}\|_2^2. \quad (4)$$

19 We then derived a predictive learning rule analytically by minimizing the mismatch between the actual
 20 input and the prediction. This mismatch can be interpreted as a prediction error, which can be computed
 21 with local synaptic states and in real-time based on the dynamics of the inputs (see Methods). By letting
 22 the synaptic weights \vec{w}_t evolve in real-time with the dynamics of the input, we obtained our predictive

plasticity rule

$$\vec{w}_t = \vec{w}_{t-1} + \eta \left[\vec{\epsilon}_t v_{t-1} + (\vec{\epsilon}_t^\top \vec{w}_{t-1}) \vec{p}_{t-1} \right]. \quad (5)$$

Here, η defines the timescale of plasticity, \vec{p}_{t-1} is an input-specific eligibility trace (see Methods), and $\vec{\epsilon}_t$ is the prediction error

$$\vec{\epsilon}_t \equiv \vec{x}_t - v_{t-1} \vec{w}, \quad (6)$$

which defines the sign and amplitude of plasticity. Consequently, synaptic weights undergo potentiation or depotentiation depending on the predictability of the inputs. Thus, a synapse gets respectively potentiated or suppressed if the associated input anticipates or is anticipated by other pre-synaptic inputs.

The computational steps of the predictive neuron model are as follows: (1) At each time step t , the objective function \mathcal{L}_t is evaluated as the neuron learns to predict the current input (Figure 1a, top); (2) the prediction error ϵ_t is used to drive plasticity and update the synaptic weights via Equation (5), and the current input \vec{x}_t is encoded by updating the state variables of the neuron (Figure 1a, bottom). The rule is composed of three terms: (1) A first-order correlation term $\vec{x}_t v_{t-1}$; (2) a normalization term $-v_{t-1}^2 \vec{w}_{t-1}$, which stabilizes learning [17, 18, 19] as has been observed experimentally [20, 21, 22]; (3) an heterosynaptic plasticity term $(\vec{\epsilon}_t^\top \vec{w}_{t-1}) \vec{p}_{t-1}$ [23, 24]. Accordingly, the prediction of future inputs can be computed at the synaptic level in the point-neuron approximation based only on locally available information. On a long timescale, the neuron learns a specific set of synaptic strengths by adjusting the synaptic weight continuously as it collects evidence in its membrane potential.

To illustrate the development of anticipatory firing for a simple example, we exposed the neuron to a sequence of two input spikes coming from two different pre-synaptic neurons that fire with a relative delay of 4 ms (Figure 1b). In this simple scenario, the first pre-synaptic input is predictive of the following pre-synaptic input and should thus be potentiated, driving the neuron to fire ahead of the EPSP (excitatory post-synaptic potential) caused by the second input spike. We trained the model by repeating the input pattern for 300 epochs of duration $T = 500$ ms. During the training period, the neuron learns to adjust its output spike time and to eventually fire ahead of the pre-synaptic input 2, which arrives at 6 ms (Figure 1b). The neuron converges onto an anticipatory “solution” by a selective adjustment of the synaptic weights (Figure 1c, top). In particular, the neuron assigns credit to the pre-synaptic input 1, which arrives at 2 ms, and depotentiates the strength of the input arriving at 6 ms. Accordingly, this leads to the anticipation of the predictable input (Figure 1c, bottom). We further observed that the parameter space given by (w_1, w_2) is partitioned in different regions depending on the amount of spikes fired by the post-synaptic neuron (Figure 1d). The symmetry of the weight dynamics is broken when the membrane potential reaches the threshold and an output spike is fired (Figure 1d). The learning dynamics are qualitatively the same when the initial conditions lie in regions of multiple output spikes (Supplementary Figure S1).

3. We further clarified the structure of the Result section. In the revised version of the manuscript, we summarized and clarified the text of the second sub-paragraph “Dynamics of anticipation and predictive plasticity” and we moved the revised text in the first sub-paragraph “Model of prediction at the single neuron level” (here in line 40 to line 55, see previous point). We therefore removed the sub-paragraph “Dynamics of anticipation and predictive plasticity” from the revised version of the manuscript.
4. We recognized with hindsight that the Discussion of the original manuscript was not sufficiently well organized. In the revised manuscript, the Discussion section now has a clear summary and also has a clear structure. In brief, the revised version is divided in 4 sub-sections: (a) we summarize the main results described in the manuscript; (b) we discuss the relationship with predictive processing models; (c) we discuss the relationship with experimental evidence and phenomenological models for STDP; (d) we discuss the relationship to learning rules at the network level. The four paragraphs of the revised Discussion section are quoted below:

- (a) A summary of the main results described in the manuscript:

The anticipation of future events is a core feature of intelligence and critical for the survival of the

organism. Here, we studied how individual neurons can learn to predict and fire ahead of sensory inputs. We propose a plasticity rule based on predictive processing, where an individual neuron learns a low-rank model of the synaptic input dynamics in its membrane potential. Accordingly, the sign and magnitude of synaptic plasticity are determined by the timing of the pre-synaptic inputs. That is, synapses are potentiated to the degree that they are predictive of future input states, which provides a solution to an optimization problem that can be implemented at the single-neuron level using only local information. We show that neurons endowed with such plasticity rule can learn sequences over long timescales and shift their spikes towards the first inputs in a sequence (i.e. anticipation). Furthermore, neurons represent their inputs in a more efficient manner (i.e. with reduced overall membrane potential). This anticipatory mechanism was able to explain the development of anticipatory signalling and recall in response to sequence stimuli. Finally, we demonstrated that the learning rule described here gives rise to several experimentally observed STDP mechanisms, including: asymmetric STDP kernels [25, 26], as well as symmetric ones [27, 28] given the initial conditions; the frequency-dependence of STDP [29]; the number of post-synaptic spikes in a burst or post-pre pairing [30]; the dependence of (de)potentiation on the initial synaptic strength [31]. Together, our results indicate that prediction may be a guiding principle that orchestrates learning and synaptic plasticity in single neurons, providing a novel interpretation of STDP phenomena.

- (b) A discussion on the relationship between the model described in the main text and predictive processing models of sensory perception in the cortex. We highlighted the main differences and we discussed how the model described here can complement on previous theories:

We first discuss how our results relate to previous theories of coding in cortical networks that emphasize the importance of predictions. An influential theory of cortical function is hierarchical predictive coding (HPC). The basic understanding of HPC is that the brain maintains a model or representation of current and future states in the outside world, and updates this model as new information comes in. HPC posits that the inference process is implemented by the feedforward routing of surprising or unpredicted signals (i.e. prediction errors), and the routing of sensory predictions down the hierarchy via feedback (FB) projections [32, 33, 34]. The predictive plasticity mechanism that we described here differs from HPC models in several aspects, for example: (1) In HPC, prediction is the result of network interactions, in particular the cancellation of feedforward drive by inhibitory feedback. In our model, prediction results from plasticity at a single neuron level. (2) Different from HPC, in our model the neuron does not explicitly transmit (encode) prediction and error signals. (3) Both in HPC and our model, neurons may exhibit reduced firing for predicted as compared to unpredicted sensory inputs. Yet, in our model this is due to depotentiation of predictable inputs, whereas in HPC it is due to inhibitory feedback mediated by top-down projections. We note that our plasticity model is fully compatible with another flavor of predictive processing, namely “coding for prediction”. According to this theory, neurons primarily transmit information about sensory inputs that carry predictive information about the future, as observed in the retinal neural circuits [35]. The findings here may also be relevant to understand the development of anticipatory firing in sensory systems [36, 1, 37], temporal difference learning [38, 39], as well as the compression of sequences during resting state based on prior experience [2, 40]. Finally, a recent work showed that neural activity in the auditory cortex can be predicted roughly 10-20 ms in advance and that these predictions can be exploited at the single neuron level to achieve high performance in classification tasks [41]. However, prediction in the model of [41] does not happen in a unsupervised manner in time as their method relies on the combination of the single neuron prediction with a supervised teaching signal, a novel implementation of Contrastive Hebbian Learning [42].

- (c) A discussion on the relationship between the learning rule derived in the main text and experimental evidence and phenomenological models of STDP. We also discussed what biological substrate of our learning rule is:

Next, we discuss how our findings relate to STDP experiments and models, and the biological substrate of the learning rule described here. STDP is an established experimental phenomenon

which has been widely observed in-vitro [26, 25]. There is evidence for a variety of STDP kernels [43], which dependent on several post-synaptic variables like backpropagating action potentials (bAP) [44], post-synaptic bursts [45] and the dendritic location of inputs [22]. These experimental findings are all obtained in in-vitro preparations. Thus it is unclear what the nature of STDP in-vivo is. The standard protocol for testing STDP has two major limitations that deviate from the normal physiological setting: (1) The protocol involves current injection in the post-synaptic neuron. The current injection itself is not subject to (physiological) plasticity and might therefore not be a good “proxy” for post-synaptic depolarization induced by natural pre-synaptic inputs in-vivo. (2) Several studies have pointed out that different post-synaptic signals (e.g. spike times, depolarization level, dendritic spikes) are relevant for STDP [44, 31, 46]. It is still a matter of debate what is the crucial post-synaptic variable for plasticity [47]. In principle, STDP models might apply both to cases with artificial currents as well as physiological pre-synaptic inputs. An artificial depolarization caused by current injections can lead to plasticity in both our model and in STDP models. Yet it is an open experimental question what is the nature of learning rules when it comes to physiological synaptic inputs and their timing relationships. For example, it is known that to induce LTP (Long Term Potentiation), it is not necessary to evoke a post-synaptic spike [46, 48]. The learning rule proposed here predicts that, in-vivo, pre-synaptic inputs causing a post-synaptic spike will eventually become depotentiated if they are anticipated (i.e. predicted) by other pre-synaptic inputs in a sequence. Our model entails that the prediction of future inputs is driven by the interaction between different synapses, i.e. heterosynaptic plasticity, which has been observed experimentally [23, 24, 49] and proposed as a computationally powerful mechanism for learning in single neurons [50, 51]. NMDA receptors could naturally operate as voltage-gated units for prediction errors. This is for two reasons: (1) NMDARs are voltage-dependent; and (2) NMDARs allow for a comparisons between the internal state of the neuron with external inputs [52]. Finally, experimental evidence [53, 54, 55, 56] and theoretical studies [57] support the hypothesis that active ion-channels along dendritic compartments strongly enrich the dynamical repertoire of neurons and underpin higher computational capabilities.

Comparing the present results to previous work, we emphasize that we did not construct a learning rule to reproduce experimentally observed STDP phenomena. Indeed, several phenomenological (i.e. descriptive) STDP models have previously been proposed to fit experimental data [58, 59, 60, 61] providing mathematical tools to describe, model and predict the behavior of neurons. However, these phenomenological models might not fully explain the computational significance of STDP mechanisms, nor the algorithms from which these biological implementations can emerge. Our approach differs from these models in that we took an optimization problem based on prediction of future inputs as a starting point. From this optimization problem we derived a learning rule which gave rise to experimentally observed STDP mechanisms. Our results, together with previous studies [62, 63, 64], suggest that STDP is a consequence of a general learning rule given the particular state of the system, the stimulation protocol and the specific properties of the input. As a consequence, several STDP learning windows which are described by other phenomenological rules are predicted by our model, as well as the dependence on synaptic strength and depolarization level.

An example of an established phenomenological model of STDP is the one developed by Clopath et al [65]. The authors, guided by experimental evidence [66, 31, 22], modeled the role of the membrane voltage as the relevant post-synaptic variable for synaptic plasticity. The plasticity model described in [65] can accurately reproduce a wide range of experimental findings which, to our knowledge, is not possible with STDP learning rules that are only based on spike timing [61]. The Clopath rule is based on the two-threshold dynamics observed by Artola et al [66] and the authors assume an Adaptive-Exponential I&F (AeI&F) model for the voltage dynamics [67], together with additional variables for the spike after-potential and an adaptive threshold. In agreement with the results of [65], we were able to account for a wide range of phenomena with a simpler model of voltage dynamics, supporting the idea that the history-dependent effect of the membrane potential is pivotal to plasticity. Our model also predicts the experimental observation that the amount of LTP has an inverse dependence on the initial strength of the synaptic input [31]. To our knowledge, this

finding is not described or predicted by the model of [65], because it does not include a dependence on the initial strength of the synapse. Another unique feature of the learning rule described here is that it can produce different several STDP kernels (e.g. asymmetrical, symmetrical) depending on the initial conditions.

The learning rule described here and phenomenological STDP models might also lead to similar behavior in terms of spiking output in response to sequences. In agreement with the present work, previous studies have shown that phase precession can lead to the learning of temporal sequences through an asymmetric learning window as in spike-timing-dependent plasticity (STDP) [68, 69, 70]. Modelling studies have shown that a post-synaptic neuron endowed with an LTD-dominated STDP model can exhibit potentiation of the first synaptic inputs in a temporal sequence, leading to a decrease in the latency of the post-synaptic response [71, 72]. A key difference with our work, however, is that the predictive plasticity rule described here does not produce asymmetric STDP under all conditions. In fact, the degree of potentiation and depotentiation in our model depends on the initial state. That is, there is no fixed STDP kernel in our model. Another difference is that our model can anticipate sequences independent from the initial conditions of synaptic weights (in contrast to [72]), for a wide range of sequence lengths, and pre-synaptic population size. In the model described here, we show that the anticipation of sequences is a convergence point during learning and thereby it is a general solution for a wide set of model parameters.

- (d) A discussion on relationship to learning rules at the network level and the relationships to approximation techniques for training spiking neural network (SNN) models such as e-prop and surrogate-gradients [7, 8]:

Thus, we propose that a single neuron perspective on prediction and anticipatory mechanisms is important as the implementation of any plasticity rule is ultimately achieved at the neuronal level, thereby guiding behavior at the system level. Yet, it is obvious that single neurons are embedded into networks and different means of communication can lead to more complex learning rules in which the single-neuron learning rule described here might be one component. Indeed, it is possible that certain empirical phenomena like sequence recall additionally depend on network dynamics instead of single-neuron learning rules. For example, the faster recall of sequences in the visual system observed in [1] was reproduced in a recent work [73]. The authors developed a biologically realistic network model which differs from our implementation in several ways: (a) a network of both excitatory and inhibitory neurons, (b) a random gaussian connection probability, (c) a leaky I&F model with conductance-based AMPA, GABA and NMDA synaptic currents, (d) several network hyperparameters such as synaptic delays, (e) a short-term depression model and a specific multiplicative, NN-STDP model. While the model in [73] gives a biological explanation based on the conductive properties of the ion-channels, the network implementation of our plasticity rule provides a principle approach to understand fast sequence recall as it is observed also in other brain areas, e.g. different primary sensory areas [36, 74] or the hippocampus [40]. We qualitatively reproduced the faster recall of sequences with a much simpler model, supporting the pivotal role of spike times and excitability for the phenomenon.

Our work is further related to novel approximation algorithms for learning in neural networks such as e-prop [7] or surrogate-gradients techniques [8] as our model provides an online approximation for training spiking neural networks (SNN). In [7] the authors showed that learning in spiking recurrent neural networks can be decomposed into two terms, a global loss, and an eligibility trace which depends on the local state of neurons and results in synaptic weight changes according to local Hebbian plasticity rules. In our model the optimization problem is defined directly at the level of single neurons. Thus, all learning is local in space, i.e. there is no global loss at the network level. By definition, our model hence avoids the problem of propagating the gradient on recurrent connections [7]. An interesting question for future research is to combine these two approaches by obtaining a completely local learning rule for optimization at the single and network level simultaneously. Indeed, experimental evidence [11, 12] shows that local learning rules implemented by neurons provide a substrate on which global feedback signals act [13], which may provide a biological mechanism for error backpropagation [14]. As our loss function depends only on the

membrane potential of the cell, our model avoids the problem of propagating the gradient through discrete spikes. It follows that we did not implement a surrogate-gradient approximation [8].

Looking forward, further investigation on single neurons anticipating local inputs and on their interplay through network interactions is key to understand how complex prediction strategies can emerge. Moreover, synapses can be located far from the soma along the dendritic arbor and might not be able to access the somatic membrane potential directly, with strong consequences on plasticity [22]. Other post-synaptic events such as NMDA spikes or plateau potentials can have an effect on plasticity rules based on membrane voltage, see e.g. [75]. Thereby, spatially segregated dendrites and spatio-temporal integration of events along the neuronal compartments could drastically increase the complexity of prediction obtainable at the single neuron level.

There are also many typos and ambiguities.

We have very carefully proof-read the revised manuscript to remove any typos, grammar mistakes or ambiguities.

Left and right qualifiers in the caption of Fig. 1c should be swapped. In the caption of Fig. 1c, “Straight (dashed) line correspond to the input at 2 ms (6 ms)” seems to refer to horizontal lines at 2 s and 6 s on panel c right, which are all straight and dashed. The same problem is in Figure S1. Figure 1 caption, last sentence: shared –; shaded.

In the revised version of the manuscript, we added a graphical description of the example simulation in Figure 1b and we clarified the graphical description of the results. We corrected all the typos and we changed the caption of Fig 1c as follows:

‘ Top: Dynamics of the weights for different initial conditions (i.e. the weights at epoch 0). The unbroken and dashed lines correspond, respectively, to the pre-synaptic inputs arriving at 2 ms (w_1 , pre-synaptic neuron 1) and 6 ms (w_2 , pre-synaptic neuron 2). Bottom: evolution of the output spike times across epochs. The bottom and top plot have the same color code. .’

It is not obvious from Equation 2 why the output spike will reset $v(t)$ to zero, which is mentioned in the text.

The reset is implemented as a negative feedback that reduces the membrane potential to a constant value after an output spike, similarly to the spike response model from Gerstner et al [76]. In the revised version of the manuscript, we clarified the description of the reset mechanism in the Methods:

“We used a leaky Integrate-and-Fire-like (LIF) model of the form

$$\tau_m \frac{dv(t)}{dt} = -v(t) + \vec{w}^\top \vec{x}(t) - v_{(th)} \sum_j \delta(t - t_j). \quad (7)$$

Here, $v \in \mathbb{R}$ is the membrane potential, τ_m is the membrane time constant, $\vec{x}(t) \in \mathbb{R}^N$ is the pre-synaptic input, $\vec{w} \in \mathbb{R}^N$ is the weight vector and $v_{(th)}$ is the spiking threshold (the subscript (th) : “threshold”). The sum in the last term runs over all the post-synaptic spike times t_j and $\delta(\cdot)$ is the Dirac delta function. Without loss of generality, we set the resting state of the membrane potential to zero. We used a discrete time-step h to discretize equation (7), yielding a recurrence model of the form

$$\begin{cases} v_t = \alpha v_{t-1} - v_{(th)} s_{t-1} + \vec{w}^\top \vec{x}_t \\ s_t = H(v_t - v_{(th)}). \end{cases} \quad (8)$$

Here, $\alpha \equiv 1 - h/\tau_m$ and $H(\cdot)$ is the Heaviside function. The variable $s_t \in \{0, 1\}$ takes binary values and indicates the presence or absence of an output spike at timestep t . *If the voltage exceeds the threshold, an output spike is emitted and this event reduces the membrane potential by a constant value $v_{(th)}$ at the next time step. This implementation of the membrane potential reset relates our model to the spike response model [76]. We set*

$h = 0.05$ ms in all numerical simulations. ”

We also clarified this point in the novel version of the Result section:

Many equations do not specify timestep subscript for w . For example, the timestep subscripts seem to be missing in Eqs. 2 and 3

We moved Equation 2 and 3 from the methods to the first part of the result section and we added the timestep subscript for the synaptic weight vector \vec{w} where it was relevant.

Line 16: I don't think that the use of “surprising information” is appropriate. The proposed learning rule relies on repeated application of patterns so that the first spike of a pattern can hardly be qualified as surprising.

We reformulated the sentence as follows:

“Predicting future states entails that a system can anticipate and signal events ahead of time. Indeed, there is evidence for anticipatory neural activity in various brain systems [36, 68, 74, 1, 37, 77, 34]. Furthermore, the predictability of sensory events can evoke different neuronal signals, in particular enhanced firing rates for surprising inputs, which may guide the update of model predictions in other brain areas [32, 78, 34, 79]. Yet, the associations among sensory events and their predictability should not only result in specific patterns of neural activity, but should also have specific consequences for synaptic plasticity and neuronal outputs [80]. ”

Line 94: “Thus a synapse gets potentiated if associated input anticipates other presynaptic inputs and it gets suppressed if the related input can be predicted by the past”. I have a hard time understanding this sentence.

We clarified our point in the revised Results section of the manuscript: *“In this simple scenario, the first pre-synaptic input is predictive of the following pre-synaptic input and should thus be potentiated, driving the neuron to fire ahead of the EPSP (excitatory post-synaptic potential) caused by the second input spike. ”*

Line 223: “The plasticity of a pre-synaptic input is defined by its relative timing and it gets potentiated if, on average, it anticipates successive inputs triggering post synaptic spikes”. I have a hard time figuring out what “it” refers to.

We clarified our point in the main text: *“Thus, a synapse gets respectively potentiated or suppressed if the associated input anticipates or is anticipated by other pre-synaptic inputs. ”*

Fig. 2 caption: homogeneously distributed –¿ uncorrelated and uniformly distributed?

We corrected the caption of panel a in Figure 2.

References

- [1] Shengjin Xu et al. “Activity recall in a visual cortical ensemble”. In: *Nature neuroscience* 15.3 (2012), pp. 449–455. 56 57
- [2] David R Euston, Masami Tatsuno, and Bruce L McNaughton. “Fast-forward playback of recent memory sequences in prefrontal cortex during sleep”. In: *science* 318.5853 (2007), pp. 1147–1150. 58 59
- [3] Thomas J Davidson, Fabian Kloosterman, and Matthew A Wilson. “Hippocampal replay of extended experience”. In: *Neuron* 63.4 (2009), pp. 497–507. 60 61
- [4] Albert K Lee and Matthew A Wilson. “Memory of sequential experience in the hippocampus during slow wave sleep”. In: *Neuron* 36.6 (2002), pp. 1183–1194. 62 63
- [5] Zoltán Nádasdy et al. “Replay and time compression of recurring spike sequences in the hippocampus”. In: *Journal of Neuroscience* 19.21 (1999), pp. 9497–9507. 64 65
- [6] William E Skaggs and Bruce L McNaughton. “Replay of neuronal firing sequences in rat hippocampus during sleep following spatial experience”. In: *Science* 271.5257 (1996), pp. 1870–1873. 66 67
- [7] Guillaume Bellec et al. “A solution to the learning dilemma for recurrent networks of spiking neurons”. In: *Nature communications* 11.1 (2020), pp. 1–15. 68 69
- [8] Emre O Neftci, Hesham Mostafa, and Friedemann Zenke. “Surrogate gradient learning in spiking neural networks: Bringing the power of gradient-based optimization to spiking neural networks”. In: *IEEE Signal Processing Magazine* 36.6 (2019), pp. 51–63. 70 71 72
- [9] Herbert Robbins and Sutton Monro. “A stochastic approximation method”. In: *The annals of mathematical statistics* (1951), pp. 400–407. 73 74
- [10] Bin Yang. “Projection approximation subspace tracking”. In: *IEEE Transactions on Signal processing* 43.1 (1995), pp. 95–107. 75 76
- [11] Verena Pawlak et al. “Timing is not everything: neuromodulation opens the STDP gate”. In: *Frontiers in synaptic neuroscience* 2 (2010), p. 146. 77 78
- [12] Wulfram Gerstner et al. “Eligibility traces and plasticity on behavioral time scales: experimental support of neohebbian three-factor learning rules”. In: *Frontiers in neural circuits* 12 (2018), p. 53. 79 80
- [13] J Yu Angela and Peter Dayan. “Uncertainty, neuromodulation, and attention”. In: *Neuron* 46.4 (2005), pp. 681–692. 81 82
- [14] Timothy P Lillicrap et al. “Backpropagation and the brain”. In: *Nature Reviews Neuroscience* 21.6 (2020), pp. 335–346. 83 84
- [15] Guillaume Bellec et al. “Long short-term memory and learning-to-learn in networks of spiking neurons”. In: *arXiv preprint arXiv:1803.09574* (2018). 85 86
- [16] Friedemann Zenke and Surya Ganguli. “Superspike: Supervised learning in multilayer spiking neural networks”. In: *Neural computation* 30.6 (2018), pp. 1514–1541. 87 88
- [17] Elie L Bienenstock, Leon N Cooper, and Paul W Munro. “Theory for the development of neuron selectivity: orientation specificity and binocular interaction in visual cortex”. In: *Journal of Neuroscience* 2.1 (1982), pp. 32–48. 89 90 91
- [18] Erkki Oja. “Simplified neuron model as a principal component analyzer”. In: *Journal of mathematical biology* 15.3 (1982), pp. 267–273. 92 93
- [19] Richard Kempter, Wulfram Gerstner, and J Leo Van Hemmen. “Hebbian learning and spiking neurons”. In: *Physical Review E* 59.4 (1999), p. 4498. 94 95
- [20] Gina G Turrigiano et al. “Activity-dependent scaling of quantal amplitude in neocortical neurons”. In: *Nature* 391.6670 (1998), pp. 892–896. 96 97
- [21] Gina G Turrigiano. “The self-tuning neuron: synaptic scaling of excitatory synapses”. In: *Cell* 135.3 (2008), pp. 422–435. 98 99
- [22] Per Jesper Sjöström and Michael Häusser. “A cooperative switch determines the sign of synaptic plasticity in distal dendrites of neocortical pyramidal neurons”. In: *Neuron* 51.2 (2006), pp. 227–238. 100 101

- 102 [23] Sébastien Royer and Denis Paré. “Conservation of total synaptic weight through balanced synaptic de-
103 pression and potentiation”. In: *Nature* 422.6931 (2003), pp. 518–522.
- 104 [24] Won Chan Oh, Laxmi Kumar Parajuli, and Karen Zito. “Heterosynaptic structural plasticity on local
105 dendritic segments of hippocampal CA1 neurons”. In: *Cell reports* 10.2 (2015), pp. 162–169.
- 106 [25] Guo-qiang Bi and Mu-ming Poo. “Synaptic modifications in cultured hippocampal neurons: dependence
107 on spike timing, synaptic strength, and postsynaptic cell type”. In: *Journal of neuroscience* 18.24 (1998),
108 pp. 10464–10472.
- 109 [26] Henry Markram et al. “Regulation of synaptic efficacy by coincidence of postsynaptic APs and EPSPs”.
110 In: *Science* 275.5297 (1997), pp. 213–215.
- 111 [27] Jiang-teng Lu et al. “Spike-timing-dependent plasticity of neocortical excitatory synapses on inhibitory
112 interneurons depends on target cell type”. In: *Journal of Neuroscience* 27.36 (2007), pp. 9711–9720.
- 113 [28] Rajiv K Mishra et al. “Symmetric spike timing-dependent plasticity at CA3–CA3 synapses optimizes
114 storage and recall in autoassociative networks”. In: *Nature communications* 7.1 (2016), pp. 1–11.
- 115 [29] Robert C Froemke et al. “Contribution of individual spikes in burst-induced long-term synaptic modifi-
116 cation”. In: *Journal of neurophysiology* (2006).
- 117 [30] Robert C Froemke, Dominique Debanne, and Guo-Qiang Bi. “Temporal modulation of spike-timing-
118 dependent plasticity”. In: *Frontiers in synaptic neuroscience* 2 (2010), p. 19.
- 119 [31] Per Jesper Sjöström, Gina G Turrigiano, and Sacha B Nelson. “Rate, timing, and cooperativity jointly
120 determine cortical synaptic plasticity”. In: *Neuron* 32.6 (2001), pp. 1149–1164.
- 121 [32] Rajesh PN Rao and Dana H Ballard. “Predictive coding in the visual cortex: a functional interpretation
122 of some extra-classical receptive-field effects”. In: *Nature neuroscience* 2.1 (1999), pp. 79–87.
- 123 [33] Andre M Bastos et al. “Canonical microcircuits for predictive coding”. In: *Neuron* 76.4 (2012), pp. 695–
124 711.
- 125 [34] Georg B Keller and Thomas D Mrsic-Flogel. “Predictive processing: a canonical cortical computation”.
126 In: *Neuron* 100.2 (2018), pp. 424–435.
- 127 [35] Stephanie E Palmer et al. “Predictive information in a sensory population”. In: *Proceedings of the Na-
128 tional Academy of Sciences* 112.22 (2015), pp. 6908–6913.
- 129 [36] Michael J Berry et al. “Anticipation of moving stimuli by the retina”. In: *Nature* 398.6725 (1999),
130 pp. 334–338.
- 131 [37] Jeffrey P Gavornik and Mark F Bear. “Learned spatiotemporal sequence recognition and prediction in
132 primary visual cortex”. In: *Nature neuroscience* 17.5 (2014), pp. 732–737.
- 133 [38] Shogo Ohmae and Javier F Medina. “Climbing fibers encode a temporal-difference prediction error
134 during cerebellar learning in mice”. In: *Nature neuroscience* 18.12 (2015), pp. 1798–1803.
- 135 [39] Rajesh PN Rao and Terrence J Sejnowski. “Spike-timing-dependent Hebbian plasticity as temporal dif-
136 ference learning”. In: *Neural computation* 13.10 (2001), pp. 2221–2237.
- 137 [40] Kamran Diba and György Buzsáki. “Forward and reverse hippocampal place-cell sequences during rip-
138 ples”. In: *Nature neuroscience* 10.10 (2007), pp. 1241–1242.
- 139 [41] Artur Luczak, Bruce L McNaughton, and Yoshimasa Kubo. “Neurons learn by predicting future activity.”
140 In: *bioRxiv* (2020).
- 141 [42] Fernando J Pineda. “Generalization of back-propagation to recurrent neural networks”. In: *Physical re-
142 view letters* 59.19 (1987), p. 2229.
- 143 [43] Daniel E Feldman. “The spike-timing dependence of plasticity”. In: *Neuron* 75.4 (2012), pp. 556–571.
- 144 [44] Jeffrey C Magee and Daniel Johnston. “A synaptically controlled, associative signal for Hebbian plastic-
145 ity in hippocampal neurons”. In: *Science* 275.5297 (1997), pp. 209–213.
- 146 [45] Thomas Nevian and Bert Sakmann. “Spine Ca²⁺ signaling in spike-timing-dependent plasticity”. In:
147 *Journal of Neuroscience* 26.43 (2006), pp. 11001–11013.

- [46] Nace L Golding, Nathan P Staff, and Nelson Spruston. “Dendritic spikes as a mechanism for cooperative long-term potentiation”. In: *Nature* 418.6895 (2002), pp. 326–331. 148 149
- [47] John Lisman and Nelson Spruston. “Questions about STDP as a general model of synaptic plasticity”. In: *Frontiers in synaptic neuroscience* 2 (2010), p. 140. 150 151
- [48] Jason Hardie and Nelson Spruston. “Synaptic depolarization is more effective than back-propagating action potentials during induction of associative long-term potentiation in hippocampal pyramidal neurons”. In: *Journal of Neuroscience* 29.10 (2009), pp. 3233–3241. 152 153 154
- [49] Rachel E Field et al. “Heterosynaptic plasticity determines the set point for cortical excitatory-inhibitory balance”. In: *Neuron* 106.5 (2020), pp. 842–854. 155 156
- [50] Jen-Yung Chen et al. “Heterosynaptic plasticity prevents runaway synaptic dynamics”. In: *Journal of Neuroscience* 33.40 (2013), pp. 15915–15929. 157 158
- [51] Friedemann Zenke, Everton J Agnes, and Wulfram Gerstner. “Diverse synaptic plasticity mechanisms orchestrated to form and retrieve memories in spiking neural networks”. In: *Nature communications* 6.1 (2015), pp. 1–13. 159 160 161
- [52] Peter H Seeburg et al. “The NMDA receptor channel: molecular design of a coincidence detector”. In: *Proceedings of the 1993 Laurentian Hormone Conference*. Elsevier. 1995, pp. 19–34. 162 163
- [53] Panayiota Poirazi, Terrence Brannon, and Bartlett W Mel. “Pyramidal neuron as two-layer neural network”. In: *Neuron* 37.6 (2003), pp. 989–999. 164 165
- [54] Alon Polsky, Bartlett W Mel, and Jackie Schiller. “Computational subunits in thin dendrites of pyramidal cells”. In: *Nature neuroscience* 7.6 (2004), pp. 621–627. 166 167
- [55] Michael London and Michael Häusser. “Dendritic computation”. In: *Annu. Rev. Neurosci.* 28 (2005), pp. 503–532. 168 169
- [56] Albert Gidon et al. “Dendritic action potentials and computation in human layer 2/3 cortical neurons”. In: *Science* 367.6473 (2020), pp. 83–87. 170 171
- [57] Ilenna Simone Jones and Konrad Paul Kording. “Might a Single Neuron Solve Interesting Machine Learning Problems Through Successive Computations on Its Dendritic Tree?” In: *Neural Computation* 33.6 (2021), pp. 1554–1571. 172 173 174
- [58] Wulfram Gerstner et al. “A neuronal learning rule for sub-millisecond temporal coding”. In: *Nature* 383.6595 (1996), pp. 76–78. 175 176
- [59] Jean-Pascal Pfister and Wulfram Gerstner. “Triplets of spikes in a model of spike timing-dependent plasticity”. In: *Journal of Neuroscience* 26.38 (2006), pp. 9673–9682. 177 178
- [60] Michael Graupner and Nicolas Brunel. “Calcium-based plasticity model explains sensitivity of synaptic changes to spike pattern, rate, and dendritic location”. In: *Proceedings of the National Academy of Sciences* 109.10 (2012), pp. 3991–3996. 179 180 181
- [61] Claudia Clopath and Wulfram Gerstner. “Voltage and spike timing interact in STDP—a unified model”. In: *Frontiers in synaptic neuroscience* 2 (2010), p. 25. 182 183
- [62] Geoffrey Hinton. “How to do backpropagation in a brain”. In: *Invited talk at the NIPS’2007 deep learning workshop*. Vol. 656. 2007. 184 185
- [63] Harel Z Shouval, Samuel S-H Wang, and Gayle M Wittenberg. “Spike timing dependent plasticity: a consequence of more fundamental learning rules”. In: *Frontiers in computational neuroscience* 4 (2010), p. 19. 186 187 188
- [64] Manu Srinath Halvagal and Friedemann Zenke. “The combination of Hebbian and predictive plasticity learns invariant object representations in deep sensory networks”. In: *bioRxiv* (2022). 189 190
- [65] Claudia Clopath et al. “Connectivity reflects coding: a model of voltage-based STDP with homeostasis”. In: *Nature neuroscience* 13.3 (2010), p. 344. 191 192
- [66] Alain Artola, S Bröcher, and Wolf Singer. “Different voltage-dependent thresholds for inducing long-term depression and long-term potentiation in slices of rat visual cortex”. In: *Nature* 347.6288 (1990), pp. 69–72. 193 194 195

- 196 [67] Romain Brette and Wulfram Gerstner. “Adaptive exponential integrate-and-fire model as an effective
197 description of neuronal activity”. In: *Journal of neurophysiology* 94.5 (2005), pp. 3637–3642.
- 198 [68] Mayank R Mehta, Michael C Quirk, and Matthew A Wilson. “Experience-dependent asymmetric shape
199 of hippocampal receptive fields”. In: *Neuron* 25.3 (2000), pp. 707–715.
- 200 [69] Jason J Moore et al. “Linking hippocampal multiplexed tuning, Hebbian plasticity and navigation”. In:
201 *Nature* 599.7885 (2021), pp. 442–448.
- 202 [70] Eric Torsten Reifenshtein, Ikhwan Bin Khalid, and Richard Kempster. “Synaptic learning rules for se-
203 quence learning”. In: *Elife* 10 (2021), e67171.
- 204 [71] Rudy Guyonneau, Rufin VanRullen, and Simon J Thorpe. “Neurons tune to the earliest spikes through
205 STDP”. In: *Neural Computation* 17.4 (2005), pp. 859–879.
- 206 [72] Timothée Masquelier, Rudy Guyonneau, and Simon J Thorpe. “Spike timing dependent plasticity finds
207 the start of repeating patterns in continuous spike trains”. In: *PloS one* 3.1 (2008), e1377.
- 208 [73] Xuhui Huang et al. “Different propagation speeds of recalled sequences in plastic spiking neural net-
209 works”. In: *New Journal of Physics* 17.3 (2015), p. 035006.
- 210 [74] Xiaofeng Lu and James Ashe. “Anticipatory activity in primary motor cortex codes memorized move-
211 ment sequences”. In: *Neuron* 45.6 (2005), pp. 967–973.
- 212 [75] Jacopo Bono, Katharina A Wilmes, and Claudia Clopath. “Modelling plasticity in dendrites: from single
213 cells to networks”. In: *Current opinion in neurobiology* 46 (2017), pp. 136–141.
- 214 [76] Wulfram Gerstner et al. *Neuronal dynamics: From single neurons to networks and models of cognition*.
215 Cambridge University Press, 2014.
- 216 [77] Peter E Keller, Giacomo Novembre, and Michael J Hove. “Rhythm in joint action: psychological and
217 neurophysiological mechanisms for real-time interpersonal coordination”. In: *Philosophical Transac-
218 tions of the Royal Society B: Biological Sciences* 369.1658 (2014), p. 20130394.
- 219 [78] Karl Friston. “The free-energy principle: a unified brain theory?” In: *Nature reviews neuroscience* 11.2
220 (2010), pp. 127–138.
- 221 [79] Cem Uran et al. “Predictive coding of natural images by V1 firing rates and rhythmic synchronization”.
222 In: *Neuron* 110.7 (2022), pp. 1240–1257.
- 223 [80] Donald Olding Hebb. *The organization of behavior: A neuropsychological theory*. Psychology Press,
224 2005.

Reviewer 2

In this paper the authors present a learning rule for detecting repetitive sequences of synaptic input based on the idea that the state of the membrane potential of a neuron can encode a prediction of future input. This is a novel idea and it is interesting to see their spike-timing-dependent rule derived in a principled way, rather than the more common approach of imposing a kernel to replicate experimental findings. The algorithmic perspective is a strong point of the manuscript. However, the results need further development and more effort needs to be made to place the work in context. This is a rather brief simulation study, and the current manuscript leaves open questions about the broader relevance of the learning rule, regimes in which it is effective, and whether it can be differentiated from other proposed rules in terms of what it can do or what it predicts experimentally. Overall these concerns dampen enthusiasm for the manuscript in its present form.

We thank the reviewer for their comments which helped to improve the manuscript. The main additions to the manuscript are colored in dark red.

Reviewer point 1) Two applications are presented – sequence detection in Figure 2, and sequence replay in Figure 3. Based on the basic demonstrations presented, I do not have a clear understanding of the full potential and limitations of the proposed rule. I feel a deeper exploration is warranted for a manuscript at this level. For instance, some typical questions for sequence detection: What is the sequence specificity of the response once it is learned? Can multiple sequences be stored by the same neuron, and if so, what is the capacity, and how does it depend on overlap between participating synapses? How do the results depend on synaptic and membrane time constants? For replay, a single example is shown of sequence completion, but the properties of this phenomenon are not characterized in any generality. For instance, how many inputs are sufficient to trigger replay? Again I was wondering whether such a network has capacity for learning more than one sequence. Because the rule is temporally local, it is unclear what would happen if two sequences shared synapses.

We thank the reviewer for these comments, which led to several new analyses and clarifications in the revised manuscript:

1. To answer the reviewer’s point on multiple-sequence prediction and model capacity, we performed several analyses. Our first addition was the Supplementary Figure S5 (see next page). As we show in Fig S5b, the neuron learns to predict and anticipate several components of the input structure by potentiating the synaptic weights corresponding to the start of each sub-sequence. In panel c we show the dynamics of the synaptic weights across epochs. In panel d-e we quantify the capacity of the model and how the latter depends on the model parameters. We edited the text in the Result section accordingly:

” Finally, we considered the case where the input pattern is composed by different sub-sequences which were spaced in time and belonged to independent subsets of pre-synaptic neurons. We show that the neuron exhibits anticipatory firing also in case of multiple sub-sequences (Fig S5). ”

Overall, these examples shows that individual neurons can learn multiple sequences, at least when these sequences arrive on different synapses. Physiologically, the interpretation of this phenomenon could be dendrite-specific learning. We emphasize that in our model, the neuron neither produces sequences, nor recognizes sequences. Rather, it assigns credit to those synapses that have predictive value. If sequences are defined over the same synapses, then the weight assignment will be a mixture of those sequences. However, in our next point, we show that in a network, neurons may fire selectively for one out of several sequences in the presence of inhibition.

2. Next, we performed another new simulation where we study learning of multiple sequences in a network. We show that the network can learn multiple sequences via the competition between different neurons mediated by recurrent inhibitory mechanisms. Previous work suggests that inhibition leads to a reduction in inter-neuronal correlations [1], and contributes to competitive mechanisms in winner-take-all models

Figure S5: Learning multiple independent sequences and the capacity of the model. Relates to Figure 2 of the main text. a) Example spike sequence during different training epochs. The input spike trains contain a sequence given by the correlated activity of 24 pre-synaptic neurons that fire sequentially with relative delays of 2 ms (purple spike pattern). Each epoch contains two different sources of noise: (1) jitter of the spike times (random jitter between -2 and 2 ms, which is applied randomly in each epoch); and (2) random firing following an homogeneous Poisson process with rate $\lambda = 5$ Hz. The total sequence is divided into 3 sub-sequences of $N = 8$ separated by 20 ms. **b)** Dynamics of the post-synaptic spiking activity during learning for the four different sub-sequences. The neuron learns to represent several components of the input structure by potentiating the unpredictable part, i.e. the start, of each sub-sequence. **c)** Dynamics of the synaptic weights during training. The synaptic weights are ordered from 1 to 24 following the temporal order of the sequence. The synaptic weights corresponding to the first inputs of each sub-sequence are maximally potentiated. **d)** Capacity of the model α for different values of model parameters τ_m (membrane time constant) and v_{th} (spiking threshold). The capacity α is defined as the percentage of sub-sequences that the neuron anticipates at the end of training (same performance criterion as in Figure S3). The sequence was composed by the subsequent firing of $N = 100$ pre-synaptic inputs, divided in N_{sub} sub-sequences with $N_{sub} = 5$, $N_{sub} = 10$ and $N_{sub} = 20$ from left to right. In each condition, the neuron anticipates every sub-sequence in the input for a broad range of model parameters. **e)** Capacity of the model as a function of the number of sub-sequences N_{sub} for a different total number of pre-synaptic neurons N . Here we fixed $v_{th} = 1$ and $\tau_m = 20$ ms. Each color corresponds to a different total number of pre-synaptic neurons. We then quantified the capacity for different lengths of the sub-sequences (i.e. N_{seq}). For example, there are 2 sub-sequences of length N_{seq} for a total number of 200 pre-synaptic neurons. The figure shows that full capacity (i.e. $\alpha = 1$) can be reached even for small sub-sequences, and for both small and large numbers of pre-synaptic neurons. Values higher than $\alpha = 1$ correspond to values of $\alpha = 1$ and are shown as such for visualization purposes.

[2] or STDP-like models [3, 4]. Here, we share with the reviewers a Figure X1 (for review only, see next page) describing how we implemented inhibition in a neural network model and the main results of our simulations. Yet, we believe that this novel set of results is beyond the scope of the current manuscript, and would like to further elaborate and publish this result in a separate manuscript. We believe that this result is important and should be a central rather than side point in a manuscript.

Panel a shows a schematic of the network model, where 10 neurons received 2 different sequences of the type of Figure 2 of the main text, as described in the caption. We implemented all-to-all recurrent inhibition as a synaptic input with fixed, negative weight between the neurons. We show how the specificity of the network output evolved during training for the two presented sequences, see example neurons in Figure X1b. First, the network learn to compress the representation of the sequence in time as in the results in the manuscript (Fig X1c). Strikingly, the neurons in the network learn to represent only one specific sequence and the specificity of the network activity increases across epochs (Fig X1d and e). Before learning, most of the neurons fire for both sequences while, at the end of learning, neurons either fire for one specific sequence or stay silent. The network organizes such that different neurons fire for a different sequence, and the selectivity shows a inverted-U-curve dependence on the total amount of inhibition (Figure X1f). Our preliminary results suggest that at an intermediate level of inhibition, there

is maximal independent encoding of multiple sequences. We also performed other simulations where we increased the number of independent sequences in the input and we quantified the network capacity – also in this case we obtained selectivity (data not shown).

Figure X1: Prediction of multiple sequence via recurrent inhibition. **a)** Schematic of the network model. In this example we considered a network of 10 ordered neurons with all-to-all inhibitory coupling, that is each neuron inhibits the other neurons in the network via an inhibitory synapse with fixed strength $w = 0.05$. Each neurons receives two input spike trains containing two different sequences given by the correlated activity of a total of 100 pre-synaptic neurons firing sequentially with a relative delay δt drawn uniformly between 0 and 4 ms. All the neurons in the pre-synaptic population contributes to both sequences in a different order obtained by randomly shuffling the spike times associated with each neuron. Each example contains two different sources of noise: (1) jitter of the spike times (maximal jitter 1 ms), which is applied randomly in each epoch; and (2) random firing following an homogeneous Poisson process with rate $\lambda = 5$ Hz. **b)** Dynamics of the post-synaptic spiking activity during learning for three example neuron in the network. Shown are a neuron selective to sequence 1, a neuron selective to sequence 2 and a neuron which stops firing for both sequences. Top and bottom plot correspond to firing for sequence 1 and 2, respectively. **c)** Evolution of the mean total duration of network activity for both sequences, across epochs. This duration is defined as the time between the first spike and the last spike in the network. The duration decreases with training, i.e. the network is active during a smaller period of time. **d)** Evolution of selectivity metrics across epochs. The labels "1 seq", "both seqs" and "not firing" refer, respectively, to the percentage of neurons that fire only for one sequence, for both sequences, or for neither sequence. The selectivity metrics are averaged across 100 simulations with different noise realizations. With learning, the network becomes more selective, as most neurons are active for either one or the other sequence. **e)** Left: Evolution of the mean hamming distance between the network selectivity indices across epochs. The Hamming distance is computed, for each epoch separately, between the binary activity for the two sequences (e.g., [1010010001] and [0100100010] yields a Hamming distance of 7). The selectivity metrics are averaged across 100 simulations with different noise realizations. Right: Evolution of the mean Hamming distance between the network selectivity before training and across epochs, computed separately for both sequences. With learning, the activity patterns become more dissimilar between the two sequences (left panel), as well as to the original activity pattern (right panel). **f)** Mean selectivity metrics at the end of training for different values of the fixed strength of inhibitory synapses w , in absolute value. When inhibition is very weak, neurons fire for both sequences; when inhibition is very strong, neurons do not fire; for an intermediate level of inhibition, selectivity emerges.

3. Concerning the question on generalization, we performed a parameter space analysis on the results of Figure 2. We included a novel Supplementary Figure S2 where we show how the learned solution depends on the combination of model parameters τ_m (membrane time constant) and τ_x (synaptic time constant). Panel b shows the total number of output spikes at the end of training for different values of model parameters τ_m and τ_x . Following the reviewer's comment, we also included a parameter sweep analysis for the combination of τ_m and v_{th} (spiking threshold) (Figure S2a). Our analysis show that the model can learn to anticipate sequences for a substantial part of the parameter space. These results are described in the Results text:

"The neuron model is able to predict and anticipate temporal structures in the input for a substantial

range of model parameters (Fig S2). These results do not depend on the initial weight vector (Fig S3a) and the model demonstrates good performances even for longer sequences (Fig S3b) and increased noise amplitude and number of distractors (Fig S3c). ”

Figure S2: Effect of model parameters. Relates to Figure 2 of the main text. a) Simulations were performed as in Figure 2, but now with different values of model parameters τ_m and v_{th} . The color map corresponds to the total number of output spikes at the end of learning. We show regions for the parameter space where the total number of spikes was smaller than 200. The black line outlines the region of parameter space where the neuron (in epoch 1000) fires between 2 and 10 ms after the onset of the input sequence, which can be interpreted as the predictive or anticipatory solution. For the vast portion of the parameter space, the model converges to the anticipatory solution. b) Same as in a) for different values of model parameters τ_m and τ_x .

- As suggested by the reviewer, we improved the description of the potential and limitations of the learning rule by clarifying how our results depend on the type and size of each noise source considered. In Figure S3', we show in panel a (this is panel Figure S3c in the revised manuscript) how the performance depends on the number of distractors in the input, showing that the model can extract and anticipate sequences for a larger number of distractor inputs.

We note that, in the previous version of Fig S3 (which was called Fig S2 in the original manuscript) we showed how the proportion of successful trials depends on the different noise sources. In the revised manuscript, we now show the error as 1 minus the percentage of successful trials in a set of 100 simulations and we show how the error depends on the different noise sources.

Figure S3': Performance as a function of the proportion of distractor inputs - Extracted from Supplementary Figure S3. To quantify the performance we fixed the number of training epochs (2000 epochs), we performed 100 numerical simulation for each condition and we labeled successful simulation based on the criteria of input selectivity and fast anticipation (see Methods). a) Performance as a function of the proportion of distractors inputs, that is pre-synaptic inputs which do not participate to the sequence. The inputs spike trains from these inputs are drawn randomly following an homogeneous Poisson process with rate λ distributed between 0 and 10 Hz.

- With respect to the point of replay, we added a novel Supplementary Figure S6 where we show how the number of inputs needed to trigger a full replay of the sequence changes across learning. At the beginning of learning, the full input sequence is needed, while at the end of training only the first input is necessary to trigger the full replay. We have described this in the Results section:

“We furthermore quantified the number of neurons that needed to be activated in order for the network

to recall the full sequence. We found that this required number of neurons decreased gradually across epochs, indicating a gradual reorganization of the synaptic weight distribution during learning (Figure S6). ”

Figure S6: Number of neurons in the network that need to be activated for sequence recall to occur. Relates to Figure 3 of the main text. a) The y-axis corresponds to the number of neurons in the network that need to be activated by the input sequence in order for recall to occur. For each epoch, we tested how many neurons needed to be activated by the corresponding pre-synaptic sequence to obtain full recall. First, we presented the pre-synaptic sequence corresponding to the first neuron in the network. We observed how many neurons in the network were active after the sequence presentation. Note that if a neuron is not activated by the pre-synaptic sequence, the pre-synaptic neurons still exhibit background firing. Then, we systematically increased the number of neurons in the network which received the corresponding pre-synaptic sequence. We examined the minimum number of neurons in the network that needed to be active such that every other neuron in the network also fired sequentially. We repeated this analysis for each training epoch and for 100 simulations with different noise realizations. At the beginning of training, each sequence input to each neuron in the network is needed to obtain a full recall of the sequence. At the end of training, only the inputs to the first neuron are required to trigger a full recall of the input sequence in the network.

Reviewer point 2) As presented, the claims about emergent STDP are slightly misleading. The experimental plasticity protocols the authors claim to replicate involve current injection to elicit postsynaptic spikes. In their implementation in Figure 4, the postsynaptic spike is elicited by another synapse. I assume the triggering synapse is also subject to plasticity. If this is the case, do the results still hold if that synaptic weight is frozen to provide a more faithful approximation of the experimental protocols? Given the difference in implementation, it is also a hard to see how the proposed rule can provide a new interpretation of these classic experimental results. As it stands, wouldn't it require a neuron to have an internal model of current injection as well as the synaptic input? If I have misunderstood, then please clarify the text.

1. Our simulations that are related to STDP phenomena involved one supra-threshold synaptic input to trigger post-synaptic spikes (which was always pre-synaptic input 2). The reviewer correctly points out that this synapse was also subject to plasticity. This of course would approximate the natural biological situation (with multiple synaptic inputs) very well, but perhaps not necessarily an artificial experimental protocol with current injections. The reviewer suggested more simulations, namely reproducing STDP windows while the supra-threshold synaptic input is fixed (i.e. not plastic). In the revised manuscript, we have added a novel Supplementary Figure S10 (see next page) where we simulated the same pre-post protocol, yet with the essential difference that the synaptic weight of the supra-threshold input was now frozen. Panel (a) of this respective figure show qualitatively similar behavior as compared to the simulation in the main text where the weight of the supra-threshold input was subject to plasticity. We added this result in the main text:

” To further relate our findings to experimental observations in which a current injection protocol was used, we performed simulations in which we fixed the synaptic weight of the supra-threshold input (i.e., in this case it was not adjusted by plasticity). Also in this case, the model displayed an anti-symmetric STDP kernel (Figure S10). ”

Figure S10: Spike-timing-dependent-plasticity with fixed synapses. Relates to Figure 4 of the main text. Simulations were performed as in Figure 3, however we fixed the synaptic weight of the supra-threshold input, that is the one eliciting a spike. Only the weight of the sub-threshold input was plastic. **a)** Weight change (in percentage) as a function of the delay between the two input spikes.

2. In the revised version of the manuscript, we clarified the text in the Results section:

“To directly investigate the relation between STDP and the predictive learning rule described here, we investigated the the dependence of plasticity on the relative timing between inputs. To this end, we performed a simulation that resembled the standard STDP protocol (see Methods). We trained the predictive plasticity model with a sequence of two input spikes from two different pre-synaptic inputs x_1 and x_2 arriving at a relative delay Δt . To approximate the STDP protocol with a current injection that triggers a post-synaptic spike, the initial conditions were chosen such that x_2 triggered a post-synaptic spike, and x_1 was a sub-threshold input. ”

and in the Discussion section:

“These experimental findings are all obtained in in-vitro preparations. Thus it is unclear what the nature of STDP in-vivo is. The standard protocol for testing STDP has two major limitations that deviate from the normal physiological setting: (1) The protocol involves current injection in the post-synaptic neuron. The current injection itself is not subject to (physiological) plasticity and might therefore not be a good “proxy” for post-synaptic depolarization induced by natural pre-synaptic inputs in-vivo. (2) Several studies have pointed out that different post-synaptic signals (e.g. spike times, depolarization level, dendritic spikes) are relevant for STDP [5, 6, 7]. It is still a manner of debate what is the crucial post-synaptic variable for plasticity [8]. In principle, STDP models might apply both to cases with artificial currents as well as physiological pre-synaptic inputs. An artificial depolarization caused by current injections can lead to plasticity in both our model and in STDP models. Yet it is an open experimental question what is the nature of learning rules when it comes to physiological synaptic inputs and their timing relationships. ”

Reviewer point 3) Many STDP-type rules have been proposed over the years, so it important to communicate in the manuscript how this one differs in terms of what it can achieve computationally or what it predicts experimentally. The authors have noted that the two main problems studied (Figures 2 and 3) can be solved with previous approaches, so they may wish to focus on making testable predictions that are unique to their model. While this work is certainly based on an interesting idea, more effort should be made to convince biologists in the audience that this could be a viable alternative to current models.

We thank the reviewer for these comments. We have performed new analyses and improved our discussion to address these questions. We describe here these improvements:

1. In the revised manuscript, we have added new simulations to further test the set of experimental findings that our model is able to reproduce. We performed new simulations where we studied the effect of the initial conditions on the same pre-post pairing protocol. We added the novel results in Figure S9. Our

model describes a switch from potentiation to (minor) depression depending on the initial strength of the synapse, which matches previous experimental findings (Figure 5C in [6]). We added a description of

Figure S9: Dependence of STDP on the initial synaptic weight. Relates to Figure 4 of the main text. a) Simulations were performed as in Figure 4b, however we simulated different initial conditions for the synaptic weight corresponding to the sub-threshold input. Weight change (in percentage) as a function of different initial values for the synaptic weights.

the novel results in the related text of the Results section:

“ A key factor that determines the specific shape of the STDP window may be the initial strength of the synapse. In agreement, experimental work has shown that the amount of plasticity in a standard STDP protocol depends on the initial strength of the pre-synaptic weight [6]. To investigate this, we examined the potentiation of the first input depending on its initial synaptic weight. We found that there was a switch from potentiation to depotentiation as the initial synaptic strength increased (Figure S9), consistent with the experimental observations [6]. ”

2. We have now clearly summarized the different post-dictions and pre-dictions that can be derived from the predictive learning rule. We have changed the Discussion of the revised manuscript as follows:
 - We have clarified how the model proposed in the main text is able to account for several STDP phenomena and what is the relation with standard STDP protocols (in the quotation below, line 1 to 7 and in line 8 to 23);
 - We have revised the specific comparison of our model with the Clopath rule and we clarified how the model described in the main text can stand as a viable alternative (here in line 55 to 62).
 - We have clarified which are the the unique predictions of our model (here in line 22-27 and in line 59-62).
 - We emphasized that the central point of our manuscript is not to provide a model of STDP, but rather a generic learning rule that yields predictive neural activity. Despite the observation that our model can reproduce various experimental phenomena and STDP kernels, we do not aim to directly provide an alternative phenomenological model of STDP, which would be directly optimized to fit specific experimental data. In the Discussion of the revised manuscript, we have clarified this point (here in line 35-47).

Here we report the related parts of the Discussion of the revised manuscript.

First, we improved the summary of our findings:

“Finally, we demonstrated that the learning rule described here gives rise to several experimentally observed STDP mechanisms, including: asymmetric STDP kernels [9, 10], as well as symmetric ones [11, 12] given the initial conditions; the frequency-dependence of STDP [13]; the number of post-synaptic spikes in a burst or post-pre pairing [14]; the dependence of (de)potentiation on the initial synaptic strength [6]. Together, our results indicate that prediction may be a guiding principle that orchestrates learning and synaptic plasticity in single neurons, providing a novel interpretation of STDP phenomena. ”

We furthermore provided a much more organized and in-depth discussion of STDP phenomena:

“Next, we discuss how our findings relate to STDP experiments and models, and the biological substrate of the learning rule described here. STDP is an established experimental phenomenon which has been

widely observed in-vitro [10, 9]. There is evidence for a variety of STDP kernels [15], which dependent on several post-synaptic variables like backpropagating action potentials (bAP) [5], post-synaptic bursts [16] and the dendritic location of inputs [17]. These experimental findings are all obtained in in-vitro preparations. Thus it is unclear what the nature of STDP in-vivo is. The standard protocol for testing STDP has two major limitations that deviate from the normal physiological setting: (1) The protocol involves current injection in the post-synaptic neuron. The current injection itself is not subject to (physiological) plasticity and might therefore not be a good “proxy” for post-synaptic depolarization induced by natural pre-synaptic inputs in-vivo. (2) Several studies have pointed out that different post-synaptic signals (e.g. spike times, depolarization level, dendritic spikes) are relevant for STDP [5, 6, 7]. It is still a matter of debate what is the crucial post-synaptic variable for plasticity [8]. In principle, STDP models might apply both to cases with artificial currents as well as physiological pre-synaptic inputs. An artificial depolarization caused by current injections can lead to plasticity in both our model and in STDP models. Yet it is an open experimental question what is the nature of learning rules when it comes to physiological synaptic inputs and their timing relationships. For example, it is known that to induce LTP (Long Term Potentiation), it is not necessary to evoke a post-synaptic spike [7, 18]. The learning rule proposed here predicts that, in-vivo, pre-synaptic inputs causing a post-synaptic spike will eventually become depotentiated if they are anticipated (i.e. predicted) by other pre-synaptic inputs in a sequence. Our model entails that the prediction of future inputs is driven by the interaction between different synapses, i.e. heterosynaptic plasticity, which has been observed experimentally [19, 20, 21] and proposed as a computationally powerful mechanism for learning in single neurons [22, 23]. NMDA receptors could naturally operate as voltage-gated units for prediction errors. This is for two reasons: (1) NMDARs are voltage-dependent; and (2) NMDARs allow for a comparisons between the internal state of the neuron with external inputs [24]. Finally, experimental evidence [25, 26, 27, 28] and theoretical studies [29] support the hypothesis that active ion-channels along dendritic compartments strongly enrich the dynamical repertoire of neurons and underpin higher computational capabilities.

Comparing the present results to previous work, we emphasize that we did not construct a learning rule to reproduce experimentally observed STDP phenomena. Indeed, several phenomenological (i.e. descriptive) STDP models have previously been proposed to fit experimental data [30, 31, 32, 33] providing mathematical tools to describe, model and predict the behavior of neurons. However, these phenomenological models might not fully explain the computational significance of STDP mechanisms, nor the algorithms from which these biological implementations can emerge. Our approach differs from these models in that we took an optimization problem based on prediction of future inputs as a starting point. From this optimization problem we derived a learning rule which gave rise to experimentally observed STDP mechanisms. Our results, together with previous studies [34, 35, 36], suggest that STDP is a consequence of a general learning rule given the particular state of the system, the stimulation protocol and the specific properties of the input. As a consequence, several STDP learning windows which are described by other phenomenological rules are predicted by our model, as well as the dependence on synaptic strength and depolarization level.

An example of an established phenomenological model of STDP is the one developed by Clopath et al [37]. The authors, guided by experimental evidence [38, 6, 17], modeled the role of the membrane voltage as the relevant post-synaptic variable for synaptic plasticity. The plasticity model described in [37] can accurately reproduce a wide range of experimental findings which, to our knowledge, is not possible with STDP learning rules that are only based on spike timing [33]. The Clopath rule is based on the two-threshold dynamics observed by Artola et al [38] and the authors assume an Adaptive-Exponential I&F (AeI&F) model for the voltage dynamics [39], together with additional variables for the spike after-potential and an adaptive threshold. In agreement with the results of [37], we were able to account for a wide range of phenomena with a simpler model of voltage dynamics, supporting the idea that the history-dependent effect of the membrane potential is pivotal to plasticity. Our model also predicts the experimental observation that the amount of LTP has an inverse dependence on the initial strength of the synaptic input [6]. To our knowledge, this finding is not described or predicted by the model of [37], because it does not include a dependence on the initial strength of the synapse. Another unique feature of the learning rule described here is that it can produce different several STDP kernels (e.g. asymmetrical, symmetrical) depending on the initial conditions.

The learning rule described here and phenomenological STDP models might also lead to similar behavior

in terms of spiking output in response to sequences. In agreement with the present work, previous studies have shown that phase precession can lead to the learning of temporal sequences through an asymmetric learning window as in spike-timing-dependent plasticity (STDP) [40, 41, 42]. Modelling studies have shown that a post-synaptic neuron endowed with an LTD-dominated STDP model can exhibit potentiation of the first synaptic inputs in a temporal sequence, leading to a decrease in the latency of the post-synaptic response [43, 44]. A key difference with our work, however, is that the predictive plasticity rule described here does not produce asymmetric STDP under all conditions. In fact, the degree of potentiation and depotentiation in our model depends on the initial state. That is, there is no fixed STDP kernel in our model. Another difference is that our model can anticipate sequences independent from the initial conditions of synaptic weights (in contrast to [44]), for a wide range of sequence lengths, and pre-synaptic population size. In the model described here, we show that the anticipation of sequences is a convergence point during learning and thereby it is a general solution for a wide set of model parameters.

”

3. With respect to the question of other solutions to the two main problems solved here, we have performed new simulations and added clarifications to the revised manuscript:

- Previous studies [43, 44] have shown that a post-synaptic neuron endowed with a specific LTD-dominated STDP model can decrease the latency of its post-synaptic response to pre-synaptic sequences. In the revised version of the manuscript, we did novel simulation experiments where we show how our model generalizes these results for a substantial range of pre-synaptic population size and for different number of distractors (see point 1 of Reviewer 2). In the Discussion of the revised manuscript, we clarified now the model exhibits sequence anticipation in a general and principled way (here in line 66 to 76, see previous point 2).
- In [45] the authors observed a recall of learned sequences in awake V1 which was 3 to 4 time faster than the actual input, suggesting that sequence recall is primarily determined by the network dynamics than by the speed of conditioning motion. In the Discussion of the revised version of the manuscript, we discuss how the model proposed in the main text differs from previous models and how it qualitatively reproduces the observation in [45] (here in line 84 to 94):

“Thus, we propose that a single neuron perspective on prediction and anticipatory mechanisms is important as the implementation of any plasticity rule is ultimately achieved at the neuronal level, thereby guiding behavior at the system level. Yet, it is obvious that single neurons are embedded into networks and different means of communication can lead to more complex learning rules in which the single-neuron learning rule described here might be one component. Indeed, it is possible that certain empirical phenomena like sequence recall additionally depend on network dynamics instead of single-neuron learning rules. For example, the faster recall of sequences in the visual system observed in [45] was reproduced in a recent work [46]. The authors developed a biologically realistic network model which differs from our implementation in several ways: (a) a network of both excitatory and inhibitory neurons, (b) a random gaussian connection probability, (c) a leaky I&F model with conductance-based AMPA, GABA and NMDA synaptic currents, (d) several network hyperparameters such as synaptic delays, (e) a short-term depression model and a specific multiplicative, NN-STDP model. While the model in [46] gives a biological explanation based on the conductive properties of the ion-channels, the network implementation of our plasticity rule provides a principle approach to understand fast sequence recall as it is observed also in other brain areas, e.g. different primary sensory areas [47, 48] or the hippocampus [49]. We qualitatively reproduced the faster recall of sequences with a much simpler model, supporting the pivotal role of spike times and excitability for the phenomenon.

Our work is further related to novel approximation algorithms for learning in neural networks such as e-prop [50] or surrogate-gradients techniques [51] as our model provides an online approximation for training spiking neural networks (SNN). In [50] the authors showed that learning in spiking recurrent neural networks can be decomposed into two terms, a global loss, and an eligibility trace which depends on the local state of neurons and results in synaptic weight changes according to local Hebbian plasticity rules. In our model the optimization problem is defined directly at the level of single neurons. Thus, all learning is local in space, i.e. there is no global loss at the network level. By definition, our model hence avoids the problem of propagating the gradient on recurrent connections [50]. An interesting question

103 *for future research is to combine these two approaches by obtaining a completely local learning rule*
104 *for optimization at the single and network level simultaneously. Indeed, experimental evidence [52, 53]*
105 *shows that local learning rules implemented by neurons provide a substrate on which global feedback*
106 *signals act [54], which may provide a biological mechanism for error backpropagation [55]. As*
107 *our loss function depends only on the membrane potential of the cell, our model avoids the problem*
108 *of propagating the gradient through discrete spikes. It follows that we did not implement a surrogate-*
109 *gradient approximation [51].*

110 *Looking forward, further investigation on single neurons anticipating local inputs and on their inter-*
111 *play through network interactions is key to understand how complex prediction strategies can emerge.*
112 *Moreover, synapses can be located far from the soma along the dendritic arbor and might not be able to*
113 *access the somatic membrane potential directly, with strong consequences on plasticity [17]. Other post-*
114 *synaptic events such as NMDA spikes or plateau potentials can have an effect on plasticity rules based on*
115 *membrane voltage, see e.g. [56]. Thereby, spatially segregated dendrites and spatio-temporal integra-*
116 *tion of events along the neuronal compartments could drastically increase the complexity of prediction*
117 *obtainable at the single neuron level. ”*

Minor comments

1. In common with similar studies, it is inaccurate to imply (as the authors do in the abstract) that the rule is truly local. This rule requires that each synapse has access to the somatic membrane potential, but in real neurons, synapses can be located 100s of microns away on the dendrites. So it is only local in the context of the simplifying assumption of a point neuron. See e.g. Bono and Clopath, Nat. Comms. 2017, for an extension of ref. [42] to a more realistic scenario.

We agree with the reviewer that the description of the model in the main text was not accurate. In the revised version of the manuscript, we now specify the point-neuron assumption in the main text:

”Accordingly, the prediction of future inputs can be computed at the synaptic level in the point-neuron approximation based only on locally available information. ”

We now also discuss how our results can be extended with an implementation of the learning rule derived here in neurons with morphological structure (here from line 112 to line 117, see previous point in the Discussion).

2. The claimed relationship to predictive coding theories in the introduction and discussion is hard to follow. The cited studies [4-7] are about using top-down generative models of sensory input for perception, whereas the main application of the learning rule is sequence detection and replay. The authors may wish to lead with the latter for the biological motivation, and save the more tangential links for the discussion.

Indeed the reference to Predictive Coding models in the Introduction was misleading. In the current version of the manuscript, we revised our Introduction to discuss how the model proposed in the manuscripts related to common concepts of prediction in sensory perception, and also added references related to anticipatory firing (to add a better motivation):

“Predicting future states entails that a system can anticipate and signal events ahead of time. Indeed, there is evidence for anticipatory neural activity in various brain systems [47, 40, 48, 45, 57, 58, 59]. Furthermore, the predictability of sensory events can evoke different neuronal signals, in particular enhanced firing rates for surprising inputs, which may guide the update of model predictions in other brain areas [60, 61, 59, 62]. Yet, the associations among sensory events and their predictability should not only result in specific patterns of neural activity, but should also have specific consequences for synaptic plasticity and neuronal outputs [63]. ”.

We have improved the organization of our Discussion in the revised manuscript. We now more clearly discuss how the model proposed in the main text relates to several theories of cortical function which rely on

predictive processes:

“We first discuss how our results relate to previous theories of coding in cortical networks that emphasize the importance of predictions. An influential theory of cortical function is hierarchical predictive coding (HPC). The basic understanding of HPC is that the brain maintains a model or representation of current and future states in the outside world, and updates this model as new information comes in. HPC posits that the inference process is implemented by the feedforward routing of surprising or unpredicted signals (i.e. prediction errors), and the routing of sensory predictions down the hierarchy via feedback (FB) projections [60, 64, 59]. The predictive plasticity mechanism that we described here differs from HPC models in several aspects, for example: (1) In HPC, prediction is the result of network interactions, in particular the cancellation of feed-forward drive by inhibitory feedback. In our model, prediction results from plasticity at a single neuron level. (2) Different from HPC, in our model the neuron does not explicitly transmit (encode) prediction and error signals. (3) Both in HPC and our model, neurons may exhibit reduced firing for predicted as compared to unpredicted sensory inputs. Yet, in our model this is due to depotentiation of predictable inputs, whereas in HPC it is due to inhibitory feedback mediated by top-down projections. We note that our plasticity model is fully compatible with another flavor of predictive processing, namely “coding for prediction”. According to this theory, neurons primarily transmit information about sensory inputs that carry predictive information about the future, as observed in the retinal neural circuits [65]. The findings here may also be relevant to understand the development of anticipatory firing in sensory systems [47, 45, 57], temporal difference learning [66, 67], as well as the compression of sequences during resting state based on prior experience [68, 49]. Finally, a recent work showed that neural activity in the auditory cortex can be predicted roughly 10-20 ms in advance and that these predictions can be exploited at the single neuron level to achieve high performance in classification tasks [69]. However, prediction in the model of [69] does not happen in a unsupervised manner in time as their method relies on the combination of the single neuron prediction with a supervised teaching signal, a novel implementation of Contrastive Hebbian Learning [70]. ”.

3. Where multiple runs of simulations are performed, please provide error bars in the plots (e.g. Figure S2)

We included error bars for every simulation in which we had multiple repeats with different noise realization.

3. For a simulation study it is preferable to have access to the code during review. In my view this should be considered to be part of the manuscript.

Together with the novel version of the manuscript, we now provide the “predictive_neuron_code.zip” file containing the code needed to reproduce the figures of the main text. The GitHub repository of the full project will be public upon acceptance for publication.

References

- 118 [1] Alfonso Renart et al. “The asynchronous state in cortical circuits”. In: *science* 327.5965 (2010), pp. 587–
119 590.
- 120 [2] Wolfgang Maass. “On the computational power of winner-take-all”. In: *Neural computation* 12.11 (2000),
121 pp. 2519–2535.
- 122 [3] Wulfram Gerstner, Raphael Ritz, and J Leo van Hemmen. “A biologically motivated and analytically
123 soluble model of collective oscillations in the cortex”. In: *Biological cybernetics* 68.4 (1993), pp. 363–
124 374.
- 125 [4] Timothée Masquelier, Rudy Guyonneau, and Simon J Thorpe. “Competitive STDP-based spike pattern
126 learning”. In: *Neural computation* 21.5 (2009), pp. 1259–1276.
- 127 [5] Jeffrey C Magee and Daniel Johnston. “A synaptically controlled, associative signal for Hebbian plastic-
128 ity in hippocampal neurons”. In: *Science* 275.5297 (1997), pp. 209–213.
- 129 [6] Per Jesper Sjöström, Gina G Turrigiano, and Sacha B Nelson. “Rate, timing, and cooperativity jointly
130 determine cortical synaptic plasticity”. In: *Neuron* 32.6 (2001), pp. 1149–1164.
- 131 [7] Nace L Golding, Nathan P Staff, and Nelson Spruston. “Dendritic spikes as a mechanism for cooperative
132 long-term potentiation”. In: *Nature* 418.6895 (2002), pp. 326–331.
- 133 [8] John Lisman and Nelson Spruston. “Questions about STDP as a general model of synaptic plasticity”.
134 In: *Frontiers in synaptic neuroscience* 2 (2010), p. 140.
- 135 [9] Guo-qiang Bi and Mu-ming Poo. “Synaptic modifications in cultured hippocampal neurons: dependence
136 on spike timing, synaptic strength, and postsynaptic cell type”. In: *Journal of neuroscience* 18.24 (1998),
137 pp. 10464–10472.
- 138 [10] Henry Markram et al. “Regulation of synaptic efficacy by coincidence of postsynaptic APs and EPSPs”.
139 In: *Science* 275.5297 (1997), pp. 213–215.
- 140 [11] Jiang-teng Lu et al. “Spike-timing-dependent plasticity of neocortical excitatory synapses on inhibitory
141 interneurons depends on target cell type”. In: *Journal of Neuroscience* 27.36 (2007), pp. 9711–9720.
- 142 [12] Rajiv K Mishra et al. “Symmetric spike timing-dependent plasticity at CA3–CA3 synapses optimizes
143 storage and recall in autoassociative networks”. In: *Nature communications* 7.1 (2016), pp. 1–11.
- 144 [13] Robert C Froemke et al. “Contribution of individual spikes in burst-induced long-term synaptic modifi-
145 cation”. In: *Journal of neurophysiology* (2006).
- 146 [14] Robert C Froemke, Dominique Debanne, and Guo-Qiang Bi. “Temporal modulation of spike-timing-
147 dependent plasticity”. In: *Frontiers in synaptic neuroscience* 2 (2010), p. 19.
- 148 [15] Daniel E Feldman. “The spike-timing dependence of plasticity”. In: *Neuron* 75.4 (2012), pp. 556–571.
- 149 [16] Thomas Nevian and Bert Sakmann. “Spine Ca²⁺ signaling in spike-timing-dependent plasticity”. In:
150 *Journal of Neuroscience* 26.43 (2006), pp. 11001–11013.
- 151 [17] Per Jesper Sjöström and Michael Häusser. “A cooperative switch determines the sign of synaptic plas-
152 ticity in distal dendrites of neocortical pyramidal neurons”. In: *Neuron* 51.2 (2006), pp. 227–238.
- 153 [18] Jason Hardie and Nelson Spruston. “Synaptic depolarization is more effective than back-propagating
154 action potentials during induction of associative long-term potentiation in hippocampal pyramidal neu-
155 rons”. In: *Journal of Neuroscience* 29.10 (2009), pp. 3233–3241.
- 156 [19] Sébastien Royer and Denis Paré. “Conservation of total synaptic weight through balanced synaptic de-
157 pression and potentiation”. In: *Nature* 422.6931 (2003), pp. 518–522.
- 158 [20] Won Chan Oh, Laxmi Kumar Parajuli, and Karen Zito. “Heterosynaptic structural plasticity on local
159 dendritic segments of hippocampal CA1 neurons”. In: *Cell reports* 10.2 (2015), pp. 162–169.
- 160 [21] Rachel E Field et al. “Heterosynaptic plasticity determines the set point for cortical excitatory-inhibitory
161 balance”. In: *Neuron* 106.5 (2020), pp. 842–854.
- 162 [22] Jen-Yung Chen et al. “Heterosynaptic plasticity prevents runaway synaptic dynamics”. In: *Journal of*
163 *Neuroscience* 33.40 (2013), pp. 15915–15929.

- [23] Friedemann Zenke, Everton J Agnes, and Wulfram Gerstner. “Diverse synaptic plasticity mechanisms orchestrated to form and retrieve memories in spiking neural networks”. In: *Nature communications* 6.1 (2015), pp. 1–13. 164
165
166
- [24] Peter H Seeburg et al. “The NMDA receptor channel: molecular design of a coincidence detector”. In: *Proceedings of the 1993 Laurentian Hormone Conference*. Elsevier. 1995, pp. 19–34. 167
168
- [25] Panayiota Poirazi, Terrence Brannon, and Bartlett W Mel. “Pyramidal neuron as two-layer neural network”. In: *Neuron* 37.6 (2003), pp. 989–999. 169
170
- [26] Alon Polsky, Bartlett W Mel, and Jackie Schiller. “Computational subunits in thin dendrites of pyramidal cells”. In: *Nature neuroscience* 7.6 (2004), pp. 621–627. 171
172
- [27] Michael London and Michael Häusser. “Dendritic computation”. In: *Annu. Rev. Neurosci.* 28 (2005), pp. 503–532. 173
174
- [28] Albert Gidon et al. “Dendritic action potentials and computation in human layer 2/3 cortical neurons”. In: *Science* 367.6473 (2020), pp. 83–87. 175
176
- [29] Ilenna Simone Jones and Konrad Paul Kording. “Might a Single Neuron Solve Interesting Machine Learning Problems Through Successive Computations on Its Dendritic Tree?” In: *Neural Computation* 33.6 (2021), pp. 1554–1571. 177
178
179
- [30] Wulfram Gerstner et al. “A neuronal learning rule for sub-millisecond temporal coding”. In: *Nature* 383.6595 (1996), pp. 76–78. 180
181
- [31] Jean-Pascal Pfister and Wulfram Gerstner. “Triplets of spikes in a model of spike timing-dependent plasticity”. In: *Journal of Neuroscience* 26.38 (2006), pp. 9673–9682. 182
183
- [32] Michael Graupner and Nicolas Brunel. “Calcium-based plasticity model explains sensitivity of synaptic changes to spike pattern, rate, and dendritic location”. In: *Proceedings of the National Academy of Sciences* 109.10 (2012), pp. 3991–3996. 184
185
186
- [33] Claudia Clopath and Wulfram Gerstner. “Voltage and spike timing interact in STDP—a unified model”. In: *Frontiers in synaptic neuroscience* 2 (2010), p. 25. 187
188
- [34] Geoffrey Hinton. “How to do backpropagation in a brain”. In: *Invited talk at the NIPS’2007 deep learning workshop*. Vol. 656. 2007. 189
190
- [35] Harel Z Shouval, Samuel S-H Wang, and Gayle M Wittenberg. “Spike timing dependent plasticity: a consequence of more fundamental learning rules”. In: *Frontiers in computational neuroscience* 4 (2010), p. 19. 191
192
193
- [36] Manu Srinath Halvagal and Friedemann Zenke. “The combination of Hebbian and predictive plasticity learns invariant object representations in deep sensory networks”. In: *bioRxiv* (2022). 194
195
- [37] Claudia Clopath et al. “Connectivity reflects coding: a model of voltage-based STDP with homeostasis”. In: *Nature neuroscience* 13.3 (2010), p. 344. 196
197
- [38] Alain Artola, S Bröcher, and Wolf Singer. “Different voltage-dependent thresholds for inducing long-term depression and long-term potentiation in slices of rat visual cortex”. In: *Nature* 347.6288 (1990), pp. 69–72. 198
199
200
- [39] Romain Brette and Wulfram Gerstner. “Adaptive exponential integrate-and-fire model as an effective description of neuronal activity”. In: *Journal of neurophysiology* 94.5 (2005), pp. 3637–3642. 201
202
- [40] Mayank R Mehta, Michael C Quirk, and Matthew A Wilson. “Experience-dependent asymmetric shape of hippocampal receptive fields”. In: *Neuron* 25.3 (2000), pp. 707–715. 203
204
- [41] Jason J Moore et al. “Linking hippocampal multiplexed tuning, Hebbian plasticity and navigation”. In: *Nature* 599.7885 (2021), pp. 442–448. 205
206
- [42] Eric Torsten Reifenstein, Ikhwan Bin Khalid, and Richard Kempster. “Synaptic learning rules for sequence learning”. In: *Elife* 10 (2021), e67171. 207
208
- [43] Rudy Guyonneau, Rufin VanRullen, and Simon J Thorpe. “Neurons tune to the earliest spikes through STDP”. In: *Neural Computation* 17.4 (2005), pp. 859–879. 209
210

- 211 [44] Timothée Masquelier, Rudy Guyonneau, and Simon J Thorpe. “Spike timing dependent plasticity finds
212 the start of repeating patterns in continuous spike trains”. In: *PloS one* 3.1 (2008), e1377.
- 213 [45] Shengjin Xu et al. “Activity recall in a visual cortical ensemble”. In: *Nature neuroscience* 15.3 (2012),
214 pp. 449–455.
- 215 [46] Xuhui Huang et al. “Different propagation speeds of recalled sequences in plastic spiking neural net-
216 works”. In: *New Journal of Physics* 17.3 (2015), p. 035006.
- 217 [47] Michael J Berry et al. “Anticipation of moving stimuli by the retina”. In: *Nature* 398.6725 (1999),
218 pp. 334–338.
- 219 [48] Xiaofeng Lu and James Ashe. “Anticipatory activity in primary motor cortex codes memorized move-
220 ment sequences”. In: *Neuron* 45.6 (2005), pp. 967–973.
- 221 [49] Kamran Diba and György Buzsáki. “Forward and reverse hippocampal place-cell sequences during rip-
222 ples”. In: *Nature neuroscience* 10.10 (2007), pp. 1241–1242.
- 223 [50] Guillaume Bellec et al. “A solution to the learning dilemma for recurrent networks of spiking neurons”.
224 In: *Nature communications* 11.1 (2020), pp. 1–15.
- 225 [51] Emre O Neftci, Hesham Mostafa, and Friedemann Zenke. “Surrogate gradient learning in spiking neural
226 networks: Bringing the power of gradient-based optimization to spiking neural networks”. In: *IEEE*
227 *Signal Processing Magazine* 36.6 (2019), pp. 51–63.
- 228 [52] Verena Pawlak et al. “Timing is not everything: neuromodulation opens the STDP gate”. In: *Frontiers in*
229 *synaptic neuroscience* 2 (2010), p. 146.
- 230 [53] Wulfram Gerstner et al. “Eligibility traces and plasticity on behavioral time scales: experimental support
231 of neohebbian three-factor learning rules”. In: *Frontiers in neural circuits* 12 (2018), p. 53.
- 232 [54] J Yu Angela and Peter Dayan. “Uncertainty, neuromodulation, and attention”. In: *Neuron* 46.4 (2005),
233 pp. 681–692.
- 234 [55] Timothy P Lillicrap et al. “Backpropagation and the brain”. In: *Nature Reviews Neuroscience* 21.6
235 (2020), pp. 335–346.
- 236 [56] Jacopo Bono, Katharina A Wilmes, and Claudia Clopath. “Modelling plasticity in dendrites: from single
237 cells to networks”. In: *Current opinion in neurobiology* 46 (2017), pp. 136–141.
- 238 [57] Jeffrey P Gavornik and Mark F Bear. “Learned spatiotemporal sequence recognition and prediction in
239 primary visual cortex”. In: *Nature neuroscience* 17.5 (2014), pp. 732–737.
- 240 [58] Peter E Keller, Giacomo Novembre, and Michael J Hove. “Rhythm in joint action: psychological and
241 neurophysiological mechanisms for real-time interpersonal coordination”. In: *Philosophical Transac-*
242 *tions of the Royal Society B: Biological Sciences* 369.1658 (2014), p. 20130394.
- 243 [59] Georg B Keller and Thomas D Mrsic-Flogel. “Predictive processing: a canonical cortical computation”.
244 In: *Neuron* 100.2 (2018), pp. 424–435.
- 245 [60] Rajesh PN Rao and Dana H Ballard. “Predictive coding in the visual cortex: a functional interpretation
246 of some extra-classical receptive-field effects”. In: *Nature neuroscience* 2.1 (1999), pp. 79–87.
- 247 [61] Karl Friston. “The free-energy principle: a unified brain theory?” In: *Nature reviews neuroscience* 11.2
248 (2010), pp. 127–138.
- 249 [62] Cem Uran et al. “Predictive coding of natural images by V1 firing rates and rhythmic synchronization”.
250 In: *Neuron* 110.7 (2022), pp. 1240–1257.
- 251 [63] Donald Olding Hebb. *The organization of behavior: A neuropsychological theory*. Psychology Press,
252 2005.
- 253 [64] Andre M Bastos et al. “Canonical microcircuits for predictive coding”. In: *Neuron* 76.4 (2012), pp. 695–
254 711.
- 255 [65] Stephanie E Palmer et al. “Predictive information in a sensory population”. In: *Proceedings of the Na-*
256 *tional Academy of Sciences* 112.22 (2015), pp. 6908–6913.

- [66] Shogo Ohmae and Javier F Medina. “Climbing fibers encode a temporal-difference prediction error during cerebellar learning in mice”. In: *Nature neuroscience* 18.12 (2015), pp. 1798–1803. 257
258
- [67] Rajesh PN Rao and Terrence J Sejnowski. “Spike-timing-dependent Hebbian plasticity as temporal difference learning”. In: *Neural computation* 13.10 (2001), pp. 2221–2237. 259
260
- [68] David R Euston, Masami Tatsuno, and Bruce L McNaughton. “Fast-forward playback of recent memory sequences in prefrontal cortex during sleep”. In: *science* 318.5853 (2007), pp. 1147–1150. 261
262
- [69] Artur Luczak, Bruce L McNaughton, and Yoshimasa Kubo. “Neurons learn by predicting future activity.” In: *bioRxiv* (2020). 263
264
- [70] Fernando J Pineda. “Generalization of back-propagation to recurrent neural networks”. In: *Physical review letters* 59.19 (1987), p. 2229. 265
266

REVIEWER COMMENTS

Reviewer #3 (Remarks to the Author):

The authors have addressed most of the questions of Reviewer 1. The manuscript is now well organized and understandable. I will discuss a few points with which I was not fully satisfied and add some additional comments about the work.

(1) Reviewer 1, point 3: The Figure 3 results of “recall of learned sequences” are not convincing. Figure 3b

bottom seems to show that all neurons are learning to spike together, starting with the ones with larger indexes, and progressively more so with more training. With more training, it seems that the output spike

of the first neuron would eventually be treated as the first spike of a “pattern” by all other neurons, and other neurons will generate their output spikes when they receive the output spike from the first neuron.

So, I wonder if there are limitations on the kind of patterns that can be recalled? In any case, it would be great to show recall for other types of patterns as well.

In reply to this point, the authors have added a network simulation with a constrained connectivity. This network does indeed learn to recall a sequence in a compressed manner. However, the network connectivity is very much tailored to achieve this result. The neurons in the network receive disjunct inputs and the neurons are connected in a chain which is consistent with the order of the target activation order of the neurons in the recall. This setup cannot be considered a real test of the learning rule for the task. The authors should consider a recurrent network with random connectivity both for the input connections and for the recurrent connections.

(2) A remark to the reply to point 4 of Reviewer 1 and to the general derivation of the learning rule.

The authors state that because the objective is defined locally at the single neuron level, one can avoid the backpropagation of gradients through the recurrent interactions of the neurons.

This is not really true. Even though the objective is defined locally, the true gradient needs to be backpropagated if one considers a recurrent network of neurons. This is because in the recurrent network, the output of the neuron influences the activity of other neurons which in turn influence the inputs to the neuron and its membrane potential. This was ignored by the authors and confused me on

first reading. In the derivation of the learning rule, the authors assume that the inputs to the neuron are fixed (i.e., independent from the output of the neuron). This was never stated in the derivation.

In the derivation on page 12, I am not sure whether eqs. (8) and (9) are correct. I think the authors arrive at the correct result (with the assumption noted above), but doesn't $\nabla_w L_t$ also need to include the dependence of v_{t-1} on w ?

There is a small typo in eq. (2): L_t on the left hand side should be L .

I have some additional concerns about the manuscript:

(3) The learning rule is biologically questionable.

The authors state several times that the learning rule is local. Of course, this is the case if one defines a local rule as a rule that uses only information available within the neuron. But there is more to locality in biology. There, the question is also what quantity is available to each synapse. In the proposed learning rule, the critical quantity is the inner product between the prediction error p_t and the weight vector w . This quantity depends on all inputs and all weights in a non-trivial manner and has to be available locally to each synapse. I have a hard time to imagine how this could be implemented in a biological neuron. In that sense, I would say that the learning rule is highly speculative.

(4) In contrast, refs [77] and [78] of the manuscript use simple STDP achieving presumably similar results. The authors note in the Discussion that their rule can achieve this independently from the initial weights. This is a valid point, as the proposed rule has some homeostatic effect, i.e., it increases predictive inputs independently of output spikes. I wonder if one could simplify their rule in order to arrive at a more plausible STDP-like rule with homeostasis that has a similar effect.

(5) The comparison with STDP in Section "Emergence of spike-timing-dependent-plasticity rules" is questionable. STDP is defined on the temporal difference between a pre-synaptic spike and a post-synaptic spike, while the proposed rule is completely independent of post-synaptic spikes. Hence, any experiment where a pre-synaptic spike is paired with post-synaptic spikes via current injection cannot be reproduced.

In summary, the authors did a good job to revise their manuscript according to the remarks of Reviewer 1, although their treatment of point 3 did not convince me. The learning rule is quite interesting, but my main concern is its biological plausibility and its weak link to STDP.

REVIEWER 3

The authors have addressed most of the questions of Reviewer 1. The manuscript is now well organized and understandable. I will discuss a few points with which I was not fully satisfied and add some additional comments about the work.

We thank the reviewer for the comments, which helped to improve the manuscript. The additions to the manuscript are colored in red. Please note that, due to formatting, the numbering of the equations in this document might not be the same as in the revised manuscript.

(1) Reviewer 1, point 3: The Figure 3 results of “recall of learned sequences” are not convincing. Figure 3b bottom seems to show that all neurons are learning to spike together, starting with the ones with larger indexes, and progressively more so with more training. With more training, it seems that the output spike of the first neuron would eventually be treated as the first spike of a “pattern” by all other neurons, and other neurons will generate their output spikes when they receive the output spike from the first neuron. So, I wonder if there are limitations on the kind of patterns that can be recalled? In any case, it would be great to show recall for other types of patterns as well.

In reply to this point, the authors have added a network simulation with a constrained connectivity. This network does indeed learn to recall a sequence in a compressed manner. However, the network connectivity is very much tailored to achieve this result. The neurons in the network receive disjunct inputs and the neurons are connected in a chain which is consistent with the order of the target activation order of the neurons in the recall. This setup cannot be considered a real test of the learning rule for the task. The authors should consider a recurrent network with random connectivity both for the input connections and for the recurrent connections.

We thank the reviewer for these comments, which led to several new analyses and clarifications in the revised manuscript, obtaining similar results as in the original Figure 3.

1. We clarified the specific choice of the network structure in the revised version of the manuscript.

As stated in the original manuscript, we aimed to reproduce the anticipation and (stimulus-evoked) sequence recall as observed in the primary visual cortex (V1) of rats [1]. In this work, the authors showed that the repeated presentation of a sequence of flashes (at different retinotopic locations) gradually leads to a reorganization of spiking activity in the order of the presented sequence. The specific choice of the network architecture in the original manuscript provides a simplified model of the retinotopic structure of rat V1. In this network, neurons received independent flashes coming at different receptive fields, and they are connected following a simplified retinotopic structure with recurrent excitation between near-by receptive fields. We clarified the specific choice of the network architecture in the Results section:

” We defined the timing of external inputs to each neuron in the network, and the recurrent connections between neurons following a simplified retinotopic structure with recurrent excitation between near-by receptive fields. In particular, ”

2. We performed novel simulations as suggested by the reviewer. We show the results of the simulations in the novel Supplementary Figure S8 (see next page).

We simulated a recurrent neural network with all-to-all, random connections, and each neuron in the network received a random subset of the inputs. We show that we obtained similar results as with the network structure of Figure 3. In particular, the network with random connectivity showed sequential firing upon the presentation of the first inputs, and that sequential firing took place at a compressed timescale. The learning rule led to a reorganization of synaptic weights comparable to the results of Figure 3. We also investigated how our results depend on the total amount of random inputs received by each neuron (Figure S8e). Intuitively, increasing the number of random subsets of inputs to each neuron

decreases the likelihood of a full recall of the input sequence. We edited the text in the Result section accordingly:

” We also studied a network scheme where each neuron received a random subset of the pre-synaptic inputs, that is, we did not enforce a sequential activation of the neurons consistent with the sequential order of the pre-synaptic firing. Furthermore, the network had a random, all-to-all connectivity scheme (Figure S8). Similarly to the results of Figure 3, the network exhibited a reorganization of the synaptic weight which led to the recall of the full sequence with a compressed timescale. These results were dependent on the total number of input subsets to each neuron in the network (Figure S8e). ”

Figure S8: Anticipation and recall of sequences with randomly connected network. Relates to Figure 3 of the main text. We explored the dynamics of a network model where $N = 8$ neurons received an input sequence distributed across a total of 16 pre-synaptic neurons. The pre-synaptic neurons are divided in 8 subsets of 2 pre-synaptic neurons that fired sequentially with a relative delay of 2 ms. The sequence onset of pre-synaptic inputs for the $j + 1$ -th subset started 8 ms after the sequence onset of the j -th subset. For example, the pre-synaptic neurons of the first subset fired at 2 and 4 ms, the pre-synaptic neurons of the second subset fired at 10 and 12 ms, etc. Each epoch contains two different sources of noise: (1) jitter of the spike times (random jitter between -2 and 2 ms, which is applied randomly in each epoch); and (2) random firing following an homogeneous Poisson process with rate $\lambda = 5$ Hz. Each neuron in the network received randomly chosen input subsets. The network had a recurrent, all-to-all connectivity scheme, and the recurrent connections were initialized at random following a homogeneous distribution between 0 and a maximal value. Accordingly, the pattern that each neuron tries to predict is composed of the random subsets of pre-synaptic inputs and the internally generated activity of the network. **a)** Illustration of the network model, where only 5 neurons of the network are represented for simplicity. **b)** The synaptic weight matrix before learning and at the end of learning (epoch 3000). The bottom 8 entries of the i -th column correspond to the synaptic weights from the other neurons to the i -th neuron in the network. The top 16 entries correspond to the weights for the pre-synaptic inputs to the i -th neuron in the network, where only the inputs from the randomly chosen subsets had non-zero values. **c)** Raster plot of the network's activity during different epochs of training: (1) The “before” case, where only the pre-synaptic neurons corresponding to the first two input subsets exhibited sequential firing. (2) The “learning” or conditioning case, where we presented the entire sequence (which was repeated 3000 times). (3) The “after” or “recall” condition, which was the same as the before condition (now after learning). (4) Same as (3), but an example where spontaneous recall occurs due to the background stochastic firing. The neurons are ordered following their firing times. **d)** Evolution of the duration of network activity across epochs. We computed the temporal difference between the last spike of the last neuron and the first spike of the first neuron to estimate the total duration of the network's activity. The panel shows the mean and standard deviation computed over 100 different simulations. **e)** Left: percentage of neurons in the network that were active after learning (epoch 3000), as a function of the number of random subsets of inputs to each neuron in the network. Here, only the pre-synaptic neurons corresponding to the first two input subsets exhibited sequential firing, as in the “before” case in panel c. Right: duration of the network activity as a function of the number of random subsets of inputs to each neuron in the network. We computed the duration of the network activity as in panel d.

(2) A remark to the reply to point 4 of Reviewer 1 and to the general derivation of the learning rule.

The authors state that because the objective is defined locally at the single neuron level, one can avoid the backpropagation of gradients through the recurrent interactions of the neurons. This is not really true. Even though the objective is defined locally, the true gradient needs to be backpropagated if one considers a recurrent network of neurons. This is because in the recurrent network, the output of the neuron influences the activity of other neurons which in turn influence the inputs to the neuron and its membrane potential. This was ignored by the authors and confused me on first reading. In the derivation of the learning rule, the authors assume that the inputs to the neuron are fixed (i.e., independent from the output of the neuron). This was never stated in the derivation.

We thank the reviewer for this observation. We clarified this point here and in the revised manuscript.

We agree with the reviewer that, in the original manuscript, we did not fully clarify how our learning rule is implemented in recurrent neural networks. The reviewer correctly pointed out that, in a recurrent neural network, one has to consider how synaptic weights changes affect the output spiking activity of a given neuron and, in turn, how this affects the inputs received by other neurons. The contributions to the gradients of those recurrent connections from a neuron j to a neuron i are proportional to the product of terms that have a discontinuous effect in time (at the moment of the output spikes):

$$\propto \frac{\partial s_{t,j}}{\partial s_{t,i}} \frac{\partial s_{t,i}}{\partial v_{t,i}},$$

where $s_{t,i}$ and $v_{t,i}$ are the output spike variable and the membrane voltage variable, respectively. The contribution to the gradient of these recurrent connections is non-negligible only if one considered surrogate-gradient descent methods, see ref [81] and the subsection "Jacobian and surrogate-gradient methods" in the Methods section of the manuscript. In our simulations of the recurrent neural networks, we neglected the contribution of those recurrent connections to the estimation of the gradient. Embedding these single-neuron learning dynamics in a recurrent network remains to be undertaken.

In the revised version of the manuscript, we now clarified this assumption in the "Sequence anticipation and recall in a network with recurrent connectivity" subsection of the Result section:

"The recurrent connections between neurons in the network have a discontinuous effect in time - at the moment of the output spikes - and thus their contribution to the gradient can be neglected (see Methods)."

and in the "Jacobian and surrogate gradient" subsection of the Methods section:

"In the network implementations of Figure 3 , Figure S7 and Figure S8 , each neuron in the network receives inputs from other neurons, and the associated recurrent connections are defined by the specific connectivity scheme. The contributions to the gradient of the recurrent connections from a neuron j to a neuron i in the network are proportional to

$$\propto \frac{\partial s_{t,j}}{\partial s_{t,i}} \frac{\partial s_{t,i}}{\partial v_{t,i}}, \quad (1)$$

and have a discontinuous effect in time (at the moment of the output spikes). Thus, we neglected these contributions to the gradient."

In the revised version of the manuscript, we edited the following sentences from the Methods section

(line 443)

"Different from other works [80], our loss function (7) is defined locally at the single neuron level and we thus directly avoid the problem of backpropagation of the gradient through the recurrent interactions between neurons. Therefore, our approximated learning rule is completely online as it only requires information available at time step t ." → "Our approximated learning rule is completely online as it only requires information available at time step t ."

In the derivation on page 12, I am not sure whether eqs. (8) and (9) are correct. I think the authors arrive at the correct result (with the assumption noted above), but doesn't $\nabla_w L_t$ also need to include the dependence of v_{t-1} on w ?

We clarify this point here and show that our derivations are correct.

The reviewer correctly mentioned that one has to consider how the loss \mathcal{L} depends directly and indirectly on \vec{w} for an exact calculation of the gradient. In the following, we show that Equation (8) and (9) are correct.

The Equation (7) in the manuscript can be further specified as follows

$$\mathcal{L} = \sum_{t=0}^T \mathcal{L}_t = \sum_{t=0}^T \mathcal{L}_t(\vec{x}_t, v_{t-1}(\vec{w}), \vec{w}) = \sum_{t=0}^T \frac{1}{2} \|\vec{x}_t - v_{t-1}(\vec{w}) \vec{w}\|_2^2. \quad (2)$$

Here, \vec{x}_t is independent of \vec{w} as the neuron received only external inputs, that is, inputs from pre-synaptic neurons to which the post-synaptic neuron does not project back. The gradient of \mathcal{L}_t w.r.t. \vec{w} contains a term for the direct contribution of \vec{w} and another term for the indirect contribution via the membrane potential $v_{t-1}(\vec{w})$, which is exactly Equation (8) in the main text:

$$\vec{\nabla}_{\vec{w}} \mathcal{L}_t = \sum_{t=0}^T \frac{1}{2} \left(\vec{\nabla}_{\vec{w}} \mathcal{L}_t + \frac{\partial \mathcal{L}_t}{\partial v_{t-1}} \vec{\nabla}_{\vec{w}} v_{t-1} \right). \quad (3)$$

The loss \mathcal{L}_t is a function of the product of two variables which depend on the synaptic weights \vec{w} . To calculate the two terms in the gradient $\vec{\nabla}_{\vec{w}} \mathcal{L}_t$, we proceed as follows.

First, we calculate the gradient of \mathcal{L}_t while keeping $v_t(\vec{w})$ fixed:

$$\vec{\nabla}_{\vec{w}} \mathcal{L}_t = \vec{\nabla}_{\vec{w}} \|\vec{x}_t - v_{t-1} \vec{w}\|_2^2 = \vec{\nabla}_{\vec{w}} (\vec{x}_t - v_{t-1} \vec{w})^\top (\vec{x}_t - v_{t-1} \vec{w}) = -2(\vec{x}_t - v_{t-1} \vec{w}) v_t, \quad (4)$$

which is Equation (9) in the original manuscript.

Second, we calculate the partial derivate of \mathcal{L}_t w.r.t. $v_{t-1}(\vec{w})$ and the gradient of v_{t-1} w.r.t. \vec{w} , while keeping \vec{w} fixed:

$$\frac{\partial}{\partial v_{t-1}} \mathcal{L}_t = \frac{\partial}{\partial v_{t-1}} \|\vec{x}_t - v_{t-1} \vec{w}\|_2^2 = \frac{\partial}{\partial v_{t-1}} (\vec{x}_t - v_{t-1} \vec{w})^\top (\vec{x}_t - v_{t-1} \vec{w}) = -2(\vec{x}_t - v_{t-1} \vec{w})^\top \vec{w}, \quad (5)$$

then,

$$\vec{\nabla}_{\vec{w}} v_t = \left(\alpha - v_{(\text{th})} \frac{\partial s_{t-1}}{\partial v_{t-1}} \right) \vec{\nabla}_{\vec{w}} v_{t-1} + \vec{x}_t, \quad (6)$$

which is the term that includes the dependence of v_{t-1} on \vec{w} .

There is a small typo in eq. (2): L_t on the left hand side should be \mathcal{L} .

We corrected the typo mentioned by the reviewer.

I have some additional concerns about the manuscript:

We thank the reviewer for these helpful additional comments. We hope that our clarification of some of the points above and below already takes away some of the reviewer’s concerns. In the revised version of the manuscript, we have also added a novel Supplementary Material where we recapitulate parts of our clarifications to the reviewer. The Supplementary Material is shown at the end of this document.

(3) The learning rule is biologically questionable. The authors state several times that the learning rule is local. Of course, this is the case if one defines a local rule as a rule that uses only information available within the neuron. But there is more to locality in biology. There, the question is also what quantity is available to each synapse. In the proposed learning rule, the critical quantity is the inner product between the prediction error p_t and the weight vector w . This quantity depends on all inputs and all weights in a non-trivial manner and has to be available locally to each synapse. I have a hard time to imagine how this could be implemented in a biological neuron. In that sense, I would say that the learning rule is highly speculative.

We thank the reviewer for this comment. We have added a discussion paragraph to the manuscript to address this point.

We wish to emphasize that we have proposed a computational theory of predictive learning in single neurons, which leads to sequence anticipation and efficient encoding of inputs. Importantly, the predictive learning rule does reproduce, as we have extensively discussed in the previous manuscript, many of the experimentally observed STDP phenomena, including: (a) the frequency-dependence of STDP, (b) the typical STDP kernel as well as symmetrical kernels, (c) the dependence of plasticity on initial synaptic weight, and (d) plasticity for higher-order patterns. This yields a novel computational and functional interpretation of these empirically observed STDP phenomena based on the principle of predictive learning, which at the same time suggests that the learning rule has a biological substrate.

The question of the reviewer pertains to how the mathematical form of the predictive learning rule would be implemented in a biological neuron. Specifically, the review comments on the point that the weight update at a single synapse depends in a non-trivial manner on the inner product of prediction errors $\vec{\epsilon}_t$ with the weights \vec{w}_t at the other synapses. This is a great question, and we clarify several points:

1. The reviewer correctly pointed out that the second term in the proposed learning rule is given by

$$(\vec{\epsilon}_t^\top \vec{w}_{t-1}) p_{t-1,j} \equiv \mathcal{E}_t p_{t-1,j}. \quad (7)$$

It is important to note that this yields a scalar, global term that we re-defined as \mathcal{E}_t , i.e. the specific information of $\vec{\epsilon}_t$ and \vec{w}_{t-1} does not have to be available at each synapse. This is, in fact, an important and interesting aspect of the predictive learning rule that suggests it can be implemented inside a single neuron and at the level of a synapse, as all that is needed is a *global* signal which interacts with a local trace of pre-synaptic inputs. Such interactions can be well implemented at local synapses through coincidence detection, e.g. the NMDA receptor.

It is possible that there is a specific molecular pathway for the global signal \mathcal{E}_t , however it is interesting to note that we can express the global signal \mathcal{E}_t in terms of voltage changes and voltages, which can be available globally and interact with local inputs at the level of individual synapses. Importantly, such global signals are also typically assumed in other phenomenological state-of-the-art models of STDP, like the Clopath rule [2]. Interestingly, the form of \mathcal{E}_t entails that transient, unpredicted increases in the voltage contribute to LTP, whereas more sustained depolarizations contribute to LTD. In principle, this fits well with the notion that the temporal pattern of calcium changes rather than the absolute level of calcium is an important determinant of plasticity [3, 4, 5].

In the revised version of the manuscript, we now included more discussion on the biological substrate of the proposed learning rule, and the mechanisms available at the single neuron level that might support it:

”The learning rule in Equation (3) depends on synaptic mechanisms that are biologically plausible, as they rely on information that is locally available at the level of single neurons. In this model, synaptic plasticity depends on the interaction between synaptic variables and global signals, that in turn depend on the pre- and post-synaptic activity (the prediction errors $\vec{\epsilon}_t$) and the strength of the synapses \vec{w}_{t-1} . These processes can be implemented with local mechanisms such as NMDA receptors, voltage-gated calcium channels (VGCCs) [6, 7], and synaptic interactions via intracellular signals or membrane depolarizations [8, 9, 10, 11, 12, 13]. In particular, the second term in Equation (3) defines the interaction of a local trace of pre-synaptic inputs with a global, post-synaptic term \mathcal{E}_t . This global term entails that transient, unpredicted increases in the voltage contribute to LTP, whereas more sustained depolarizations contribute to LTD. In fact, experimental evidences show that different molecular pathways depending on global, post-synaptic variables - e.g. membrane depolarization [14, 15], intracellular $[Ca^{2+}]$ transients [3] - are key in determining the sign and amplitude of synaptic plasticity. ”

Accordingly, we also added the new definition of \mathcal{E}_t into the Results and Methods section.

The precise mapping of the learning rule detailed here to the level of receptor interactions etc. will be extremely important, but is best addressed in future work. It is possible that this calls for a modification or extension (with more parameters) of the predictive learning rule proposed here. It is important to note here, though, that we still have a somewhat limited understanding of the molecular basis of STDP, in particular how the temporal pattern of synaptic inputs results in plasticity changes [16, 17, 18]. Also, the STDP research that has so far been conducted may not necessarily mimic the mechanisms for plasticity in vivo [13, 19, 10]. Hence, a biologically realistic description of a learning rule at a molecular level for *in vivo* plasticity requires further research effort. Nevertheless, a computational framework and learning rule as proposed here can be of great significance to interpret and design experiments, but also in the implementation of unsupervised learning in recurrent neural network models.

2. We fully agree with the reviewer that there can exist different definitions of “locality” and “local information”. Information can be local at the level of a single neuron, and it can be local at the level of single synapses. In the revised version of the manuscript, we now clarified that we refer to the proposed learning rule as “local” because it only requires information available within the neuron. We edited the text of the whole manuscript accordingly:

(line 29) *”Based on this principle, we derive a local predictive learning rule.”* → *”Based on this principle, we derive a predictive learning rule.”*

(line 31) We removed the line *”The mismatch between the present input and its prediction is computed locally at every synapse.”* from the caption of Figure 1.

(line 49) *”This mismatch can be interpreted as a prediction error, which can be computed with local synaptic states and in real-time based on the dynamics of the inputs (see Methods).”* → *”This mismatch can be interpreted as a prediction error, which can be computed with information available within the neuron and in real-time based on the dynamics of the inputs (see Methods).”*

(line 63) *”Accordingly, the prediction of future inputs can be computed at the synaptic level in the point-neuron approximation based only on locally available information.”* → *”Accordingly, the prediction of future inputs can be computed at the synaptic level in the point-neuron approximation based only on information available within the single neuron.”*

(line 246) *”That is, synapses are potentiated to the degree that they are predictive of future input states, which provides a solution to an optimization problem that can be implemented at the single-neuron level using only local information.”* → *”That is, synapses are potentiated to the degree that they are predictive of future input states, which provides a solution to an optimization problem that can be implemented using only information available at the single-neuron level.”*

(4) In contrast, refs [77] and [78] of the manuscript use simple STDP achieving presumably similar results. The authors note in the Discussion that their rule can achieve this independently of the initial weights. This is a valid point, as the proposed rule has some homeostatic effect, i.e., it increases predictive inputs independently of output spikes. I wonder if one could simplify their rule in order to arrive at a more plausible STDP-like rule with homeostasis that has a similar effect.

We thank the reviewer for the interesting comments and suggestions. We provide here some additional clarifications and discussion for the reviewer:

First, we wish to highlight that the proposed plasticity rule accounts for a wide range of phenomena further than sequence anticipation, as we extensively discussed in the original manuscript. While it might be possible to derive a more "STDP-like" version of the proposed learning, it is unlikely that all the results described in the manuscript will still hold true. Nonetheless, the reviewer raised a very interesting point, and we wish to address possible "STDP-like" modifications of the proposed learning rule in future work. Here, we further discuss what STDP models can and cannot reproduce in terms of the results presented in the original manuscript.

1. The predictive learning rule described in the main text is capable of generating different types of STDP kernels, such as asymmetrical or symmetrical. This would not be possible if one would consider a specific STDP-like rule with either a symmetrical or anti-symmetrical kernel (as done in [77] and [78]).
2. Typically, stability mechanisms can abolish the competitive interaction between synapses that is usually at the basis of STDP phenomena [20, 21, 18], such as the decrease in output latency in [77] and [78]. Such competition is required to reproduce the decrease in the latency of response as observed in these works.
3. The predictive learning rule is determined by the gradient of an objective function, leading to weight updates that converge to specific fixed points regardless of initial conditions. By that means, our model replicates the experimental observation that the amount of Long-Term Potentiation (LTP) depends on the initial strength of the synaptic input (Figure 1d and Figure S9), a phenomenon observed experimentally [22, 23], and not reproducible with STDP rules that do not directly depend on synaptic weights [21].

Second, it is important to note that the authors in [77] and [78] used a specific STDP rule with two internal homeostatic/regulatory mechanisms, rather than a simple STDP model. The Restricted Nearest-Neighbours (Restricted-NN) STDP rule was employed, where weight changes only occur for the first post-synaptic spike immediately following or preceding a pre-synaptic spike, resulting in alternations between potentiation and depression in time. Additionally, the authors used a specific STDP window that is biased towards depotentiation. These regulatory/homeostatic mechanisms were necessary to reproduce the results in [77] and [78]. We refer to [24] for a more comprehensive analysis of the results in [78].

Finally, by definition, the Restricted-NN STDP rule only induces weight changes for the first post-synaptic spike immediately following or preceding a pre-synaptic spike. As a result, this plasticity rule is unable to replicate how plasticity behaves with higher-order spike patterns or how plasticity is influenced by the frequency of pre-post pairings. In contrast, the predictive plasticity rule proposed in this manuscript is capable of reproducing both of these effects.

Although these discussions are interesting, we note that the results of [77] and [78] are not the main subject of our paper and we feel that further discussion related to that work, in the interest of space, is better left for future work.

(5) The comparison with STDP in Section "Emergence of spike-timing-dependent-plasticity rules" is questionable. STDP is defined on the temporal difference between a pre-synaptic spike and a post-synaptic spike, while the proposed rule is completely independent of post-synaptic spikes. Hence, any experiment where a pre-synaptic spike is paired with post-synaptic spikes via current injection cannot be reproduced.

We thank the reviewer for this comment. We add here some additional clarifications and discussion for the

reviewer.

As we significantly discussed in the original and the revised manuscript, the predictive learning rule proposed in the manuscript has a strong link with STDP. We elaborate in the following:

1. The proposed learning rule does reproduce several experimental observations of STDP, both qualitatively and quantitatively. Accordingly, the phenomenological model that can, to our knowledge, reproduce the majority of experimental observations, indeed involves the membrane potential as a pivotal variable for STDP (the work from Clopath et al [2]) (see also the Discussion section in the original manuscript, lines 320-329).
2. Experimental evidence suggests that STDP is more broadly defined by changes in post-synaptic membrane potential that occur at different moments in time [8, 9, 10, 11, 12, 13], and that in turn can trigger different molecular cascades [25, 3, 6, 5]. In STDP protocols *in-vitro*, synaptic plasticity depends on the timing between voltage changes due to pre-synaptic inputs, and voltage changes due to current injection and associated back-propagating action potential (bAP). The proposed learning rule depends on the temporal relations between voltage changes caused by different pre-synaptic inputs, and other sources of voltage changes - such as current injection, or bAPs - can also influence plasticity. By its very nature, the predictive learning rule can thereby reproduce STDP mechanisms in line with experimental evidence *in-vitro*, as well as give rise to timing-dependent phenomena, such as sequence anticipation and sequence recall at compressed timescales, that might take place *in-vivo* [26, 27].
3. The predictive learning rule is not completely independent of post-synaptic spikes. On the contrary, it crucially depends on the temporal relationships between inputs, where LTP or LTD is observed depending on the timing of the post-synaptic spikes. Every time a spike is emitted, the membrane potential receives negative feedback mimicking the spiking reset mechanism. This reset mechanism crucially influence learning, and it is at the basis of our main results, see for example Figure 1c and Figure 1d.

We also wish to emphasize that we did not construct a learning rule to reproduce experimentally observed STDP phenomena. Rather, we derived a learning rule from an optimization problem based on prediction of future inputs, and this learning rule gave rise to experimentally observed STDP mechanisms.

Furthermore, in the original manuscript, we extensively discussed how our results in the section "Emergence of spike-timing-dependent plasticity rules" relate to STDP experiments and models (lines 282-333). We argued about how our results relate to *in-vitro* findings, and what are the major limitations of interpreting and modelling STDP experiments. Importantly, STDP models that solely depend on the temporal difference between pre- and post-synaptic spikes are typically limited in terms of explaining STDP phenomena [17]. Several studies have pointed out that different post-synaptic signals (depolarization level, dendritic spikes, calcium concentration) are relevant for STDP, and this variability cannot be model with simple STDP rules [22, 10, 28, 16]. As we discussed in the original version of the manuscript (lines 290-298), it is still a manner of debate what is the crucial post-synaptic signal for plasticity [13].

In summary, the authors did a good job to revise their manuscript according to the remarks of Reviewer 1, although their treatment of point 3 did not convince me. The learning rule is quite interesting, but my main concern is its biological plausibility and its weak link to STDP.

We thank the reviewer for their helpful comments and positive feedback. We have addressed in detail the point of biological plausibility above (see point 3). Furthermore, as explained extensively in the previous and revised manuscript (see point 5), there is a strong link between the predictive learning rule and STDP.

In the revised version of the manuscript, we now included a novel Supplementary Material where we recapitulate parts of our comments to the reviewer.

”

1. Relations between the predictive learning rule and spike-timing dependent plasticity (STDP)

1.1. Relations with STDP models

We further discuss what are the relations between the predictive plasticity rule described in the main text and phenomenological models for STDP. In particular, we focus on what STDP models can and cannot reproduce in terms of the results presented in the main text.

1. STDP rules lead to unlimited increase or decrease of synaptic weights and additional stability mechanisms are typically required [29, 30, 31]. For example, it is common to artificially limit the set of possible weight values or to include homeostatic plasticity rules [20, 31]. However, stability mechanisms can abolish the competitive interaction between synapses that is usually at the basis of STDP phenomena [20, 21, 18], such as the decrease in output latency [32, 33, 34]. A specific combination of STDP rules and regulatory mechanisms is usually required to obtain specific results. For example, for the decrease in the response latency for incoming input spike trains in [33, 34], the authors used an STDP rule with a specific learning window, and with internal regulatory mechanisms: (1) the authors used the Restricted Nearest-Neighbours (Restricted-NN) STDP rule, where a weight change occurs only for the first post-synaptic spike immediately following or preceding a pre-synaptic spike - that is, potentiation and depression alternates in time - and (2) the authors used a specific STDP window which is biased towards depotentiation. These regulatory/homeostatic mechanisms are necessary to reproduce the results in [24]. To the best of our knowledge, [33, 34] are the only works showing anticipation of spike sequences similarly to the results presented in the main text.
2. Experimental evidence has shown that higher-order spike patterns (beyond pre-post pairing) during STDP have diverse effects [22, 35, 36, 17]. STDP rules with restricted spike-spike interactions only allow weight updates for a limited number of coincidences between pre- and post-synaptic spikes. As a result, these STDP rules are unable to replicate how plasticity behaves with higher-order spike patterns or how plasticity is influenced by the frequency of pre-post pairings [37]. In contrast, the predictive plasticity rule proposed in this manuscript is capable of reproducing both of these effects. Only unrestricted STDP rules that explicitly consider higher-order spike interactions [38] or explicitly depend on the post-synaptic membrane voltage [2] can also account for higher-order STDP effects.
3. The predictive learning rule described in the main text is capable of generating different types of STDP kernels, such as asymmetrical or symmetrical. This cannot be possible if one would consider STDP-like learning rules, as they are defined by specific learning windows [33, 34, 21].
4. The predictive learning rule is determined by the gradient of an objective function, leading to weight updates that converge to specific fixed points regardless of initial conditions. By that means, our model replicates the experimental observation that the amount of Long-Term Potentiation (LTP) depends on the initial strength of the synaptic input, a phenomenon observed experimentally [22, 23], and not reproducible with STDP rules that do not directly depend on synaptic weights [21].

1.2. Relations with STDP experiments

We further discuss the link between the predictive learning rule described in the main text and STDP experiments in both in-vivo and in-vitro.

1. Experimental evidence suggests that STDP is more broadly defined by changes in the post-synaptic membrane potential that occur at different moments in time [8, 9, 10, 11, 12, 13], and that in turn can trigger different molecular cascades [25, 3, 6, 5]. In STDP protocols in-vitro, synaptic plasticity depends on the timing between voltage changes due to pre-synaptic inputs, and voltage changes due to current injection and associated back-propagating action potential (bAP). The proposed learning rule depends on the temporal relations between voltage changes caused by different pre-synaptic inputs, and other sources of voltage changes - such as current injection, or bAPs - can also influence plasticity. By its very nature, the predictive learning rule can thereby reproduce STDP mechanisms in line with experimental evidence in-vitro, as well as give rise to timing-dependent phenomena, such as sequence anticipation and sequence recall at compressed timescales, that might take place in-vivo [26, 27].

2. *The predictive plasticity rule does reproduce several experimental observations of STDP, both qualitatively and quantitatively. Accordingly, the phenomenological model that can, to our knowledge, reproduce the majority of experimental observations, indeed involves the membrane potential as a pivotal variable for STDP (the work from Clopath et al [2]) (see also the Discussion section in the main text).*
3. *The predictive learning rule is not independent of post-synaptic spikes. On the contrary, it crucially depends on the temporal relationships between inputs, where LTP or LTD is observed depending on the timing of the post-synaptic spikes. Every time a spike is emitted, the membrane potential receives negative feedback, mimicking the spiking reset mechanism. This reset mechanism crucially influences learning, and it is one of the bases of the results in the main text.*

”

References

- [1] Shengjin Xu, Wanchen Jiang, Mu-ming Poo, and Yang Dan. Activity recall in a visual cortical ensemble. *Nature neuroscience*, 15(3):449–455, 2012.
- [2] Claudia Clopath, Lars Büsing, Eleni Vasilaki, and Wulfram Gerstner. Connectivity reflects coding: a model of voltage-based stdp with homeostasis. *Nature neuroscience*, 13(3):344, 2010.
- [3] Shao-Nian Yang, Yun-Gui Tang, and Robert S Zucker. Selective induction of ltp and ltd by postsynaptic [ca²⁺] i elevation. *Journal of neurophysiology*, 81(2):781–787, 1999.
- [4] P Jesper Sjostrom, Ede A Rancz, Arnd Roth, and Michael Hausser. Dendritic excitability and synaptic plasticity. *Physiological reviews*, 88(2):769–840, 2008.
- [5] Michael Graupner and Nicolas Brunel. Calcium-based plasticity model explains sensitivity of synaptic changes to spike pattern, rate, and dendritic location. *Proceedings of the National Academy of Sciences*, 109(10):3991–3996, 2012.
- [6] John Lisman, Ryohei Yasuda, and Sridhar Raghavachari. Mechanisms of camkii action in long-term potentiation. *Nature reviews neuroscience*, 13(3):169–182, 2012.
- [7] Yang Dan and Mu-Ming Poo. Spike timing-dependent plasticity: from synapse to perception. *Physiological reviews*, 86(3):1033–1048, 2006.
- [8] Massimo Scanziani, Robert C Malenka, and Roger A Nicoll. Role of intercellular interactions in heterosynaptic long-term depression. *Nature*, 380(6573):446–450, 1996.
- [9] Alain Artola and Wolf Singer. Long-term depression of excitatory synaptic transmission and its relationship to long-term potentiation. *Trends in neurosciences*, 16(11):480–487, 1993.
- [10] Nace L Golding, Nathan P Staff, and Nelson Spruston. Dendritic spikes as a mechanism for cooperative long-term potentiation. *Nature*, 418(6895):326–331, 2002.
- [11] Sébastien Royer and Denis Paré. Conservation of total synaptic weight through balanced synaptic depression and potentiation. *Nature*, 422(6931):518–522, 2003.
- [12] Christopher D Harvey, Ryohei Yasuda, Haining Zhong, and Karel Svoboda. The spread of ras activity triggered by activation of a single dendritic spine. *Science*, 321(5885):136–140, 2008.
- [13] John Lisman and Nelson Spruston. Questions about stdp as a general model of synaptic plasticity. *Frontiers in synaptic neuroscience*, 2:140, 2010.
- [14] Alain Artola, S Bröcher, and Wolf Singer. Different voltage-dependent thresholds for inducing long-term depression and long-term potentiation in slices of rat visual cortex. *Nature*, 347(6288):69–72, 1990.
- [15] John Lisman and Nelson Spruston. Postsynaptic depolarization requirements for ltp and ltd: a critique of spike timing-dependent plasticity. *Nature neuroscience*, 8(7):839–841, 2005.
- [16] Harel Z Shouval, Samuel S-H Wang, and Gayle M Wittenberg. Spike timing dependent plasticity: a consequence of more fundamental learning rules. *Frontiers in computational neuroscience*, 4:19, 2010.
- [17] Daniel E Feldman. The spike-timing dependence of plasticity. *Neuron*, 75(4):556–571, 2012.
- [18] Marina Chistiakova, Nicholas M Bannon, Maxim Bazhenov, and Maxim Volgushev. Heterosynaptic plasticity: multiple mechanisms and multiple roles. *The Neuroscientist*, 20(5):483–498, 2014.
- [19] Jason Hardie and Nelson Spruston. Synaptic depolarization is more effective than back-propagating action potentials during induction of associative long-term potentiation in hippocampal pyramidal neurons. *Journal of Neuroscience*, 29(10):3233–3241, 2009.

- [20] Kenneth D Miller and David JC MacKay. The role of constraints in hebbian learning. *Neural computation*, 6(1):100–126, 1994.
- [21] Abigail Morrison, Markus Diesmann, and Wulfram Gerstner. Phenomenological models of synaptic plasticity based on spike timing. *Biological cybernetics*, 98(6):459–478, 2008.
- [22] Per Jesper Sjöström, Gina G Turrigiano, and Sacha B Nelson. Rate, timing, and cooperativity jointly determine cortical synaptic plasticity. *Neuron*, 32(6):1149–1164, 2001.
- [23] Guo-qiang Bi and Mu-ming Poo. Synaptic modifications in cultured hippocampal neurons: dependence on spike timing, synaptic strength, and postsynaptic cell type. *Journal of neuroscience*, 18(24):10464–10472, 1998.
- [24] P Hathway and DFM Goodman. [re] spike timing dependent plasticity finds the start of repeating patterns in continuous spike trains. *ReScience*, 4, 2018. doi: 10.5281/zenodo.1327348. URL <http://dx.doi.org/10.5281/zenodo.1327348>.
- [25] John Lisman. A mechanism for the hebb and the anti-hebb processes underlying learning and memory. *Proceedings of the National Academy of Sciences*, 86(23):9574–9578, 1989.
- [26] Kamran Diba and György Buzsáki. Forward and reverse hippocampal place-cell sequences during ripples. *Nature neuroscience*, 10(10):1241–1242, 2007.
- [27] Jeffrey P Gavornik and Mark F Bear. Learned spatiotemporal sequence recognition and prediction in primary visual cortex. *Nature neuroscience*, 17(5):732–737, 2014.
- [28] Per Jesper Sjöström and Michael Häusser. A cooperative switch determines the sign of synaptic plasticity in distal dendrites of neocortical pyramidal neurons. *Neuron*, 51(2):227–238, 2006.
- [29] Erkki Oja. Simplified neuron model as a principal component analyzer. *Journal of mathematical biology*, 15(3):267–273, 1982.
- [30] Larry F Abbott and Sacha B Nelson. Synaptic plasticity: taming the beast. *Nature neuroscience*, 3(11):1178–1183, 2000.
- [31] Mark CW Van Rossum, Guo Qiang Bi, and Gina G Turrigiano. Stable hebbian learning from spike timing-dependent plasticity. *Journal of neuroscience*, 20(23):8812–8821, 2000.
- [32] Sen Song, Kenneth D Miller, and Larry F Abbott. Competitive hebbian learning through spike-timing-dependent synaptic plasticity. *Nature neuroscience*, 3(9):919–926, 2000.
- [33] Rudy Guyonneau, Rufin VanRullen, and Simon J Thorpe. Neurons tune to the earliest spikes through stdp. *Neural Computation*, 17(4):859–879, 2005.
- [34] Timothée Masquelier, Rudy Guyonneau, and Simon J Thorpe. Spike timing dependent plasticity finds the start of repeating patterns in continuous spike trains. *PloS one*, 3(1):e1377, 2008.
- [35] Robert C Froemke and Yang Dan. Spike-timing-dependent synaptic modification induced by natural spike trains. *Nature*, 416(6879):433–438, 2002.
- [36] Robert C Froemke, Ishan A Tsay, Mohamad Raad, John D Long, and Yang Dan. Contribution of individual spikes in burst-induced long-term synaptic modification. *Journal of neurophysiology*, 2006.
- [37] Thomas Nevian and Bert Sakmann. Spine ca²⁺ signaling in spike-timing-dependent plasticity. *Journal of Neuroscience*, 26(43):11001–11013, 2006.
- [38] Jean-Pascal Pfister and Wulfram Gerstner. Triplets of spikes in a model of spike timing-dependent plasticity. *Journal of Neuroscience*, 26(38):9673–9682, 2006.

REVIEWERS' COMMENTS

Reviewer #3 (Remarks to the Author):

With the revision, the authors have improved the manuscript at several places.

The experiment on sequence anticipation in a randomly connected recurrent network better illustrates the capabilities of the learning rule. The authors have also improved the presentation of the derivation of the learning rule, making assumptions explicit.

On the positive side, the learning rule accomplishes the prediction of temporal structures in the input and the recall of simple temporal sequences. The learning rule is robust to initial conditions. The mathematical derivations are sound.

On the negative side, the learning rule is very complex. While the operations are neuron-local, there is no experimental evidence that such involved mathematical operations are implemented in biological neurons.

I am still not convinced by the claims of the manuscript with respect to STDP. Figure 4b shows STDP curves that are similar to the ones known from experiments. However, the Δt on the x-axis refers to a time difference between two presynaptic spikes, whereas in the experimental literature it refers to the time difference between a presynaptic and a postsynaptic spike. It is true that experimental results indicate that it is really the postsynaptic potential that is the relevant variable (instead of the postsynaptic spike). Still, in this case, the investigations in the manuscript should be based on two variables, the presynaptic spike time and the postsynaptic voltage (instead of another presynaptic spike time).

Although the work is publishable, my assessment is that its significance in the field is limited.

REVIEWER 3

With the revision, the authors have improved the manuscript at several places.

The experiment on sequence anticipation in a randomly connected recurrent network better illustrates the capabilities of the learning rule. The authors have also improved the presentation of the derivation of the learning rule, making assumptions explicit.

On the positive side, the learning rule accomplishes the prediction of temporal structures in the input and the recall of simple temporal sequences. The learning rule is robust to initial conditions. The mathematical derivations are sound.

On the negative side, the learning rule is very complex. While the operations are neuron-local, there is no experimental evidence that such involved mathematical operations are implemented in biological neurons. I am still not convinced by the claims of the manuscript with respect to STDP. Figure 4b shows STDP curves that are similar to the ones known from experiments. However, the Δt on the x-axis refers to a time difference between two presynaptic spikes, whereas in the experimental literature it refers to the time difference between a presynaptic and a postsynaptic spike. It is true that experimental results indicate that it is really the postsynaptic potential that is the relevant variable (instead of the postsynaptic spike). Still, in this case, the investigations in the manuscript should be based on two variables, the presynaptic spike time and the postsynaptic voltage (instead of another presynaptic spike time).

Although the work is publishable, my assessment is that its significance in the field is limited.

We are glad the reviewer deem the manuscript publishable and are happy with the various positive comments. We thank the reviewer for the comments, which have helped tremendously to improve the manuscript.

As far as the discussion is concerned about the STDP kernel shown in Figure 4b: In response to the previous comments from reviewer 1, 2 and 3, we have already added extensive discussion about the biological realism of our learning rule, its relation to STDP, and the various agreements to experimental data in the current manuscript. In addition, in response to the original Reviewer 1, we also included new simulations in which a pre-synaptic input preceded a post-synaptic spike, in order to directly emulate the STDP experimental literature, obtaining similar results as in Figure 4b. The revised discussion also included extensive discussion on the voltage dependence of plasticity in the predictive learning rule and the dependence on post-synaptic spiking. Given these considerable improvements in previous revision rounds, we believe we have therefore adequately addressed the point concerning the relationship of the STDP kernel produced by the predictive learning rule and *in vitro* STDP experiments.